# Deconfined criticalities and dualities between chiral spin liquid, topological superconductor and charge density wave Chern insulator

Xue-Yang Song[1] and Ya-Hui Zhang[2]

[1]*Department of Physics,Massachusetts Institute of Technology, Massachusetts 02139, USA and*

[2]*Department of Physics and Astronomy, Johns Hopkins University, Baltimore, Maryland 21218, USA*

(Dated: September 12, 2023)

We propose bi-critical and tri-critical theories between chiral spin liquid (CSL), topological super-conductor (SC) and charge density wave (CDW) ordered Chern insulator with Chern number $C = 2$ on square, triangular and kagome lattices. The three CDW order parameters form a manifold of $S^2$ or $S^1$ depending on whether there is easy-plane anisotropy. The skyrmion defect of the CDW order carries physical charge $2e$ and its condensation leads to a topological superconductor. The CDW-SC transitions are in the same universality classes as the celebrated deconfined quantum critical points (DQCP) between Neel order and valence bond solid order on square lattice. Both SC and CDW order can be accessed from the CSL phase through a continuous phase transition. At the CSL-SC transition, there is still CDW order fluctuations although CDW is absent in both sides. We propose three different theories for the CSL-SC transition (and CSL to easy-plane CDW transition): a $U(1)$ theory with two bosons, a $U(1)$ theory with two Dirac fermions, and an $SU(2)$ theory with two bosons. Our construction offers a derivation of the duality between these three theories as well as a promising physical realization. The $SU(2)$ theory offers a unified framework for a series of fixed points with explicit $SO(5), O(4)$ or $SO(3) \times O(2)$ symmetry. There is also a transparent duality transformation mapping SC order to easy-plane CDW order. The CSL-SC-CDW tri-critical points are invariant under this duality mapping and have an enlarged $SO(5)$ or $O(4)$ symmetry. The DQCPs between CDW and SC inherit the enlarged symmetry, emergent anomaly, and self-duality from the tri-critical point. Our analysis unifies the well-studied DQCP between symmetry breaking phases into a larger framework where they are proximate to a topologically ordered phase. Experimentally the theory demonstrates the possibility of a rich phase diagram and criticality through closing the Mott gap of a quantum spin liquid with projective symmetry group.

## CONTENTS

## I. INTRODUCTION

Deconfined critical points with fractionalization have attracted lots of attention because they are beyond the conventional Landau symmetry breaking framework. One classic example is the deconfined quantum critical point (DQCP) between the Neel order and valence bond solid (VBS) order for spin 1/2 system on square lattice[1,2]. The same DQCP can also happen between a quantum spin Hall insulator and a topologically trivial superconductor[3,4]. In these examples the phases in both sides are conventional symmetry breaking phases and fractionalization happens only at the quantum critical point(QCP). In contrast, there is a different class of QCP between a fractional phase and a symmetry breaking phase where fractionalization exists already in one side. One simple example is the XY* transition between a $Z_2$ topological ordered phase and a superfluid phase[5–7]. Such a transition is driven by the condensation of a Higgs boson $\varphi$, which simultaneously kills the topological order and leads to the onset of the symmetry breaking because the bilinear of $\varphi$ represents symmetry breaking order parameter.

In this paper we unify the above two classes of DQCPs in one framework. We show that there can be direct transitions between each pair of these three phases: a $U(1)_2$ chiral spin liquid (CSL), a topological superconductor (SC) and a charge density wave (CDW) ordered

Chern insulator with Chern number $C = 2$. The phase diagrams are illustrated in Fig. 1. There are three CDW orders, they have momenta $\mathbf{Q} = (\pi, 0), (0, \pi), (\pi, \pi)$ on square lattice and $\mathbf{Q} = \mathbf{M}_1, \mathbf{M}_2, \mathbf{M}_3$ on triangular and kagome lattice, labeled as $(n_3, n_4, n_5)$. On square lattice, $(n_3, n_4)$ is rotated by $C_4$. On triangular or kagome lattice, $(n_3, n_4, n_5)$ is rotated to each other by $C_6$ rotation,forming a manifold of $S^2$. On square lattice, we call $(n_3, n_4)$ CDW$_{xy}$ and $n_5$ CDW$_z$ as an analog of the Neel order with easy-plane or easy-axis anisotropy in magnets. The superconductor order is labeled as $(n_1, n_2)$. The topological $(d + id)$ superconductor can be understood from condensation of the skyrmion defects of the CDW order, which carries charge $2e$ similar to the quantum Hall ferromagnetism with $C = 2$[8]. Therefore, the CDW-SC transition on triangular/kagome lattice is in the same universality class as the DQCP between the isotropic Neel and VBS order. On square lattice, the CDW$_{xy}$-SC transition is the same as the easy plane DQCP. These two DQCPs are known to have $SO(5)$ or $O(4)$ symmetry and self-duality[2]. This article offers a new understanding of these enlarged symmetries as inherited from the CSL-CDW-SC tri-critical points.

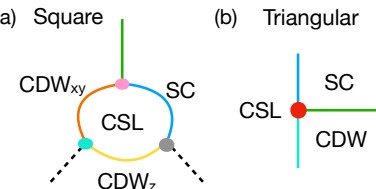

FIG. 1: Phase diagram on square (a) and triangular lattice (b), respectively. We will provide theories for the bi-critical (as boundaries between two phases) and tri-critical points(dots at intersection of three phases). The green line separating superconducting and CDW phases are described by isotropic or easy-plane DQCP. The dashed line represents first-order transitions.

We start from a chiral spin liquid and try to obtain either the SC or CDW phase through a continuous transition, focusing on half-filling. CSL phase[9,10] was proposed as an example of quantum spin liquid phase[11–14] in the early attempts to study high Tc superconductor through the resonating valence bond (RVB) mechanism[15]. It was thought that a superconductor phase can be reached from the CSL by closing the charge gap. A critical theory between the CSL and the SC phase has not been written down explicitly. A simple theory for this transition is through condensation of bosonic holons (eq (40)) starting from a mean field ansatz of the CSL phase. On square lattice, the ansatz for the fermionic spinon of the CSL phase is gauge equivalent to a $d + id$ superconductor, so a transition through slave boson condensation seems to be quite straightforward. Careful analysis shows that

this transition has a rich structure and enlarged symmetry. First, on square, triangular and kagome lattice, there are odd number of electrons per unit cell for the CSL phase with one electron per site. Thus the semion excitation is constrained to have a projective translation $T_1 T_2 = -T_2 T_1$. The fermionic spinon and bosonic holon inherit this projective translation and the minimal dimension for the irreducible representation of this projective translation is 2. Therefore, the critical theory has two bosons $\varphi_1, \varphi_2$ coupled to a $U(1)$ gauge field with a self Chern-Simons term at level $-2$. Similar to the XY* transition, bilinear terms of $\varphi = (\varphi_1, \varphi_2)^T$ represent the symmetry breaking orders. In addition to the SC order $\varphi_1^* \varphi_2$, we find three more gauge invariant order parameters from $|\varphi_1|^2 - |\varphi_2|^2$ and the monopole operator of the $U(1)$ gauge field. These three order parameters can be identified as the three CDW orders $(n_3, n_4, n_5)$. The appearance of the CDW order at the QCP is remarkable given that it is absent on both sides away from the QCP.

The CDW order turns out to have an enlarged $SO(3)$ symmetry at the QCP between CSL and SC, even though microscopically there is easy-plane anisotropy on square lattice. The emergent symmetry is best revealed in a dual theory with two Dirac fermions coupled to a $U(1)$ gauge field with self Chern-Simons term at level 1 (eq (45)). The dual theory can be derived from the standard boson fermion duality. Physically it is obtained by letting the bosonic holon $\varphi$ go through a pleateau transition into a bosonic integer quantum Hall (bIQHE) phase. In the dual theory the SC order is represented by the monopole operator. The three fermion bilinear terms $\bar{\psi}\vec{\sigma}\psi$ correspond to the CDW $(n_3, n_4, n_5)$. The $SO(3)$ symmetry of these three CDW order is transparent in the Dirac theory given that the four-fermion interaction term is irrelevant. The vortex of the SC order carries spin $1/2$ under this emergent $SO(3)$ symmetry, an emergent anomaly shared by the usual DQCP[16]. The existence of CDW order at the CSL-SC transition is required by the Lieb-Schultz-Mattis (LSM) theorem[17–19]. If the bilinear terms $\bar{\psi}\vec{\sigma}\psi$ do not carry any non-trivial quantum number of the lattice

symmetry, they can be added to the QCP. This leads to a trivial symmetric insulator, impossible with odd number of electrons per unit cell. So $\bar{\psi}\vec{\sigma}\psi$ must carry non-trivial symmetry quantum numbers. The same theory with two Dirac fermions has been proposed between a Laughlin state and a superfluid phase[20]. This is not a coincidence. Spin gap and single electron gap remain finite across the CSL-SC transition, so the transition is equivalent to the transition between Laughlin state and superfluid phase of Cooper pairs up to stacking of a $\nu = -2$ integer quantum Hall (IQHE) phase to account for the edge modes.

A parallel discussion follows for the transition between CSL and CDW$_{xy}$. The criticality is the same as the CSL-SC transition after exchanging the SC order and the CDW$_{xy}$ order. There is also an $SO(3) \times O(2)$ symmetry and the QCP can be described by $U(1)$ theory with either two bosons or two Dirac fermions. This analysis implies a duality which exchanges the SC and CDW$_{xy}$ order. Both CSL-SC and CSL-CDW$_{xy}$ transitions can be described by a theory with two bosons coupled to $U(1)$ gauge field with easy plane anisotropy $\lambda > 0$. The two theories are in the same form after a mapping between $(n_1, n_2)$ and $(n_3, n_4)$. If we tune $\lambda < 0$ in both theories, they describe the CSL to CDW$_z$ transition. So the CSL-CDW$_z$ QCP is self dual with $(n_1, n_2)$ exchanged with $(n_3, n_4)$. This implies an enlarged $O(4)$ symmetry rotating $(n_1, n_2, n_3, n_4)$. $\lambda = 0$ point of the two theories corresponds to the CSL-CDW$_{xy}$-CDW$_z$ and CSL-SC-CDW$_z$ tri-critical points. The CSL-CDW$_{xy}$-CDW$_z$ tri-critical point describes a CSL-CDW bi-critical point on triangular lattice because the easy-plane anisotropy term $\lambda$ is forbidden.

The best way to unify these various bi-critical and tri-critical points, and understand the enlarged symmetry and duality mapping is from an $SU(2)$ theory. The CSL phase can be understood as a $U(1)_2$ or $SU(2)_{-1}$ topological order. Especially the fermionic spinon can be put in an $SU(2)$ ansatz, from which the $U(1)$ ansatz descends through a Higgs term. Starting from the $SU(2)$ ansatz, a critical theory can be obtained with two bosons $\Phi_1, \Phi_2$ coupled to an $SU(2)$ gauge field $a$:

$$\mathcal{L}_{SU(2)} = \sum_{i=1,2} |(\partial_\mu - i a_\mu^s \tau^s - i\frac{1}{2} A_{c;\mu} \tau_0 \sigma_0)\Phi_i|^2 - r|\Phi|^2 + \frac{1}{4\pi} Tr[a \wedge da + \frac{2}{3} ia \wedge a \wedge a] - \frac{1}{8\pi} A_c dA_c - \mathcal{L}_{int}$$

$$\mathcal{L}_{int} = g|\Phi^\dagger \Phi|^2 + \lambda_0 \mathbf{n} \cdot \mathbf{n} - \lambda(n_1^2 + n_2^2) - \lambda'(n_3^2 + n_4^2), \tag{1}$$

where $A_c$ is the probe field for the electric charge, $\tau, \sigma$ Pauli matrices act in the $SU(2)$ spinor and flavor spaces respectively and $\mathbf{n} = (n_1, \cdots, n_5)$. The $SU(2)$ theory does not have monopole operators and all five order parameters can be written as the bilinear terms of $\Phi = (\Phi_1, \Phi_2)^T$. There are various fixed points in the

parameter space of quartic terms $(\lambda, \lambda')$ shown in fig 6. There is a duality between the $SU(2)$ theory and the $U(1)$ theory with two bosons for the CSL-SC transition. In particular there is presumably a manifold of $U(1)$ theory corresponding to different Higgs terms adding to the $SU(2)$ theory, to wit

$$\mathcal{L}_{SU(2)} = |(\partial_\mu - ia_\mu^s\tau^s - i\frac{1}{2}A_{c;\mu}\tau_0\sigma_0)\Phi|^2 - r|\Phi|^2 + \frac{1}{4\pi}Tr[a \wedge da + \frac{2}{3}ia \wedge a \wedge a] - \frac{1}{8\pi}A_c dA_c$$
$$- g|\Phi^\dagger\Phi|^2 - \lambda_0\mathbf{n}\cdot\mathbf{n} + \lambda(n_1^2 + n_2^2)$$
$$\leftrightarrow \mathcal{L}_{SU(2)} - h\Phi^\dagger\vec{m}\cdot\vec{\sigma}\tau_3\Phi$$
$$\leftrightarrow \mathcal{L}_{U(1),\vec{m}} = |(\partial_\mu - ia_\mu - i\frac{1}{2}A_{c;\mu}\sigma_3)\varphi|^2 - r|\varphi|^2 + \frac{2}{4\pi}ada - \frac{1}{8\pi}A_c dA_c - \tilde{g}(|\varphi|^2)^2 + \tilde{\lambda}|\varphi_1|^2|\varphi_2|^2.$$

$$(2)$$

The emergent $SO(3)$ symmetry generally acts non-locally in the $U(1)$ theory in the sense it changes one $U(1)$ theory to a different one. On triangular/kagome lattice, the lattice symmetry also needs to act non-locally in the $U(1)$ theory. Therefore there is no simple parton construction of the $U(1)$ theory for the CSL-SC transition on triangular/kagome lattice.

In the $SU(2)$ theory, there is an explicit duality mapping $\lambda \leftrightarrow \lambda'$ which exchanges $(n_1, n_2)$ and $(n_3, n_4)$. The special line $\lambda = \lambda'$ is self-dual and generically has an enlarged $O(4)$ symmetry. The special point $\lambda = \lambda' = 0$ has an $SO(5)$ symmetry. We argue that the CSL-CDW$_{xy}$-SC tri-critical point on square lattice is at the $\lambda = \lambda' = \lambda^*$ fixed point and thus is self dual with $O(4)$ symmetry. CSL-CDW-SC tri-critical point on triangular/kagome lattice is at $\lambda = \lambda' = 0$ and self-dual with $SO(5)$ symmetry. We also provide self-dual theory with two $U(1)$ gauge fields coupled to two bosons and two Dirac fermions for the tri-critical points. The self-duality and $SO(5)$ or $O(4)$ symmetry of the usual DQCP discussed in Ref. 2 can now be understood as inherited from the tri-critical points.

We discuss the possible experimental realizations. Chiral spin liquid has been found in (numerical simulations of) various spin $1/2$ models and Hubbard models[21–35] and also in $SU(N)$ model with $N > 2$[36–43]. Especially chiral spin liquids with spontaneous time reversal breaking were found in the intermediate $U/t$ regime of the simple Hubbard model[32] on triangular lattice and in spin $1/2$ model on kagome lattice[25,26]. Triangular lattice Hubbard model may be simulated in moiré superlattice with $U/t$ tuned by simple gating[44], offering a promising direction to search for chiral spin liquid and bandwidth tuned transitions into either superconductor or CDW phase proposed in this paper. Note both superconductor[45] and chiral CDW[33] were numerically observed from doping a chiral spin liquid. Therefore it is also interesting to search for chemical potential tuned transitions, which have dynamical exponent $z = 2$, but may still share similar structures as the critical theories we propose.

The outline of the paper is as follows: in Section II we lay out the topological order of CSL and prove that there is a unique symmetry fractionalization pattern on square, triangular and Kagome lattices, which paves the way to proposing duality for critical theories involving CSL on one side. In sec III,IV, the mean-field ansatz and projective symmetries with $SU(2), U(1)$ gauge group are discussed, respectively. The relation of $U(1)$ ansatz to $SU(2)$ is highlighted. Sec V attaches a $\nu = -2$ IQHE state of spinful electrons to trivialize the spin degrees of freedom throughout the transition, effectively rendering the elementary degrees of freedom bosonic. This helps to simplify the discussion. Sec VI,VII discusses the CSL-SC transition on square lattices described by $U(1)_{-2}$ $2\varphi$, and $U(1)_1$ $2\psi$, respectively. The duality between these two theories and consequently operator mapping are provided. Sec VIII studies the CSL-CDW transition and demonstrates a tricritical point for CSL-CDW$_{xy}$-CDW$_z$ on square lattices. Section IX and X discuss the $SU(2)$ critical theory that describe CSL-CDW (SC) transitions, its symmetries, various fixed points in Fig 6 and the duality to $U(1)_{-2}$ $2\varphi$ theory. Sec XI briefly reviews the DQCP between CDW insulator and SC. Sec XII then presents a tricritical point for CSL-CDW-SC transtion. Sec XIII comments on honeycomb lattice where at low energy only one bosonic mode ( or in dual theory one Dirac fermion) exists and there is no symmetry breaking order fluctuating at the QCP. Sec XIV discusses experimental signatures of the duality in critical theories. We conclude the paper by sec XV with various technical details delegated to appendices.

## II. CHIRAL SPIN LIQUID AND ITS SYMMETRY FRACTIONALIZATION ON DIFFERENT LATTICES

A useful way to describe a CSL phase is through the mean field ansatz of the fermionic spinon $f_{i;\sigma}$ from the parton construction: $\vec{S}_i = \frac{1}{2}f_{i;\sigma}^\dagger\vec{\sigma}_{\sigma\sigma'}f_{i;\sigma'}$. Here $\vec{\sigma}$ labels the Pauli matrices. In a CSL phase, the spinon $f_\sigma$ is put into a Chern insulator ansatz with Chern number $C = 1$ for each spin. However, the invariant gauge group (IGG) can be either $SU(2)$ or $U(1)$[46]. After integrating $f_\sigma$, we get either a $U(1)_{-2}$ gauge theory or a $SU(2)_{-1}$ gauge theory. Both describe the same topological order following the level-rank duality. The CSL phase here has anyons $I, s$ with $s$ as a spin $1/2$ semion. We note that there is a different type of CSL which has $f_\sigma$ in a $d + id$ superconductor ansatz and has only $Z_2$ IGG. Such a phase has four different anyons $I, e, m, \epsilon$. We will only discuss the first CSL with only two anyons as this is the one which was found by numerical simulations.

If we only care about topological order, the $U(1)$ and $SU(2)$ ansatz are clearly the same because of the equiv-

alence between $SU(2)_{-1}$ and $U(1)_2$ topological field theory. The next question is whether the $U(1)$ and $SU(2)$ ansatz correspond to the same phase and can be connected to each other without a phase transition. Given the topological order is the same, the only difference between them is the symmetry fractionalization. The symmetry fractionalization for the symmetric CSL phase turns out to be unique on various lattices. This was first proved on kagome lattice[47] and can be generalized to square and triangular lattice. The proof proceeded by considering a cylinder geometry with periodic boundary along $y$, with two-fold degeneracy $|1\rangle, |s\rangle$. $|s\rangle$ is obtained by nucleating a pair of semions and separating them to the edges. Operationally it is obtained by threading a flux $2\pi$ of $S^z$ along $y$: $|s\rangle = e^{i\sum_i S_i^z \frac{2\pi y_i}{L_y}}|1\rangle$, where $y_i$ is the y coordinate of the site $i$. Due to the $1/2$ spin-hall response of the C, a $S_z = \frac{1}{2}$ is pumped from $x = 0 \to x = L_x$ so that there is a semion at each end of the cylinder with $S_z = \pm\frac{1}{2}$.

Let us consider square lattices. Consider the algebraic relation for inversion around a *plaquette center* $I^2 = 1$ where $I$ inverses with respect to the middle point of the cylinder and leaves $|1\rangle, |s\rangle$ invariant. The quantum number of $I^2$ on a single semion $I^2 = \pm 1$ is $Z_2$ valued due to the fact that two semions fusion into the vacuum. Applying $I$ to $|s\rangle$ with one semion at each end, is equivalent to applying $I$ twice to a single semion. Hence $I^2$ is given by the quantum number of $|s\rangle$ under $I$ denoted as $Q_s(I) = \langle s|I|s\rangle$, relative to that of $|1\rangle$ denoted as $Q_1(I)$, i.e. $\frac{Q_s(I)}{Q_1(I)} = (I^2)_s$.

The $S^z$ flux $\phi$ is inverted by $e^{i\pi S^x}$ ($S^x$ the total $x$ component spin) and inversion also reverts the $S^z$ flux, during the adiabatic flux threading, the quantum number of $I_h e^{i\pi S^x}$ can be tracked and cannot change from $|0\rangle \to |s\rangle$, i.e.

$$\frac{Q_s(Ie^{i\pi S^x})}{Q_1(Ie^{i\pi S^x})} = (Ie^{i\pi S^x})_s^2 = 1. \qquad (3)$$

Given full $SO(3)_{spin}$ is preserved, spin rotation commutes with inversion around the plaquette center[48]. Hence $(I^2 e^{i2\pi S^x})_s = 1$. Since each semion carries spin$-1/2$, we get $(I^2)_s = -1$. Similar to the arguments in ref[47], translation $T_1 T_2 T_1^{-1} T_2^{-1} = -1$ follows from the Oshikawa's generalization[18] of Lieb-Schultz-Mattis theorem. Also we can get $(R_1 \mathcal{T})^2 = -1$ following Ref. [47]. The translation and reflection (combined with time reversal) are shown in fig 2(square),3(triangular).

The site-centered inversion $I_s$ on square and triangular lattices is a bit different due to the location of the branch cut for the $S^z$ flux[48]. When adiabatically threading an $S^z$ flux $\phi$ along the cylinder, we choose a branch cut where the $S^z$ rotation acts discontinuously and could not go through a site. Under inversion that exchanges two semions, one has to perform a spin rotation $e^{-iS^z\phi}$ for the sites encircled by the branch cut and its inversion image to restore the location of the branch cut. $\tilde{I}_s(\phi) = I_s \prod_{i\in encircled} e^{-iS_i^z\phi}$ combined with $e^{i\sum_i S_i^x \pi}$ hence can

be tracked throughout the flux threading. There are an odd number of sites encircled by the branch cut and its inversion image since one site sits at the center. When $\phi = 2\pi$ and we reach the $|s\rangle$ state, this additional rotation to restore the branch cut gives a $e^{iS^z 2\pi} = -1$ since there is a spin$-1/2$ moment at the inversion site. Hence

$$\frac{Q_s(\tilde{I}_s(2\pi)e^{i\sum_i S_i^x \pi})}{Q_0(\tilde{I}_s(0)e^{i\sum_i S_i^x \pi})} = 1 = -(I_s e^{i\pi S^x})_s^2. \qquad (4)$$

We have $I_s^2 = 1$ on triangular and square CSL, where $I_s = C_6^3$(triangular),or $I_s = C_4^2$(square).

Hence we prove that the symmetric CSL has a unique fractionalization pattern on Kagome, square and triangular lattices. Given the uniqueness of the symmetry fractionalization pattern, the $U(1)$ and $SU(2)$ ansatz for the CSL phase must correspond to the same phase on square, triangular and kagome lattice. This is quite useful to derive critical theories as now we can start from either the $U(1)$ or $SU(2)$ ansatz. Correspondingly the critical theory can be either $U(1)$ or $SU(2)$ gauge theory which must be dual to each other if we assume that there is only one universality class for the critical point.

## III. $SU(2)$ MEAN-FIELD ANSATZ, PROJECTIVE SYMMETRIES AND PROXIMATE PHASES

We introduce the $SU(2)$ slave rotor theory[11,49] to describe the chiral spin liquid phase and possible proximate phases. The CSL is realized in a Mott insulator with one electron per site. The $SU(2)$ gauge field may or may not be higgsed in the mean field ansatz. The electron operator is written as:

$$\begin{pmatrix} c_\uparrow(\mathbf{r}) \\ c_\downarrow^\dagger(\mathbf{r}) \end{pmatrix} = \begin{pmatrix} z_1^\dagger(\mathbf{r}) & z_2(\mathbf{r}) \\ -z_2^\dagger(\mathbf{r}) & z_1(\mathbf{r}) \end{pmatrix} \begin{pmatrix} f_\uparrow(\mathbf{r}) \\ f_\downarrow^\dagger(\mathbf{r}) \end{pmatrix} \equiv Z(\mathbf{r})\Psi(\mathbf{r}) \qquad (5)$$

where $Z(\mathbf{r}) \in SU(2)$ is a rotor field representing the charge degree of freedom and fermionic spinon $\Psi(\mathbf{r}) = \begin{pmatrix} f_\uparrow(\mathbf{r}) \\ f_\downarrow^\dagger(\mathbf{r}) \end{pmatrix}$ carries the spins. There is an $SU(2)$ gauge redundancy:

$$Z(\mathbf{r}) \to Z(\mathbf{r})U^\dagger(\mathbf{r}), \ \Psi(\mathbf{r}) \to U(\mathbf{r})\Psi(\mathbf{r}) \qquad (6)$$

where $U(\mathbf{r}) \in SU(2)$.

Simple algebra leads to

$$\begin{pmatrix} f_\uparrow(\mathbf{r}) \\ f_\downarrow^\dagger(\mathbf{r}) \end{pmatrix} \to U(\mathbf{r}) \begin{pmatrix} f_\uparrow(\mathbf{r}) \\ f_\downarrow^\dagger(\mathbf{r}) \end{pmatrix}, \ \begin{pmatrix} z_1(\mathbf{r}) \\ z_2^*(\mathbf{r}) \end{pmatrix} \to U(\mathbf{r}) \begin{pmatrix} z_1(\mathbf{r}) \\ z_2^*(\mathbf{r}) \end{pmatrix} \qquad (7)$$

The system has also a global $U(1)_c$ symmetry: $c_\sigma(\mathbf{r}) \to c_\sigma(\mathbf{r})e^{i\theta}$. This $U(1)$ global symmetry acts as

$$\begin{pmatrix} f_\uparrow(\mathbf{r}) \\ f_\downarrow^\dagger(\mathbf{r}) \end{pmatrix} \to \begin{pmatrix} f_\uparrow(\mathbf{r}) \\ f_\downarrow^\dagger(\mathbf{r}) \end{pmatrix}, \ \begin{pmatrix} z_1(\mathbf{r}) \\ z_2^*(\mathbf{r}) \end{pmatrix} \to e^{-i\theta} \begin{pmatrix} z_1(\mathbf{r}) \\ z_2^*(\mathbf{r}) \end{pmatrix} \qquad (8)$$

Similarly the global $SU(2)$ spin rotation symmetry transforms as

$$\begin{pmatrix} f_\uparrow(\mathbf{r}) \\ f_\downarrow(\mathbf{r}) \end{pmatrix} \to U_S \begin{pmatrix} f_\uparrow(\mathbf{r}) \\ f_\downarrow(\mathbf{r}) \end{pmatrix}, \quad \begin{pmatrix} z_1(\mathbf{r}) \\ z_2^*(\mathbf{r}) \end{pmatrix} \to \begin{pmatrix} z_1(\mathbf{r}) \\ z_2^*(\mathbf{r}) \end{pmatrix} \quad (9)$$

where $U_S \in SU(2)$.

With the holon $Z$ and the spinon $\Psi$, we can always write down a mean field theory with

$$H_M = H_{holon} + H_{spinon} \quad (10)$$

The mean field ansatz of the bosonic holon $Z$ and the fermionic spinon $\Psi$ is constrained by a projective symmetry group (PSG)[50] with an invariant Gauge group (IGG) which could be SU(2), $U(1)$ or Z$_2$. The bosonic holon could be either gapped or condensed. If $Z$ is gapped, this describes a spin liquid phase depending on the ansatz of the spinon $\Psi$. If $Z$ is condensed, we have a conventional phase which may have symmetry breaking if the PSG is non-trivial. An $SU(2)$ gauge transformation for both $\Psi$ and $Z$ corresponds to a gauge redundancy and does not change the physical state. However, if we do gauge transformation for only $Z$ or $\Psi$, we are changing the physical states. In this paper we will choose the gauge to fix the ansatz of $\Psi$, then different condensation patterns of $\langle Z \rangle$ lead to different phases. Alternatively, one may always fix the condensation of $Z$ to be $\langle z_1 \rangle \neq 0, \langle z_2 \rangle = 0$, then different phases arise from different ansatz of the fermionic spinon $\Psi$ related by gauge transformation acting only on $\Psi$. For our purpose, we take the first approach and always fix the gauge of $\Psi$.

When the bosonic holon $Z$ is gapped, we are in a Mott insulator phase with the spin degree of freedom decided by $H_\Psi$. In this section we focus on ansatz with an $SU(2)$ IGG. The fermion $\Psi_a$ is in a Chern insulator phase with $C = 1$, so the final theory is an SU(2)$_{-1}$ topological quantum field theory (TQFT) describing a chiral spin liquid (CSL)[51]. We will discuss mean field ansatz on square, triangular and kagome lattice. Note that $Z$ and $\Psi$ share the same gauge field and thus the same PSG. Once the mean field ansatz of $\Psi$ is fixed, we also know the PSG constraint on mean field ansatz of $Z$. We will show that low energy modes of $Z$ consist of two $SU(2)$ spinor $\Phi_1$ and $\Phi_2$, whose degeneracy is protected by the projective translation symmetry: $T_1T_2 = -T_2T_1$. From these two modes $\Phi_a, a = 1, 2$ we can construct five gauge invariant order parameters and derive their symmetry properties. These five order parameters are $\Phi^T \tau_2\sigma_2\Phi$ and $\Phi^\dagger \vec{\sigma}\Phi$. Here $\sigma_a$ acts in the space of $a = 1, 2$. For example, $\sigma_1$ transforms as: $\Phi_1 \leftrightarrow \Phi_2$. On the other hand, $\tau_a$ is the generator of the $SU(2)$ gauge transformation. For example, $\tau_1$ acts as: $\Phi_{a;1} \leftrightarrow \Phi_{a;2}$, where we write $\Phi_a = \begin{pmatrix} \Phi_{a;1} \\ \Phi_{a;2} \end{pmatrix}$. The modes $\Phi_a$ are the critical boson whose condensations drive the transition between the CSL and a nearby symmetry breaking phase.

Below we list the $SU(2)$ mean field ansatz for spinons and holons on square, triangular and Kagome lattices and the PSG of holons at lowest energy. The detailed solution and symmetry actions are contained in Appendix A.

### A. Square lattices

An $SU(2)$ ansatz for the CSL on square lattice is:

$$H_{spinon} = \sum_r i\Psi_{r+r_1}^\dagger \Psi_r - i(-1)^{r_1}\Psi_{r+r_2}^\dagger \Psi_r$$
$$+ i\eta(-1)^{r_1}\Psi_{r+r_1+r_2}^\dagger \Psi_r + i\eta(-1)^{r_1}\Psi_{r-r_1+r_2}^\dagger \Psi_r. \quad (11)$$

This simply describes spinons hopping on square lattice, with the hopping as

$$t_{r,r+r_1} = i, t_{r,r+r_2} = (-1)^{r_1}i,$$
$$t_{r,r+r_1+r_2} = (-1)^{r_1}i, t_{r,r+r_2-r_1} = (-1)^{r_1}i. \quad (12)$$

The holon Hamiltonian is given by the spinon mean-field values $\langle f_{r,s}^\dagger f_{r',s} \rangle$ from $H_{spinon}$. Assuming an electron hopping model $H_{KE} = \sum_{r,r',s} tc_{r,s}^\dagger c_{r',s}$ and from the parton relation Eq. 5, holon sector follow a Hamiltonian determined by the spinon mean-field value

$$H_{holon} = \sum_{r,r'} tz_{r,1}^*\langle f_{r',s}^\dagger f_{r,s}\rangle z_{r',1} - \sum_{r,r'} tz_{r,2}\langle f_{r',s}^\dagger f_{r,s}\rangle^* z_{r',2}^*. \quad (13)$$

Since the hopping amplitude for spinons $t_{ij} = \langle f_{s,i}^\dagger f_{s,j}\rangle$ from optimizing mean-field ansatz, we get the $SU(2)$ invariant Hamiltonian for holons

$$H_{holon} = \sum i z_{r+r_1}^\dagger z_r - i(-1)^{r_1} z_{r+r_2}^\dagger z_r$$
$$+ i\eta(-1)^{r_1} z_{r+r_1+r_2}^\dagger z_r + i\eta(-1)^{r_1} z_{r-r_1+r_2}^\dagger z_r, \quad (14)$$

where $z_r = (z_1(\mathbf{r}), z_2(\mathbf{r})^*)^T$.

There may also be other contributions to the holon mean field ansatz from the interaction instead of the hopping of the electrons. But these additional terms do not alter the PSG of the holons and do not influence our discussion in the following. We solve the Hamiltonian above and there are two degenerate lowest-energy states at momenta $\mathbf{Q}_{1,2} = (\pi/2, \pm\pi/2)$. (See Appendix A). We define, at low energy, two $SU(2)$ spinors for $Z = (z_1, z_2^*)^T$:

$$Z(r) = \Phi_1(r)e^{iQ_1 r} + \Phi_2(r)e^{iQ_2 r} \quad (15)$$

where $\Phi_{1,2}$ denotes the slowly-varying fields at momenta $Q_{1,2}$. Each spinor contains 2 components $(\Phi_{a;1}, \Phi_{a;2})$ for $(z_1, z_2^*)$. Under the $SU(2)$ gauge transformation, $\Phi_a(\mathbf{r}) \to U(r)\Phi_a(\mathbf{r})$, where $U(r) \in SU(2)$.

| | $T_1$ | $T_2$ | $C_4$ | $\tilde{C}_4$ | $R_1\mathcal{T}$ | $U(1)_c$ | note |
|---|---|---|---|---|---|---|---|
| $\Phi$ | $-i\sigma_1$ | $-i\sigma_2$ | $\sigma_2 e^{i\frac{\pi}{4}\sigma_3}$ | $-ie^{i\frac{\pi}{4}\sigma_3}$ | $i\sigma_1$ | $e^{i\frac{1}{2}\sigma_0\theta}$ | |
| $\Phi^\dagger\sigma_1\Phi$ | $+$ | $-$ | $-\Phi^\dagger\sigma_2\Phi$ | $\Phi^\dagger\sigma_2\Phi$ | $+$ | $+$ | $n_3$ |
| $\Phi^\dagger\sigma_2\Phi$ | $-$ | $+$ | $-\Phi^\dagger\sigma_1\Phi$ | $-\Phi^\dagger\sigma_1\Phi$ | $+$ | $+$ | $n_4$ |
| $\Phi^\dagger\sigma_3\Phi$ | $-$ | $-$ | $-\Phi^\dagger\sigma_3\Phi$ | $\Phi^\dagger\sigma_3\Phi$ | $-$ | $+$ | $n_5$ |
| $\Phi^T\sigma_2\tau_2\Phi$ | $+$ | $+$ | $-$ | $-$ | $+$ | $e^{i\theta}\Phi^T\sigma_2\tau_2\Phi$ | $n_1 - in_2$ |

TABLE I: The symmetry transforms of boson bilinears in the $SU(2)$ gauge theory on square lattices. $\tilde{C}_4 = T_2 C_4$ so $\tilde{C}_4$ acts as $-ie^{i\frac{\pi}{4}\sigma_3}$. In the following we define $A_r$ to be the gauge field of $\tilde{C}_4$. We see $(n_3, n_4)$ transforms as SO(2) rotation under $\tilde{C}_4$. $\Psi^\dagger\sigma_i\Psi$ represents 3 CDW order parameters at different momenta, while $\Psi^\dagger\sigma_2\tau^2\Psi^*$ carries one unit of $A_c$ charge and is identified as the cooper pair with $d$ wave symmetry.

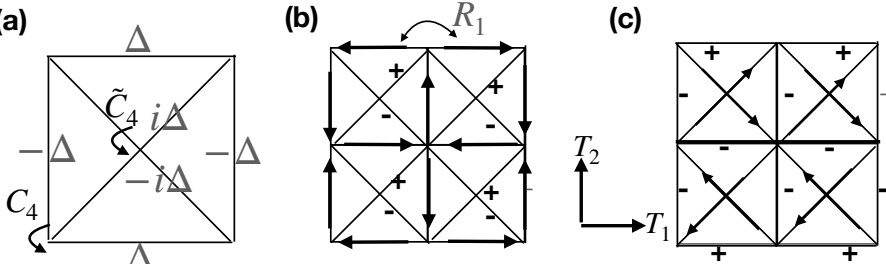

FIG. 2: (a)The translation-invariant electron $d + id$ pairing(spin singlet) amplitude when condensing holons on square lattices at low-energy fields $\Phi_{1;1} = 1, -\Phi_{2;2} = e^{i\Theta}, \Phi_{2;1} = \Phi_{1;2} = 0$, which makes $\Phi^T\tau_2\sigma_2\Phi = -2$. $\Delta = e^{i\Theta}$. (b,c) CDW patterns where $\pm$ indicates a positive, negative (real) expectation value of electron hopping across the bond.(b) CDW order on square lattice upon condensing $\Phi^\dagger\sigma_3\Phi$. The unit cell dimension along $T_{1,2}$ is both enlarged by 2. The black arrows denotes expectation value of $\langle c_i^\dagger c_j \rangle$ of the bond $\langle ij \rangle$ as $w = e^{i\pi/4}$ with direction from $i \to j$ indicated by the arrow. (c) CDW pattern upon condensing $\Phi^\dagger\sigma_1\Phi$. Unit cell is enlarged twice along $T_2$. Arrows indicates $\langle ij \rangle = i$ and direction is from $i \to j$. Another CDW pattern at momentum $(\pi, 0)$ by condensing $\Phi^\dagger\sigma_2\Phi$ is obtained by rotating (c) by $\pi/2$.

The relevant symmetries are translations $T_{1,2}$, fourfold rotations around a site $C_4$, time-reversal followed by reflection $R_{1,2}\mathcal{T}$. The spatial symmetries are listed in Fig 2. We also calculate rotation around a plaquette center defined as $\tilde{C}_4 = T_2 C_4$ in table I. Note that the sublattice structure on the square lattice gives 4 different rotations around $A, B$ sites/plaquettes centers, respectively. We only write down rotations around $B$ sites $C_4$ and the plaquette-center rotation related by $\tilde{C}_4 = T_2 C_4$.

The physical order parameters from electrons can be written from the parton construction, in particular the singlet pairing reads

$$\Delta_{rr'} = c_{r,\uparrow}c_{r',\downarrow} - c_{r,\downarrow}c_{r',\uparrow}$$
$$= \sum_s -z_{1,r}^* z_{2,r'}^* \langle f_{r,s} f_{r',s}^\dagger \rangle - z_{2,r}^* z_{1,r'}^* \langle f_{r,s} f_{r',s}^\dagger \rangle^*. \quad (16)$$

Using the $\langle f_{r,s} f_{r',s}^\dagger \rangle = t_{r,r'}$ for the mean field we use,

one could get the electron pairing when condensing the lowest-energy holons $z$. Using eq (15) one gets $\Delta_{rr'} = \Phi^T\tau_2\sigma_2\Phi$.

With $(\Phi_1, \Phi_2)$, we can define five gauge invariant order parameters: $(n_1, n_2, n_3, n_4, n_5) = (\text{Re}\Phi^T\sigma_2\tau_2\Phi, \text{Im}\Phi^T\sigma_2\tau_2\Phi, \Phi^\dagger\sigma_1\Phi, \Phi^\dagger\sigma_2\Phi, \Phi^\dagger\sigma_3\Phi)$. Their symmetry transformations are shown in Table I. Here we also include $U(1)_c$ symmetry which corresponds to the charge conservation. Our notation is that the charge of the Cooper pair is 1 and single electron carries charge $1/2$ for this $U(1)_c$ rotation. One can easily identify $n_1 + in_2$ as the Cooper pair with the same symmetry as a $d + id$ superconductor. Actually, if we condense $\Phi_1 = (1, 0)^T, \Phi_2 = (0, 1)^T$, the physical electron will be in the d+id superconductor ansatz. On the other hand, $n_3, n_4$ carry momentum $(\pi, 0)$ and $(0, \pi)$. They can be identified as the CDW$_{xy}$ order. $n_5$ carries momentum $(\pi, \pi)$ and is the CDW$_z$ order. The $d + id$ superconductor pattern is plot in Fig 2(a) and the CDW patterns are shown in Fig 2(b,c).

## B. Triangular lattices

We then move to triangular lattice. The spinon mean field reads

$$H_{spinon} = t_f \Psi^\dagger(\mathbf{r}+\widehat{r}_1)ie^{i\theta\tau_3}\Psi(\mathbf{r}) + h.c. + t_f\Psi^\dagger(\mathbf{r}+\widehat{r}_2)(-1)^{r_1}ie^{i\theta\tau_3}\Psi(\mathbf{r}) + h.c.$$
$$+ t_f\Psi^\dagger(\mathbf{r}+\widehat{r}_1+\widehat{r}_2)(-1)^{r_1+1}ie^{-i\theta\tau_3}\Psi(\mathbf{r}) + h.c. \tag{17}$$

where $\widehat{r}_{1,2}$ are primitive lattice vectors with an angle $120^o$ between them.

| | $T_1$ | $T_2$ | $C_6$ | $R\mathcal{T}$ | comment |
|---|---|---|---|---|---|
| $\Phi$ | $\Phi \to -i\sigma_1\Phi$ | $\Phi \to -i\sigma_3\Phi$ | $\Phi \to e^{i\frac{\pi}{3}}e^{-i\frac{\sigma_3+\sigma_2+\sigma_1}{\sqrt{3}}\frac{\pi}{3}}\Phi$ | $\Phi \to e^{-i\frac{\pi}{12}}e^{-i\sigma_1\frac{\pi}{4}}\Phi$ | |
| $\Phi^\dagger\sigma_1\Phi$ | $+$ | $-$ | $\Phi^\dagger\sigma_2\Phi$ | $+$ | $n_3$ |
| $\Phi^\dagger\sigma_2\Phi$ | $-$ | $-$ | $\Phi^\dagger\sigma_3\Phi$ | $\Phi^\dagger\sigma_3\Phi$ | $n_4$ |
| $\Phi^\dagger\sigma_3\Phi$ | $-$ | $+$ | $\Phi^\dagger\sigma_1\Phi$ | $\Phi^\dagger\sigma_2\Phi$ | $n_5$ |
| $\Phi^\dagger\sigma_2\tau^2\Phi^*$ | $+$ | $+$ | $e^{i\frac{2\pi}{3}}\Phi^\dagger\sigma_2\tau^2\Phi^*$ | $e^{i\frac{\pi}{6}}\Phi^\dagger\sigma_2\tau^2\Phi^*$ | $n_1+in_2$ |

TABLE II: The symmetry transforms of boson bilinears in the $SU(2)$ gauge theory on triangular lattices. $\Phi^\dagger\sigma_i\Phi$ represents 3 CDW order parameters at different momenta, while $\Phi^\dagger\sigma_2\tau^2\Phi^*$ carries one unit of $A_c$ charge and is identified as the cooper pair with $d$ wave symmetry.

For the boson $Z = (z_1, z_2^\dagger)$, we get the Hamiltonian:

$$H_{holon} = t_b Z^\dagger(\mathbf{r}+\widehat{r}_1)ie^{i\theta\tau_3}Z(\mathbf{r}) + h.c. + t_b Z^\dagger(\mathbf{r}+\widehat{r}_2)(-1)^{r_1}ie^{i\theta\tau_3}Z(\mathbf{r}) + h.c.$$
$$+ t_b Z^\dagger(\mathbf{r}+\widehat{r}_1+\widehat{r}_2)(-1)^{r_1+1}ie^{-i\theta\tau_3}Z(\mathbf{r}) + h.c. \tag{18}$$

When $\theta = 0$, the mean field is $SU(2)$ invariant.

Similar to the square case, there are 2 minima of holons represented by low energy spinors $\Phi_{1,2}$. The PSG for $\Phi_i$ are listed in table II.

The $d + id$ superconducting pattern and CDW pattern on triangular lattices upon condensing $\Phi^T\sigma_2\tau_2\Phi$, $\Phi^\dagger\sigma_I\Phi$, respectively are shown in fig 3.

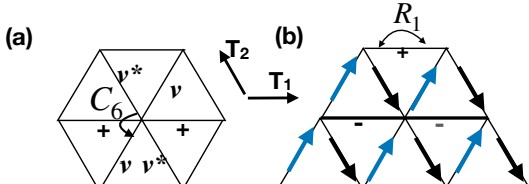

FIG. 3: (a)The translation-invariant electron $d + id$ pairing(spin singlet) amplitude when condensing holons on triangular lattices at low-energy fields $\Phi_{1;1} = -\Phi_{2;2} = 1, \Phi_{2;1} = \Phi_{1;2} = 0$, which makes $\Phi^T\tau_2\sigma_2\Phi = -1$. $v = e^{i2\pi/3}$. Hoppings between neighboring sites are identical. (b) CDW patterns where $\pm$ indicates a positive, negative (real) expectation value of electron hopping across the bond. Plotted is upon condensing $\Phi^\dagger\sigma_3\Phi$. The unit cell dimension along $T_1$ is both enlarged by 2. The black arrows denotes expectation value of $\langle c_i^\dagger c_j \rangle$ of the bond $\langle ij \rangle$ as $w = e^{i\pi/4}$ with direction from $i \to j$ indicated by the arrow, similarly blue arrows of value $iw = e^{i3\pi/4}$. Other 2 CDW patterns are obtained by rotating $2\pi/3, \pi/3$, respectively.

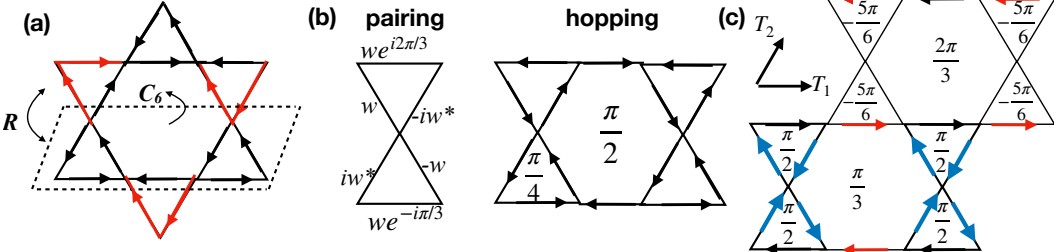

FIG. 4: (a)The $SU(2)$ invariant CSL ansatz on kagome lattices, with the arrows (irrespective of colors) indicating the direction of imaginary hopping. The unit cell is enlarged to $2 \times 1$. Black(red) bonds are additional positive(negative) real hopping that breaks $SU(2)$ to $U(1)$.(b)The electron pairing(spin singlet) and hopping amplitudes when condensing holons at low-energy fields $\Phi_{1;1} = \Phi_{2;2} = 1, \Phi_{2;1} = \Phi_{1;2} = 0$. The corresponding BCS Hamiltonian is translation invariant, though pairing symmetry is $p + ip$. $w = e^{i\pi/4}$. Arrows in the hopping pattern generally denote complex hopping we do not specify here for conciseness, and along the arrow direction is the hopping amplitude with an argument $\theta \in (-\pi/2, \pi/2)$. The hopping flux around hexagons/triangles are $\pi/2, \pi/4$ respectively. (c) CDW order on Kagome lattice where the flux around hexagones, triangles are marked.The unit cell is enlarged twice along $T_2$. The black arrows denotes expectation value of $\langle c_i^\dagger c_j \rangle$ of the bond $\langle ij \rangle$ as $v = e^{i2\pi/3}$ with direction from $i \to j$ indicatd by the arrow, similarly red arrows of value $-iv^* = e^{i5\pi/6}$, blue arrows of value $u = \sqrt{2}e^{i11\pi/12}$.

|  | $T_1$ | $T_2$ | $C_6$ | $R_1\mathcal{T}$ | note |
|---|---|---|---|---|---|
| transform | $i\sigma_2$ | $i\sigma_3$ | $e^{i\pi/6}e^{i\frac{\sigma_1+\sigma_2+\sigma_3}{\sqrt{3}}\pi/3}$ | $e^{i\frac{\pi}{4}}\frac{-\sigma_3+\sigma_1}{\sqrt{2}}$ |  |
| $\Phi^\dagger\sigma_1\Phi$ | $-$ | $-$ | $\Phi^\dagger\sigma_2\Phi$ | $-\Phi^\dagger\sigma_3\Phi$ | $n_3$ |
| $\Phi^\dagger\sigma_2\Phi$ | $+$ | $-$ | $\Phi^\dagger\sigma_3\Phi$ | $-$ | $n_4$ |
| $\Phi^\dagger\sigma_3\Phi$ | $-$ | $+$ | $\Phi^\dagger\sigma_1\Phi$ | $-\Phi^\dagger\sigma_1\Phi$ | $n_5$ |
| $\Phi^\dagger\sigma_2\tau^2\Phi^*$ | $+$ | $+$ | $e^{i\pi/3}$ | $-i\Phi^\dagger\sigma_2\tau^2\Phi^*$ | $n_1 + in_2$ |

TABLE III: The symmetry transforms of boson bilinears in the $SU(2)$ gauge theory on kagome altices. $\Phi^\dagger\sigma_i\Phi$ represents 3 CDW order parameters at different momenta, while $\Phi^\dagger\sigma_2\tau^2\Phi^*$ carries one unit of $A_c$ charge and is identified as the cooper pair.

### C. Kagome lattices

For Kagome lattices, the ansatz for CSL has an enlarged $2 \times 1$ unit cell shown in Fig 4(a). The hopping for spinons are purely imaginary and the chern number for the lowest 3 valence bands for one spinon are $-1, 1, 1$, respectively. Just retaining the NN hopping would also give a CSL, though to identify the simplest holon condensation pattern we add next nearest neighbor hopping to split the degeneracy of lowest energy states to 2-fold.

Again the holon bilinears correspond to symmetry-breaking order parameters and transform in table III.

The electron hopping and pairing descends from the holon expectation values and spinon ansatz from the relation eq (16) with the results shown in fig 4(b), where $w$ depends on NNN hopping, and the plot is for NNN hopping amplitude $\frac{i}{4}$. The CDW pattern is shown in fig 4(c).

### IV. $U(1)$ CSL ANSATZ

In the previous section we show the $SU(2)$ ansatz for the chiral spin liquid. Here we provide $U(1)$ ansatz for the same CSL phase. They are descendants of the $SU(2)$ ansatz by one additional Higgs term. Here we believe they actually belong to the same CSL phase as the $SU(2)$ ansatz: the topological order is the same and the lattice symmetry action on the semion should also be the same given the symmetry fractionalization is unique.

## A. Square lattices

On square lattice there turns out to be two different $U(1)$ ansatz for the CSL. The $U(1)$ ansatz can be obtained from adding $SU(2)$ ansatz. The low energy bosonic holon $\Phi = (\Phi_1, \Phi_2)^T$ will also feel the Higgs term. The two $U(1)$ ansatz corresponding to two different types of Higgs terms: (I)$\Phi^\dagger \sigma_3 \tau_3 \Phi$; (II) $\Phi^\dagger \tau_3 \Phi$.

### 1. Type I: $\Phi^\dagger \sigma_3 \tau_3 \Phi$

The first $U(1)$ ansatz is the stagger flux:

$$H_{spinon} = \sum_{r,s} f^\dagger_{r+\widehat{r}_1,s} e^{i\epsilon_r \theta} f_{r,s} + f^\dagger_{r+\widehat{r}_2,s} e^{-i\epsilon_r \theta} f_{r,s}$$
$$+ \eta f^\dagger_{r+\widehat{r}_1+\widehat{r}_2,s} \epsilon_r f_{r,s} - \eta f^\dagger_{r-\widehat{r}_1+\widehat{r}_2,s} \epsilon_r f_{r,s} + h.c. \quad (19)$$

where $s$ represents spin indices, $\widehat{r}_{1,2}$ are unit lattice vectors along two orthogonal directions, $\eta \in \mathcal{R}$ and $\epsilon_r$ is an alternating factor $(-1)^{r_1+r_2}$ for two sublattices of square lattice. The hopping flux around an elementary square alternates between $\pm 4\theta$ for neighboring plaquettes, hence staggered flux. This coupling form is invariant under a $U(1)$ rotation for the spinons $f_s \to e^{i\varphi} f_s$, which is a subgroup of the $SU(2)$ gauge group.

Assuming an electron hopping model $H_{KE} = \sum_{r,r',s} t c^\dagger_{r,s} c_{r',s}$ and from the parton relation eq (5) (identical to the derivation of eq (20)), holon sector follows a Hamiltonian determined by the spinon mean-field value

$$H_{holon} = \sum_{r,r'} t z^*_{r,1} \langle f^\dagger_{r',s} f_{r,s} \rangle z_{r',1} - \sum_{r,r'} t z_{r,2} \langle f^\dagger_{r',s} f_{r,s} \rangle^* z^*_{r',2}. \quad (20)$$

The spinon mean-field value $\langle f^\dagger_{r,s} f_{r+\vec{v}} \rangle$ is given by the corresponding hopping amplitude in eq (19).

Hence for staggered flux the holons are put into the ansatz:

$$H_{sqholon} = \sum_{r,s} Z^\dagger_{r+\widehat{r}_1} e^{i\epsilon_r \theta} Z_r + Z^\dagger_{r+\widehat{r}_2} e^{-i\epsilon_r \theta} Z_r$$
$$+ \eta Z^\dagger_{r+\widehat{r}_1+\widehat{r}_2} \epsilon_r Z_r - \eta Z^\dagger_{r-\widehat{r}_1+\widehat{r}_2} \epsilon_r Z_r + h.c. \quad (21)$$

where $Z_r = (z_1(r), z_2^*(r))^T$ and $\epsilon_r = \pm 1$ for sublattices $A, B$ in a checkerboard alignment, respectively.

At filling $x$ condensing holons in the 2 lowest energy states of eq (21), with equal amplitudes and a relative phase $\Theta$ gives

$$(\langle z_{1,r} \rangle, \langle z_{2,r} \rangle) = \sqrt{x}(1, \epsilon_r e^{i\Theta}). \quad (22)$$

The electrons have a finite overlap with the spinons by the relation $\psi_r = \frac{1}{\sqrt{2}} \langle Z_r \rangle \Psi_r$, and the Hamiltonian is re-lated to $H_{spinon}$ by the condensed holon values, i.e.

$$H_{electron} = \sum_{r,s} \epsilon_{ss'} [c^\dagger_{r,s} \sin\theta e^{i\Theta} c^\dagger_{r+\widehat{r}_1,s'} - c^\dagger_{r,s} \sin\theta e^{i\Theta} c^\dagger_{r+\widehat{r}_2,s'}$$
$$+ i\eta c^\dagger_{r,s} e^{i\Theta} c^\dagger_{r+\widehat{r}_1+\widehat{r}_2,s'} - i\eta c^\dagger_{r,s} e^{i\Theta} c^\dagger_{r+\widehat{r}_2-\widehat{r}_1,s'}]$$
$$+ c^\dagger_{r,s} \cos\theta c_{r+\widehat{r}_1,s} + c^\dagger_{r,s} \cos\theta c_{r+\widehat{r}_2,s} + h.c., \quad (23)$$

where $\epsilon_{ss'}$ is an antisymmetric symbol for spin indices and the state has a $d + id$ pairing structure.

*Relation to the $SU(2)$ ansatz:* The staggered flux is obtained from the $SU(2)$ invariant CSL ansatz by adding real hopping for spinons:

$$\Delta H_{csl} = \epsilon(\sum_r (-1)^{r_2} f^\dagger_{r+r_2} f_r + (-1)^{r_1+r_2} f^\dagger_{r+r_1} f_r). \quad (24)$$

One could check that adding such terms to the spinon ansatz in eq (11) would change the hopping flux around elementary square plaquette from $\pi$ to $\pm(\pi + \theta)$ in alternate plaquettes, where $\theta = 4\arctan(\epsilon)$. Hence the ansatz is gauge equivalent to the staggered flux eq (19).

For holons the additional terms would correspond to

$$\Delta H_{hcsl} = \epsilon(\sum_r (-1)^{r_2} z^\dagger_{r+r_2} \tau_3 z_r + (-1)^{r_1+r_2} z^\dagger_{r+r_1} \tau_3 z_r). \quad (25)$$

Projecting the terms to the lowest-energy holon states in eq (A2)(15), we find that it amounts to adding $\Phi^\dagger \sigma_2 \tau_3 \Phi$, which upon the redefinition by $\Phi \to e^{i\pi/4\sigma_1} \Phi$, this additional term is in the form $\Phi^\dagger \sigma_3 \tau_3 \Phi$. It naively breaks the translation and rotation symmetry in Table. I, but it is actually symmetric if one includes a gauge transformation $i\tau_1$ after the translation, $R_1 \mathcal{T}$ and $C_4$. The new symmetry action is:

$$T_1 : \Phi \to \tau_1 \sigma_1 \Phi, T_2 : \Phi \to \tau_1 \sigma_2 \Phi,$$
$$C_4 : \Phi \to -i\tau_1 \sigma_2 e^{i\frac{\pi}{4}\sigma_3} \Phi,$$
$$\tilde{C}_4 : \Phi \to -i e^{i\frac{\pi}{4}\sigma_3} \Phi, R_1 \mathcal{T} : \Phi \to \tau_1 \sigma_1 \Phi. \quad (26)$$

### 2. Type II: $\Phi^\dagger \tau_3 \Phi$

Another $U(1)$ ansatz on square lattice is by adding to the $SU(2)$ ansatz Eq. 11 longer range real spinon hopping:

$$t_{i,i+2\widehat{r}_1} = t_{i,i+2\widehat{r}_2} = \epsilon, \quad (27)$$

where $\epsilon \in \mathbb{R}$. This ansatz corresponds to adding $\Phi^\dagger \tau_3 \Phi$ to the $SU(2)$ ansatz for the holon low-energy fields $\Phi = (\Phi_1, \Phi_2)^T$.

## B. Triangular and Kagome lattices

We try to generalize the $U(1)$ ansatz to the triangular and kagome lattice by considering possible symmetric Higgs terms. Unlike the case for the square lattice,

$\Phi^\dagger \sigma_i \tau_j \Phi (i, j \neq 0)$ can not be symmetric unless $i = 0$. So the only $U(1)$ ansatz is the type II ansatz with the Higgs term as $\Phi^\dagger \tau_3 \Phi$. On triangular lattice this corresponds to adding:

$$t_{i,i+2\hat{r}_1} = t_{i,i+2\hat{r}_2} = -t_{i,i+2\hat{r}_1+2\hat{r}_2} = \epsilon, \qquad (28)$$

where $\epsilon \in \mathbb{R}$.

The $U(1)$ ansatz on kagome lattice can be obtained by adding real hopping with certans signs for spinons across neighboring bonds as the black/red bonds in Fig 4(a). In terms of the spinor $\Psi$ real hopping is proportional to $\Psi_i^\dagger \tau_3 \Psi_j$. Again for boson we can only add the Higgs term $\Phi^\dagger \tau_3 \Phi$.

The type I and type II $U(1)$ ansatz will have very different symmetry transformations. As we will show, superconductor can be obtained from simple holon condensation only from the type I $U(1)$ ansatz. On the other hand, from the type II $U(1)$ ansatz we can get the CDW$_{xy}$ order from holon condensation. From both type I and type II ansatz, the CDW$_z$ order can be obtained from condensing the holons with easy axis anisotropy. These different approaches lead to various critical theories. Especially the CSL-CDW$_z$ transition can be described starting from both type I and type II ansatz, which implies two critical theories in the same form for the same critical point. This QCP turns out to be self dual. For the CSL-SC or CSL-CDW$_{xy}$ transition, we can only start from either type I and type II ansatz. But we will find out these two critical points are described by the same $U(1)$ theory with two critical bosons. Basically. there is a duality transformation mapping the type I to type II $U(1)$ Higgs term, which induces duality mapping in the critical theories. This duality transformation will be discussed in details in Sec. IX.

## V. EQUIVALENCE BETWEEN CSL-SC TRANSITION AND LAUGHLIN STATE TO SUPERFLUID TRANSITION OF COOPER PAIR

We are going to discuss the transition between the chiral spin liquid (CSL) and the $d+id$ superconductor. The spin and single electron remain gapped in the bulk. Although the charge gap is closed at the QCP, the critical degree of freedom does not carry spin. So in the low energy we only need to consider the bosonic degree of freedom. Actually, here we show that the QCP can be viewed as a state with fractional quantum Hall effects (FQHE) to a superfluid (SF) transition of the charge $2e$ Cooper pair.

We will stack a $\nu = -2$ state with integer quantum Hall effects (IQHE) of spinful electrons to the CSL-SC transition. This IQHE state does not change across the transition and it does not have any effect in the bulk. Its role is to gap out the gapless edge mode carrying $S = \frac{1}{2}$ in the CSL-SC transition, so that we can get rid of the spin 1/2 electron in our final critical theory. We label $\tilde{A}_c$

and $\tilde{A}_s$ as probing gauge fields corresponding to charge $Q$ and spin $S_z$. Then the response of $\nu = -2$ IQHE is:

$$
\begin{aligned}
\mathcal{L}_{\text{IQHE}} &= \frac{1}{4\pi}\beta_1 d\beta_1 + \frac{1}{4\pi}\beta_2 d\beta_2 \\
&+ \frac{1}{2\pi}\tilde{A}_c d(\beta_1 + \beta_2) + \frac{1}{2\pi}\frac{1}{2}\tilde{A}_s d(\beta_1 - \beta_2) \quad (29)
\end{aligned}
$$

where $\beta_1, \beta_2$ are introduced to describe the $C = 1$ IQHE of each spin. Here we use $\beta_1, \beta_2$ to keep the information of the thermal Hall effect or chiral central charge. Throughout this paper we use $adb$ as an abbrivation of the Chern-Simons term $\epsilon^{\mu\nu\sigma} a_\mu \partial_\nu b_\sigma$, where $\epsilon^{\mu\nu\sigma}$ is the anti-symmetric tensor with $\epsilon^{012} = 1$ and $\epsilon^{\mu\nu\sigma} = -\epsilon^{\mu\sigma\nu}$.

We can integrate $\beta_1, \beta_2$ to get:

$$\mathcal{L}_{\text{IQHE}} = -\frac{2}{4\pi}\tilde{A}_c d\tilde{A}_c - \frac{1}{2}\frac{1}{4\pi}\tilde{A}_s d\tilde{A}_s - 4CS[g] \qquad (30)$$

where $CS[g]$ is the gravitational Chern-Simons term to encode the thermal Hall effect.This $CS[g]$ is given by some function of the Riemann tensor of a 4 dimension manifold. It is formally needed to produce a theory which leads to the same partition function as that from a $U(1)_1$ action, consistent with framing anomaly and gluing laws, etc. On physical grounds, a Dirac chiral fermion edge mode, associated with thermal hall effects, propagates along the edge on an open manifold from the $U(1)_1$ action. This makes the partition function diffeomorphism invariant and the $CS[g]$ holds the same issue, requiring a Dirac chiral edge mode to remedy. For a review see[52].

For the CSL part, its effective action is:

$$
\begin{aligned}
\mathcal{L}_{\text{CSL}} &= -\frac{1}{4\pi}\alpha_1 d\alpha_1 - \frac{1}{4\pi}\alpha_2 d\alpha_2 \\
&+ \frac{1}{2\pi}(\frac{1}{2}\tilde{A}_s + a)d\alpha_1 + \frac{1}{2\pi}(-\frac{1}{2}\tilde{A}_s + a)d\alpha_2 \quad (31)
\end{aligned}
$$

where $\alpha_1, \alpha_2$ are introduced to describe $C = 1$ Chern insulator phase for spin up and spin down spinon $f_\sigma$. We can also integrate $\alpha_1, \alpha_2$ to get

$$\mathcal{L}_{\text{CSL}} = \frac{2}{4\pi}ada + \frac{1}{2}\frac{1}{4\pi}\tilde{A}_s d\tilde{A}_s + 4CS[g] \qquad (32)$$

Note here $a$ is a spin gauge field meaning it couples to a fermion. Therefore, the anyon with $l = 1$ charge has statistics $\theta = -\frac{\pi}{2} + \pi = \frac{\pi}{2}$, which is a semion expected for a CSL phase.

Then the total action is:

$$\mathcal{L}_{\text{IQHE+CSL}} = \frac{2}{4\pi}ada - \frac{2}{4\pi}\tilde{A}_c d\tilde{A}_c \qquad (33)$$

One can see that the spin Hall effect and the thermal Hall effect of the CSL are cancelled by that of the IQHE phase. We can also do a redefinition: $\tilde{a} = a - A_c$, so that

$$\mathcal{L}_{\text{IQHE+CSL}} = \frac{2}{4\pi}\tilde{a}d\tilde{a} + \frac{2}{2\pi}\tilde{A}_c d\tilde{a} \qquad (34)$$

The charge of $\tilde{a}$ can now be either a fermion with $Q = 0, S = \frac{1}{2}$ or a boson with $Q = 1, S = 0$ because we can always combine a single electron with $Q = 1, S = \frac{1}{2}$. Here $Q$ is the charge under $\tilde{A}_c$ and $S$ is the physical spin. The former has statistics $\theta = \frac{\pi}{2}$, $Q = 0, S = \frac{1}{2}$ and is the usual semionic spinon. The latter has statistics $\theta = \frac{\pi}{2}$, $Q = 1, S = 0$ and can be identified as the anyon of a $\nu = -\frac{1}{2}$ Laughlin state of the Cooper pair. Actually the above Lagrangian is exactly the effective theory of the Laughlin state.

Because edge mode does not carry any $S = \frac{1}{2}$, we can view it as a phase of bosonic Cooper pair and ignore the single electron excitation. Then we define $A_c = 2\tilde{A}_c$ as the probing gauge field of the Cooper pair charge, then the effective theory is

$$\mathcal{L}_{\text{IQHE+CSL}} = \frac{2}{4\pi}ada + \frac{1}{2\pi}A_c da \qquad (35)$$

Similarly, for the $d + id$ superconductor, there is a spin Hall effect and $c = 2$ thermal Hall effect, which can also be cancelled by the $\nu = -2$ IQHE phase. Therefore, the effective theory for the d+id SC+IQHE is:

$$\mathcal{L}_{\text{IQHE+SC}} = \frac{1}{2\pi}A_c da - \frac{1}{2}\frac{1}{4\pi}A_c dA_c \qquad (36)$$

where $a$ is a gauge field representing the superfluid Goldstone mode. The first term represents a superfluid phase of the Cooper pair. The second term can be ignored for a superfluid phase. The final phase has chiral central charge $c = 0$ and is an ordinary superfluid phase.

In the above we show that the CSL to d+id SC transition can be viewed as a transition between $\nu = -\frac{1}{2}$ Laughlin state to superfluid transition of the Cooper pair up to stacking of a $\nu = -2$ IQHE phase. In the remaining of the paper, we will always stack the $\nu = -2$ IQHE phase at the critical points between CSL, SC and even CDW Chern insulator. For the CDW Chern insulator, the final phase is a trivial insulator after the stacking of the $\nu = -2$ IQHE phase. The critical theories can then be viewed as that of a pure bosonic model formed by spinless Cooper pair. As a result physical spin will be ignored in the discussions. We note that a critical theory with two Dirac fermions was proposed before for Laughlin state to superfluid transition[20] in the context of quantum Hall system, which coincides with one of our critical theories for the CSL-SC QCP (see Sec. VII). We complement with two other theories with two bosons coupled to either $U(1)$ or $SU(2)$ gauge fields. Note a $U(1)$ theory with four critical bosons was considered in Ref. 53.

Our analysis shows that there are three additional CDW orders at the CSL-SC, which is required by the LSM theorem. Without the crystal symmetry and LSM constraint, the transition is generically fine tuned and there will be an intermediate trivial insulator in between[20]. The crystal symmetries and LSM constraint required to protect the direct transition are not obviously there for the usual Laughlin state in quantum Hall system, but in our model they naturally exist.

## VI. $U(1)_{-2}$ WITH $2\varphi$ FOR THE CSL-SC TRANSITION ON SQUARE LATTICE

We first derive a critical theory starting from the $U(1)$ ansatz for the CSL on square lattice. As shown in Sec IV, on square lattice, there is a $U(1)$ mean field ansatz (the staggered flux ansatz) which is gauge equivalent to $d + id$ superconductor. Then we expect the transition from the CSL to the d+id SC is driven simply by holon condensation. But as we will show in this section, there are two bosonic fields $\varphi_1$ and $\varphi_2$ at the critical point. The degeneracy of these two bosons is protected by the projective translation symmetry $T_1T_2 = -T_2T_1$ in the PSG of the staggered flux ansatz. If we use the gauge in which the ansatz is written as a $d + id$ superconductor for $\Psi$, then naively we have trivial PSG $T_1T_2 = T_2T_1$. However, in this gauge the $U(1)$ gauge field can not be written in an explicit way. It is more convenient to work in the staggered flux ansatz where there is an explicit $U(1)$ gauge symmetry.

We consider the staggered flux ansatz discussed in section IV. It is equivalent to add a perturbation term to the $SU(2)$ ansatz:

$$H' = -\Phi^\dagger \tau_3 \sigma_3 \Phi \qquad (37)$$

where $\Phi = (\Phi_1, \Phi_2)$. $\Phi_a$ is an $SU(2)$ spinor. $\Phi_1$ and $\Phi_2$ are related by projective translation symmetry.

With this perturbation, the IGG becomes $U(1)$ and the CSL is described by a $U(1)_2$ theory. The relevant low energy holon fields are now $\varphi_1 = \Phi_{1;1}$ and $\varphi_2 = \Phi_{2,2}^*$. Note that naively $-\Phi^\dagger \tau_3 \sigma_3 \Phi$ term breaks some symmetries listed in Table. I, but the ansatz is actually symmetric if we include an $SU(2)$ gauge transformation $\pm i\tau_1$. We have new symmetry transformations for $\Phi = (\Phi_1, \Phi_2)$: $T_1 : \tau_1\sigma_1$, $T_2 : \tau_1\sigma_2$, $R_1\mathcal{T} : \tau_1\sigma_1$, $\tilde{C}_4 : -ie^{i\frac{\pi}{4}\sigma_3}$. The transformations in terms of $\varphi = (\varphi_1, \varphi_2)^T$ can be easily derived and are listed in Table. IV, where we use $\vec{\sigma}$ as Pauli matrices in the space of $(\varphi_1, \varphi_2)$. $U(1)_c$ is the physical $U(1)$ rotation with the convention that the charge is 1 for the Cooper pair. We also include an emergent $U(1)_r$ symmetry, which is the continuum version of the $\tilde{C}_4$ rotation.

| | $T_1$ | $T_2$ | $\tilde{C}_4$ | $R_1\mathcal{T}$ | C | $U(1)_c$ | $U(1)_r$ | comment |
|---|---|---|---|---|---|---|---|---|
| $\varphi = (\varphi_1, \varphi_2)^T$ | $\sigma_1\varphi^*$ | $-i\sigma_1\varphi^*$ | $-i\sigma_3 e^{i\frac{\pi}{4}\sigma_0}\varphi$ | $\sigma_1\varphi^*$ | $\varphi^*$ | $e^{i\frac{1}{2}\sigma_3\theta}\varphi$ | $e^{i\frac{1}{2}\sigma_0\theta}\varphi$ | |
| $\varphi^\dagger\sigma_1\varphi$ | + | + | − | + | + | $\varphi^\dagger(\cos\theta\sigma_1 + \sin\theta\sigma_2)\varphi$ | $\varphi^\dagger\sigma_1\varphi$ | $n_1$ |
| $\varphi^\dagger\sigma_2\varphi$ | + | + | − | − | − | $\varphi^\dagger(-\sin\theta\sigma_1 + \cos\theta\sigma_2)\varphi$ | $\varphi^\dagger\sigma_2\varphi$ | $n_2$ |
| Re $\mathcal{M}_a$ | + | − | Im $\mathcal{M}_a$ | + | + | Re $\mathcal{M}_a$ | $\cos\theta$Re $\mathcal{M}_a + \sin\theta$Im $\mathcal{M}_a$ | $n_3$ |
| Im $\mathcal{M}_a$ | − | + | −Re $\mathcal{M}_a$ | + | − | Im $\mathcal{M}_a$ | $-\sin\theta$Re $\mathcal{M}_a + \cos\theta$Im $\mathcal{M}_a$ | $n_4$ |
| $\varphi^\dagger\sigma_3\varphi$ | − | − | + | − | + | $\varphi^\dagger\sigma_3\varphi$ | $\varphi^\dagger\sigma_3\varphi$ | $n_5$ |

TABLE IV: Symmetry transformations in the $U(1)_2$ $2\varphi$ theory for the CSL-SC transition on square lattice. C is the charge conjugation which exists only for the bandwidth tuned transition. Only $R_1\mathcal{T}$ is anti-unitary. The monopole operator is defined as $\mathcal{M}_a = \mathcal{M}_a^0(\varphi^\dagger i\sigma_2\vec{\sigma}\varphi^*) \cdot (\varphi^\dagger\vec{\sigma}\varphi)$, where $\mathcal{M}_a^0$ is the bare monopole operator. This composite monopole operator is a singlet under the $SO(3)$ symmetry at the $\lambda = 0$ point generated by $\varphi \to e^{i\frac{1}{2}\vec{\sigma}\cdot\vec{n}}\varphi$. Here $\vec{n}$ is a unit vector.

From the symmetry transformation in Table. IV, it is clear that $\varphi_1^\dagger\varphi_2$ is the superconductivity order parameter. Its symmetry transformation matches that of the $d + id$ superconductor. When we have a condensation $\varphi = (1, 1)^T$, it is shown that the electron operator $c_\sigma$ acquires a $d + id$ superconductor order in section IV. When $\varphi$ is gapped, it is a CSL phase. Therefore a direct CSL to d+id SC transition can be described by the condensation of $\varphi$.

For simplicity, one will stack a $\nu = -2$ IQHE phase to cancel the spin Hall effect as done in Sec. V. The critical theory is:

$$\mathcal{L} = |\partial_\mu - ia_\mu - i\frac{1}{2}A_\mu^c\sigma_3 - i\frac{1}{2}A_\mu^r)\varphi|^2 - r|\varphi|^2 + \frac{2}{4\pi}ada$$
$$- g(|\varphi|^2)^2 + \lambda|\varphi_1|^2|\varphi_2|^2 - \frac{1}{8\pi}A^c dA^c \quad (38)$$

where $|\varphi|^2 = |\varphi_1|^2 + |\varphi_2|^2$ and Pauli matrix $\vec{\sigma}$ is acting in the $\varphi = (\varphi_1, \varphi_2)^T$ space. We use the Lorentz convention $\eta_{00} = 1, \eta_{11} = \eta_{22} = -1$, where $\eta_{\mu\nu}$ is the metric. Note that $r, g, \lambda$ need to be added a sign when compared to the Euclidean spacetime. Here $A_\mu^c$ is the probing gauge field for $U(1)_c$ symmetry. $A_\mu^r$ is a probing gauge field for the $U(1)_r$ symmetry, which is a continuum generalization of the $\tilde{C}_4$ rotation. As we discuss later, at the critical point the discrete $\tilde{C}_4$ rotation can be promoted to a continuous rotation and one has an emergent $U(1)_r$ symmetry whose transformation is listed in Table. IV. In the above $\frac{2}{4\pi}ada$ comes from the integration of the fermionic spinons. Spin Hall and thermal Hall effect are cancelled by the stacked IQHE phase. The above theory should describe a purely bosonic system with elementary physical charge 1 under $A_c$.

$r$ is the tuning parameter of this QCP. Easy-plane anisotropy $\lambda > 0$ is needed for the CSL-SC transition. $U(1)_c$ symmetry forbids $\varphi^\dagger\sigma_1\varphi$ and $\varphi^\dagger\sigma_2\varphi$. $T_1$ forbids $\varphi^\dagger\sigma_3\varphi$. Then the only gauge invariant bilinear term is $\varphi^\dagger\varphi$. One possible symmetric term is $-h\varphi^\dagger(i\sigma_3\partial_t + a_0\sigma_3 + \frac{1}{2}A_0^c + \frac{1}{2}A_0^r\sigma_3)\varphi$, One can check that this term is invariant under $T_1, T_2, \tilde{C}_4, R_1\mathcal{T}$. The couplings to $A_0^c$ and $A_0^r$ are enforced by the corresponding $U(1)$ transformations. Therefore the critical theory should contain this term and the dynamical exponent is $z = 2$,

unless fine tuning $h$ to be zero. If the probing field $A_\mu^c, A_\mu^r = 0$, then the critical point is at $r_c = 0$. However, if we add a constant $A_0^c = \delta\mu$, then $r_c$ is modified to be at $r_c = \frac{1}{2}h\delta\mu + \frac{1}{4}\delta\mu^2$, from which one can obtain $h = \frac{\partial r_c}{\partial\delta\mu}|_{\delta\mu=0}$. Physically $\delta\mu$ is obviously the change of the chemical potential. So one reaches the conclusion that the term $h = 0$ when $\frac{\partial r_c}{\partial\delta\mu}|_{\delta\mu=0} = 0$. When $h = 0$, the critical theory has a Charge conjugation symmetry:

$$C : \varphi \to \varphi^*, A_\mu^c \to -A_\mu^c, A_\mu^r \to -A_\mu^r, a_\mu \to -a_\mu. \quad (39)$$

The $h$ term will be mapped to $-h$ under $C$. So to leading order, one expects $h \sim -\delta\mu$ as $\delta\mu = A_0^c$ and $r_c \sim -\delta\mu^2$. This is also the relation shared by the superfluid-Mott insulator transition in boson Hubbard model. We will mainly focus on the point with $h = 0$ and a charge conjugation symmetry. Experimentally this fine tuned point can be easily accessed in the bandwidth controlled transition with electron density fixed at $n = 1$. On the other hand, the chemical potential tuned transition will have the $h$ term and a dynamical exponent $z = 2$.

One can also remove the coupling of $\varphi$ to $A_\mu^r$ by redefinition: $a_\mu \to a_\mu - \frac{1}{2}A_\mu^r$, then

$$\mathcal{L}_{u1-csl-sc} = |(\partial_\mu - ia_\mu - i\frac{1}{2}A_\mu^c\sigma_3)\varphi|^2 - r|\varphi|^2$$
$$- g(|\varphi|^2)^2 + \lambda|\varphi_1|^2|\varphi_2|^2$$
$$+ \frac{2}{4\pi}ada - \frac{1}{8\pi}A^c dA^c + \frac{1}{8\pi}A^r dA^r - \frac{1}{2\pi}A^r da \quad (40)$$

If $r > 0$, $\varphi$ is gapped and one is left with the following Lagrangian:

$$\mathcal{L} = \frac{2}{4\pi}ada - \frac{1}{2}\frac{1}{4\pi}A^c dA^c + \frac{1}{2}\frac{1}{4\pi}A^r dA^r - \frac{1}{2\pi}A^r da, \quad (41)$$

one can make a redefinition: $a_\mu \to a_\mu + \frac{1}{2}A_\mu^r + \frac{1}{2}A_\mu^c$, then get:

$$\mathcal{L} = \frac{2}{4\pi}ada + \frac{1}{2\pi}A^c da \quad (42)$$

which is just the effective theory for the $\nu = -\frac{1}{2}$ Laughlin state for the Cooper pair.

When $r < 0$, $\varphi$ needs to condense. For the fixed point with $\lambda > 0$, $\varphi \propto (1,1)^T$. This will higgs both $a_\mu$ and $A_\mu^c$, so $a_\mu$ is gapped and a superfluid phase for $A_\mu^c$ occurs. There is still a term in the superconductor phase:

$$\mathcal{L} = -\frac{1}{8\pi} A^c dA^c + \frac{1}{8\pi} A^r dA^r. \tag{43}$$

The term for $A^c$ can be ignored because of Meissner effects. There is a $1/2$ Hall effect for $A^r$. In our case the $U(1)_r$ symmetry is present only at the QCP, so $A^r$ is not well defined in the superconductor phase. Notwithstanding this term suggests that this is a topological superconductor.

One can identify $\varphi^\dagger \vec{\sigma} \varphi$ as $(n_1, n_2, n_5)$. We comment on the monopole operators in Table IV. In the $SU(2)$ theory there are five order parameters. The Higgs term should not alter this structure. In the $U(1)$ theory the remaining two order parameters come from the real and imaginary part of the monopole operator. We define the bare monopole operator as $\mathcal{M}_a^0$ which annihilates a $2\pi$ flux for the internal gauge field $a_\mu$. Because of the self Chern-Simons term $\frac{2}{4\pi} ada$, the bare monopole operator needs to be accompanied with operators such as $\varphi_a^* \varphi_b^*$ to be gauge charge neutral. There are various different operators which carry charge 2 under $a$. Here we choose the one which is singlet under the $SO(3)$ symmetry at $\lambda = 0$ generated by $\varphi \rightarrow e^{i\frac{1}{2}\vec{\sigma} \cdot \vec{n}} \varphi$ with $\vec{n}$ a unit vector. The monopole oprator we are looking for is $\mathcal{M}_a = \mathcal{M}_a^0 (\varphi^\dagger i\sigma_2 \vec{\sigma} \varphi^*) \cdot (\varphi^\dagger \vec{\sigma} \varphi)$.

This Monopole corresponds to the order parameter $n_3 + in_4$ because it carries charge 1 under $A_r$ and meanwhile is a $SO(3)$ singlet. On the other hand, the monopole operator such as $\mathcal{M}\varphi^\dagger i\sigma_2 \vec{\sigma} \varphi^*$ corresponds to composite order parameter $(n_3 + in_4)(n_1, n_2, n_5)$ because it is a tripet under the $SO(3)$ symmetry which rotates $(n_1, n_2, n_5)$ at $\lambda = 0$. When $\lambda \neq 0$, this $SO(3)$ symmetry is broken down to $SO(2)$ and the expression of the monopole may be deformed to have only $\mathcal{M}_a^0 (\varphi^\dagger i\sigma_2 \sigma_3 \varphi^*)(\varphi^\dagger \sigma_3 \varphi)$ component, but its symmetry transformation should remain the same as $\lambda = 0$. The symmetry actions of the monopole operator can be derived from the atomic limit of the holon states as we demonstrate in Appendix C. But the most convenient way is to start from the $SU(2)$ theory and view the theory in Eq. 40 as its Higgs descendant. Then the symmetry quantum numbers of the monopole operator should inherit the corresponding bilinear operators in the $SU(2)$ theory. We will further discuss this approach in Sec. X.

## VII.  $U(1)_1$ WITH $2\psi$ THEORY, BOSON-FERMION DUALITY AND EMERGENT SYMMETRY

We have shown a critical theory for the CSL-SC transition in the simple holon condensation picture if the parton mean field theory for the CSL is in the type I $U(1)$ ansatz. However, CSL can also have the type II $U(1)$ ansatz. Actually on triangular and kagome lattice, there is no type I $U(1)$ ansatz. For the type II $U(1)$ ansatz, holon condensation picture can only give a CDW phase because the ansatz can not be gauge transformed to a translationally invariant superconductor ansatz.

As argued before, the type I and type II $U(1)$ ansatz on square lattice should really describe the same CSL phase. So in principle there should still be a CSL-SC transition starting from the type II ansatz. There is no way to formulate it in the holon condensation picture. In this section we will provide an alternative path to the topological superconductor from the CSL phase.

### A.  $U(1)_1$ with $2\psi$ theory from plateau transition of holon

For this purpose, we can only use the $U(1)$ slave rotor theory with $c_{i,s} = b_i f_{i,s}$, where the gauge constraint is $n_f = n_b$ and $b, f_\sigma$ share the same $U(1)$ gauge field. In the $U(1)$ ansatz, both $b$ and $f_\sigma$ are in a $\pi$ flux ansatz: $\langle da \rangle = \pi$. The density of the slave boson $n_b$ is 1 per site and thus it is at filling $\nu = -2$ per magnetic unit cell. Here minus sign arises because $b$ carries opposite gauge charge compared to $f$. As before, the CSL phase corresponds to the trivial Mott insulator phase of the slave boson $b$. For the type II $U(1)$ ansatz, superfluid phase of $b$ leads to a CDW phase. However, boson at magnetic filling $\nu = -2$ can also be in a bosonic integer quantum Hall (bIQHE) phase. As shown in ref[46], bIQHE phase of the slave boson leads to a $d + id$ superconductor phase, schematically illustrated in Fig. 5. The simple understanding is that the $\frac{2}{4\pi} ada$ term of the CSL phase gets cancelled by the $\nu = -2$ bIQHE phase of $b$ and we are left with a term $\frac{1}{2\pi} A_c da$. Thus $a_\mu$ now represents the Goldstone mode of the SC. A detailed analysis shows that its topological property is equivalent to a $d + id$ superconductor[46].

In this picture, the phase transition is driven by the plateau transition of the slave boson, which is known to be described by a $N_f = 2$ QED[54,55]. The final critical theory is:

$$\mathcal{L} = \sum_{i=1,1} \bar{\psi}_i(-i\gamma_\mu \partial_\mu - b_\mu \gamma_\mu)\psi_i + m\bar{\psi}_i\psi_i - \frac{1}{4\pi}(\frac{1}{2}A_c - a)d(\frac{1}{2}A_c - a)$$
$$+ \frac{1}{2\pi} bd(\frac{1}{2}A_c - a) + \frac{2}{4\pi} ada - \frac{1}{8\pi} A_c dA_c, \tag{44}$$

where two Dirac fermions $\psi_1, \psi_2$ are introduced to describe the plateau transition of the holon, which couples to $\frac{1}{2}A_c - a$. $b_\mu$ is another internal gauge field. $m < 0$ and $m > 0$ corresponds to the trivial Mott insulator and bIQHE phase of the holon respectively. $\frac{2}{4\pi} ada$ comes from integration of the fermionic spinons. $-\frac{1}{8\pi} A_c dA_c$ comes from the stacking of the $\nu = -2$ IQHE phase discussed in Sec. V. We use the convention $\gamma_0 = \eta_3, \gamma_1 = i\eta_2, \gamma_2 = i\eta_1$ and $\bar{\psi} = \psi^\dagger \gamma_0$. $\eta_a$ is Pauli matrix acting on the spinor basis of a Dirac fermion.

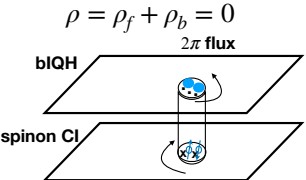

FIG. 5: The schematic illustration of the formation of superconducting phases from CSL when the holons form a bosonic integer quantum hall insulator with filling $\nu = -2$ with opposite chirality to that of the spinon. The spinon remains in a chern insulator with $\nu = 2$. A $2\pi$ flux of the internal gauge field nucleates a pair of holons and spinons, hence forming a spin singlet Cooper pair. From Ioffe-Larkin rule[11], the resistivity of holon and spinons are opposite and the physical resistivity tensor $\rho_c$ is zero, suggesting a superconducting phase.

Integrate $a_\mu$ and then one obtains

$$
\begin{aligned}
\mathcal{L}_{QED} = \sum_{i=1,2} &\bar{\psi}_i(-i\gamma_\mu\partial_\mu - b_\mu\gamma_\mu)\psi_i + m\bar{\psi}_i\psi_i \\
&- \frac{1}{4\pi}bdb + \frac{1}{2\pi}A_c db - \frac{1}{4\pi}A_c dA_c - 2CS[g],
\end{aligned}
\tag{45}
$$

where $-2CS[g]$ comes from $\frac{1}{4\pi}ada$. The information of a central charge $c = -1$ is lost after integrating $a_\mu$, so we need to add a gravitational Chern-Simons term to keep track of the thermal Hall effect.

When $m < 0$, integration of $\psi$ gives a $-\frac{1}{4\pi}bdb$ term

and a thermal Hall effect $-2CS[g]$. Finally we have:

$$
\mathcal{L} = -\frac{2}{4\pi}bdb + \frac{1}{2\pi}A_c db - \frac{1}{4\pi}A_c dA_c - 4CS[g]. \tag{46}
$$

One can check it is equivalent to the CSL phase with a stack of $\nu = -2$ IQHE phase. Note that the charge of $b_\mu$ is a fermion, so the anyon here carries statistics $\theta = \frac{\pi}{2} - \pi = -\frac{\pi}{2}$ and charge $Q = \frac{1}{2}$ under $A_c$. One combines a single electron (with charge $1/2$ under $A_c$ and physical spin $S = 1/2$) to get a neutral semion with spin $1/2$ as expected for the CSL phase.

When $m > 0$, integration of $\psi$ gives a $\frac{1}{4\pi}bdb$ term and a thermal Hall effect $2CS[g]$. Finally we have:

$$
\mathcal{L} = \frac{1}{2\pi}A_c db - \frac{1}{4\pi}A_c dA_c. \tag{47}
$$

This is the superconductor phase where $b$ higgs $A_c$. Note that the $-\frac{1}{4\pi}A_c dA_c$ term can be absorbed by $b \to b - \frac{1}{2}A_c$. This means that charge Hall effect is ill defined in a superfluid phase.

### B. Boson fermion duality

In the above we show that the CSL to SC transition is captured by a $U(1)_1$ theory with two Dirac fermions. The previous section provides a different critical theory with two complex bosons for the same transition. Assuming that there is only one universality class for this QCP, the two critical theories in Eq. 40 and in Eq. 45 must be dual to each other. If one ignores the Chern-Simons term in the two theories, the two theories are known to be dual to each other[2] and they describe the Neel to VBS deconfined quantum critical point(DQCP)[1]. Here we demonstrate the duality in the modified version with Chern-Simons terms for the CSL-SC transition.

Start from the $U(1)_{-2}$ theory with two complex bosons:

$$
\mathcal{L}_\varphi = |\partial_\mu - ia_\mu - i\frac{1}{2}A_{c;\mu}\sigma_3)\varphi|^2 - r|\varphi|^2 - g(|\varphi|^2)^2 + \lambda|\varphi_1|^2|\varphi_2|^2 + \frac{2}{4\pi}ada - \frac{1}{8\pi}A_c dA_c + \frac{1}{8\pi}A_r dA_r - \frac{1}{2\pi}A_r da, \tag{48}
$$

where $\varphi = (\varphi_1, \varphi_2)^T$ and $|\varphi|^2$ is an abbreviation of $\varphi^\dagger\varphi = |\varphi_1|^2 + |\varphi_2|^2$ .

Then apply the standard boson-fermion duality for one single boson or fermion[52,56]:

$$
|(\partial_\mu - ia_\mu)\phi|^2 - r|\phi|^2 - g|\phi|^4 \leftrightarrow \bar{\psi}(-i\gamma_\mu\partial_\mu - b_\mu\gamma_\mu)\psi \pm \frac{1}{8\pi}bdb \pm m\bar{\psi}\psi \pm \frac{1}{2\pi}bda \pm \frac{1}{4\pi}ada, \tag{49}
$$

which sends a theory describing boson condensation with internal $U(1)$ gauge field $a_\mu$ to a dual one with one Dirac fermion coupling to $U(1)$ gauge field $b_\mu$. The mass $r > 0$ ($r < 0$) corresponds to $m > 0$ ($m < 0$). There are two versions of the duality, corresponding to two sign choices (taking uniformly the upper or lower signs). For our purpose we will take the lower sign convention.

Applying the duality to $\varphi_1$ and $\varphi_2$, one gets

$$\mathcal{L}_\varphi \leftrightarrow \sum_{i=1,2} \bar{\psi}_i(-i\gamma_\mu\partial_\mu - b_{i;\mu}\gamma_\mu)\psi_i - m\bar{\psi}_i\psi_i - \sum_{i=1,2}\frac{1}{8\pi}b_idb_i - \frac{1}{2\pi}b_1d(a+\frac{1}{2}A_c) - \frac{1}{2\pi}b_2d(a-\frac{1}{2}A_c))$$

$$-\frac{1}{4\pi}(a+\frac{1}{2}A_c)d(a+\frac{1}{2}A_c) - \frac{1}{4\pi}(a-\frac{1}{2}A_c)d(a-\frac{1}{2}A_c) + \frac{2}{4\pi}ada - \frac{1}{8\pi}A_cdA_c + \frac{1}{8\pi}A_rdA_r - \frac{1}{2\pi}A_rda - 2CS[g]$$

$$= \sum_{i=1,2} \bar{\psi}_i(-i\gamma_\mu\partial_\mu - b_{i;\mu}\gamma_\mu)\psi_i - m\bar{\psi}_i\psi_i - \sum_{i=1,2}\frac{1}{8\pi}b_idb_i$$

$$-\frac{1}{2\pi}(b_1+b_2+A_r)da + \frac{1}{4\pi}A_cd(b_1-b_2) - \frac{1}{4\pi}A_cdA_c + \frac{1}{8\pi}A_rdA_r - 2CS[g], \tag{50}$$

where $\psi_i, b_i$ are introduced as dual theory of $\varphi_i$ for $i = 1, 2$. $-2CS[g]$ is introduced to match the thermal Hall effect of the left side, which is lost because $\frac{2}{4\pi}ada$ term is cancelled.

Integration of $a_\mu$ leads to $b_{1;\mu} + b_{2;\mu} = -A_{r;\mu}$. We will substitute $b_{1;\mu} = -\frac{1}{2}A_{r;\mu} + b_\mu, b_{2;\mu} = -\frac{1}{2}A_{r;\mu} - b_\mu$ and get

$$\mathcal{L}_\varphi \leftrightarrow \mathcal{L}_\psi = \bar{\psi}\gamma_\mu(-i\partial_\mu - b_\mu\sigma_3 + \frac{1}{2}A_{r;\mu})\psi - m\bar{\psi}\psi - \frac{1}{4\pi}bdb + \frac{1}{2\pi}A_cdb - \frac{1}{4\pi}A_cdA_c + \frac{1}{16\pi}A_rdA_r - 2CS[g], \tag{51}$$

where $\psi = (\psi_1, \psi_2)^T$. $\vec{\sigma}$ are Pauli matrices acting on the $(\psi_1, \psi_2)$ space. $\bar{\psi}\psi = \bar{\psi}\sigma_0\psi$.

We can make a charge conjugation transformation only for $\psi_2$: $\psi_2^c = C(\bar{\psi}_2)^T$ with $C = \gamma_1$. Using $\gamma_0 = \eta_3, \gamma_1 = i\eta_2, \gamma_2 = i\eta_1$ and $\bar{\psi} = \psi^\dagger\gamma_0$, it can be shown that $\bar{\psi}_2^c = -\psi_2^TC^{-1}$ and $C^{-1}\gamma_\mu C = -\gamma_\mu^T$. Then $m\bar{\psi}_2\psi_2 = -m\psi_2^T(\bar{\psi}_2)^T = m\bar{\psi}_2^c\psi_2^c$, $\bar{\psi}_2\gamma_\mu\psi_2 = -\psi_2^T\gamma_\mu^T(\bar{\psi}_2)^T = -\bar{\psi}_2^c\gamma_\mu\psi_2^c$ and $\bar{\psi}_2(-i\gamma_\mu\partial_\mu)\psi_2 = \psi_2^T(-i\gamma_\mu^T\partial_\mu)(\bar{\psi}_2)^T = \bar{\psi}_2^c(-i\gamma_\mu\partial_\mu)\psi_2^c$. We will replace $\psi_2$ with $\psi_2^c$, which only needs a flip of the signs for the couplings to the gauge fields. In the end one labels $\psi_2^c$ with $\psi_2$ for simplicity. Finally we obtain:

$$\mathcal{L}_\psi = \sum_{i=1,2} \bar{\psi}_i(-i\gamma_\mu\partial_\mu - b_\mu\gamma_\mu + \frac{1}{2}A_{r;\mu}\sigma_3\gamma_\mu)\psi_i + m\bar{\psi}_i\psi_i - \frac{1}{4\pi}bdb + \frac{1}{2\pi}A_cdb - \frac{1}{4\pi}A_cdA_c + \frac{1}{16\pi}A_rdA_r - 2CS[g], \tag{52}$$

If we ignore $A_r$, this is exactly the same as in Eq. 45. Now from the duality one can also derive the coupling to the probing field $A_r$, which is absent in Eq. 45. The above duality is derived by ignoring the interaction term $(2g - \lambda)|\varphi_1|^2|\varphi_2|^2$. The $r < 0$ ($r > 0$) side of Eq. 40 and $m < 0$ ($m > 0$) side of Eq. 52 describe the same phase. Integration of $\psi$ gives a term $-\frac{sgn(m)}{8\pi}((b+\frac{1}{2}A_r)d(b+\frac{1}{2}A_r) + (b-\frac{1}{2}A_r)d(b-\frac{1}{2}A_r)) - sgn(m)2CS[g] = -\frac{sgn(m)}{4\pi}bdb - \frac{sgn(m)}{16\pi}A_rdA_r - sgn(m)2CS[g]$. Thus for $m < 0$, the final theory is $\mathcal{L}_{m<0} = \frac{1}{2\pi}A_cdb - \frac{1}{4\pi}A_cdA_c + \frac{1}{8\pi}A_rdA_r$. This is a superfluid phase with $\frac{1}{8\pi}A_rdA_r$ response, the same as the $r < 0$ side of Eq. 40. When $m > 0$, we have $\mathcal{L}_{m>0} = -\frac{2}{4\pi}bdb + \frac{1}{2\pi}A_cdb - \frac{1}{4\pi}A_cdA_c - 4CS[g]$. This is

a phase with response $-\frac{1}{8\pi}A_cdA_c - 2CS[g]$, the same as the Laughlin state at $r > 0$ side of Eq. 40. The anyon has charge $Q = \frac{1}{2}$ and statistics $\theta = -\frac{\pi}{2}$[57], also consistent with the $\nu = -\frac{1}{2}$ Laughlin state of Cooper pair. Given that the two sides of the two critical theories are exactly the same, it is quite natural to expect that the two critical theories at $m = 0$ ($r = 0$) are also dual to each other. The derivation above further supports this duality. Similar theories with Dirac fermions describing CSL to XY-ordered or VBS transitions on square lattices are discussed in ref [58], where the boson-fermion duality was also formally derived. A different derivation of the duality appears in ref[59].

### C. Order parameters in the dual theory

| | $n_1$ | $n_2$ | $n_3$ | $n_4$ | $n_5$ |
|---|---|---|---|---|---|
| $U(1)_{-2}\ 2\varphi$ | $\varphi^\dagger\sigma_1\varphi$ | $\varphi^\dagger\sigma_2\varphi$ | $\mathrm{Re}\mathcal{M}_a^0(\varphi^\dagger i\sigma_2\vec{\sigma}\varphi^*)\cdot(\varphi^\dagger\vec{\sigma}\varphi)$ | $\mathrm{Im}\mathcal{M}_a^0(\varphi^\dagger i\sigma_2\vec{\sigma}\varphi^*)\cdot(\varphi^\dagger\vec{\sigma}\varphi)$ | $\varphi^\dagger\sigma_3\varphi$ |
| $U(1)_1\ 2\psi$ | $\mathrm{Re}\mathcal{M}_b^0\psi_1\psi_2$ | $\mathrm{Im}\mathcal{M}_b^0\psi_1\psi_2$ | $\bar{\psi}\sigma_1\psi$ | $\bar{\psi}\sigma_2\psi$ | $\bar{\psi}\sigma_3\psi$ |
| $SU(2)_{-1}\ 2\Phi$ | $\mathrm{Re}\Phi^T\sigma_2\tau_2\Phi$ | $\mathrm{Im}\Phi^T\sigma_2\tau_2\Phi$ | $\Phi^\dagger\sigma_1\Phi$ | $\Phi^\dagger\sigma_2\Phi$ | $\Phi^\dagger\sigma_3\Phi$ |

TABLE V: Five order parameters in the two dual theories for the CSL-SC transition. Symmetry transformations of these order parameters can be found in Table. IV. We list the corresponding operators in the $SU(2)$ theory which is going to be discussed in Sec. IX. $\mathcal{M}_a^0$ is the bare monopole operator in the $U(1)$ $2\varphi$ theory. $\mathcal{M}_b^0$ is the bare monopole operator in the $U(1)$ $2\psi$ theory.

We have shown that there are five order parameters at the CSL-SC QCP in Table. IV, which is derived in the $U(1)_{-2}$ theory with $2\varphi$. Now with a dual theory in terms

of Dirac fermions, these order parameters should exist also in the dual side. First, when deriving the duality, we start from a boson theory with the same mass $r$ for $\varphi_1$ and $\varphi_2$. Had one introduced $r_1$ and $r_2$ for $\varphi_1$ and $\varphi_2$ separately, the same procedure would have given mass term $m_1$ and $m_2$ for $\psi_1$ and $\psi_2$ in the dual side. This means that $\varphi^\dagger \sigma_3 \varphi$ is dual to $\bar\psi \sigma_3 \psi$. This is the order parameter $n_5$, representing the CDW order with momentum $(\pi, \pi)$. In the Dirac theory, it is easy to see that $\bar\psi_1 \psi_2$ carries charge $-1$ under $A_r$. Therefore one can identify it as the order parameter $n_3 + in_4$. In other words, $\bar\psi \sigma_1 \psi$ and $\bar\psi \sigma_2 \psi$ correspond to the two CDW orders $n_3$ and $n_4$. In the Dirac theory, the term $\frac{1}{2\pi} A_c db$ indicates that the monopole $\mathcal{M}_b^{0\dagger}$ carries charge 1 under $A_c$. Usually one needs to add a fermion zero mode to the monopole to make it gauge neutral. In our case, the term $-\frac{1}{4\pi} b db$ shows that the bare Monopole carries gauge charge $-1$, so one needs to attach two fermion zero modes to the monopole. Hence the gauge invariant monopole operator is $\mathcal{M}_b^{0\dagger} \psi_1^\dagger \psi_2^\dagger$. $\mathcal{M}_b^0 \psi_1 \psi_2$ is then the physical Cooper pair, whose real and imaginary parts are $n_1$ and $n_2$. We list the operator mappings in the two theories in Table. V.

The duality provides information on symmetry transformations of the operators in the Dirac theory, which is otherwise not obvious given that the microscopic content of the Dirac fermions $\psi_1, \psi_2$ are not clear. We can provide an ansatz for the Dirac fermions to satisfy the symmetry constraints, by taking a $\pi$ flux state on square lattice for spinless fermions to realize the two Dirac fermions at low energy. The lattice site now is at the plaquette center of the original lattice. Taking the mean-field $t_{i,i+\hat{x}} = (-1)^y, t_{i,i+\hat{y}} = 1$, the low-energy Lagrangian reads,

$$\mathcal{L}_{\pi-flux} = \sum_{i=1,2,\mu=0\cdots 2} \bar\psi_i (i\gamma^\mu \partial_\mu) \psi_i \qquad (53)$$

, where $\gamma_0 = \eta_3, \gamma_{1,2} = i\eta_{2,1}$ as $\eta$ are Pauli matrices acting on Lorentz indices. The symmetry action for the Dirac fermions reads under appropriate basis choice, (in $\Psi = (\psi_1, \psi_2)^T$

$$T_1 : \Psi \to i\sigma_2 \Psi, T_2 : \Psi \to i\sigma_1 \Psi$$
$$C_4 : \Psi \to e^{i\sigma_3 \frac{\pi}{4}} \sigma_1 e^{i\gamma_0 \frac{\pi}{4}} \Psi,$$
$$\tilde{C}_4 : \Psi \to e^{i\sigma_3 \frac{\pi}{4}} \sigma_3 e^{i\gamma_0 \frac{\pi}{4}} \Psi,$$
$$C : \Psi \to \gamma_1 \sigma_3 (\bar\Psi)^T, R_1 \mathcal{T} : \Psi \to -i\sigma_1 \Psi. \qquad (54)$$

One can verify that $\bar\psi \sigma_i \psi$ transform in the same as the three CDW orders in the $U(1)$ $2\varphi$ theory, as listed in Table. VI.

As for the transformation of $\mathcal{M}_b^0 \psi_1 \psi_2$, it is hard to derive its quantum numbers due to a lack of definite UV realization of the QED theory $\mathcal{L}_\psi$ (note there is a Chern-simons coupling) and the existence of associated atomic limits. So we rely on duality and match the monopole symmetries with those of $\varphi_1^* \varphi_2$ in the $U(1)$ $2\varphi$ theory.

The $U(1)$ $2\psi$ theory for the CSL-SC transition is derived from the plateau transition of the bosonic holons

| Op. mapping | $T_1$ | $T_2$ | $\tilde{C}_4$ | $R_1 \mathcal{T}$ | $C$ |
|---|---|---|---|---|---|
| $\bar\psi \sigma_3 \psi$ | $-$ | $-$ | $+$ | $-$ | $+$ |
| $\bar\psi \sigma_1 \psi$ | $-$ | $+$ | $-\bar\psi \sigma_2 \psi$ | $+$ | $-$ |
| $\bar\psi \sigma_2 \psi$ | $+$ | $-$ | $\bar\psi \sigma_1 \psi$ | $+$ | $+$ |

TABLE VI: The symmetry actions of Dirac fermions realized from square lattice $\pi$ flux state.

and it should also apply to the triangular and kagome lattice. Later an $SU(2)$ theory for the CSL-SC transition on triangular and kagome lattice is provided. Therefore this $U(1)$ $2\psi$ theory should be dual to the $SU(2)$ theory. The symmetry transformations of the five operators in the $U(1)$ $2\psi$ theory should be the same as in the $SU(2)$ theory. Especially $\bar\psi \vec\sigma \psi$ still represent three CDW orders. We will not try to regularize the Dirac fermions with a lattice model and instead rely on the duality to the $SU(2)$ theory to obtain the symmetry transformations in this low energy theory.

### D. Emergent symmetry and anomaly

Here we show that the CSL-SC QCP has an emergent symmetry $SO(3) \times O(2)$. First, the SC order parameter $(n_1, n_2)$ has a $U(1)$ global symmetry generated by $A_c$. Similarly, easy-plane CDW order $(n_3, n_4)$ has a $U(1)$ global symmetry generated by $A_r$. Note that $U(1)_r$ is already emergent, as there is only $C_4$ rotation in the lattice scale. In the Dirac fermion theory, it is believed the quartic terms are irrelevant. Then one can see that Eq. 52 has an $SO(3)$ symmetry generated by the three Pauli matrices $\sigma_1, \sigma_2, \sigma_3$, which rotate in the subspace $(n_3, n_4, n_5)$. This enlargement of $U(1)_r$ to $SO(3)$ is not transparent in the boson theory, but given the duality, one can now easily see it in the Dirac theory. In addition to $SO(3)$ and $U(1)_c$, there is also a $Z_2$ symmetry from the charge conjugation $C$, which corresponds to an improper rotation in $(n_1, n_2)$ and $(n_3, n_4, n_5)$ space. Together, we have $SO(3) \times O(2)$ symmetry. Note that there is an additional $R_1 \mathcal{T}$ symmetry which is an anti-unitary transformation and maps $n_5$ to $-n_5$.

One comment on mixed anomaly of the $U(1)_c \times U(1)_r \rtimes T_1$ or $U(1)_r \times U(1)_c \rtimes T_2$ symmetry is in order. At the QCP the lattice symmetries act like an internal symmetry. For example, $T_1$ and $T_2$ act like a $Z_2$ symmetry in Table. IV. In Eq. 52, $dA_c = 2\pi$ carries gauge charge 1 under $b_\mu$. In order to cancel the gauge charge, one needs to combine a Dirac fermion $\psi_1$ or $\psi_2$, which carries charge $\pm \frac{1}{2}$ under $A_r$. Similarly, from the boson theory in Eq. 40, one can see that $dA_r = 2\pi$ needs to combine a boson which carries charge $\pm \frac{1}{2}$ under $A_c$. This anomaly indicates that there is no trivially gapped phase without symmetry breaking proximate to the QCP. For example, from the superfluid phase of $A_c$, to reach a trivially gapped phase one needs to condense its vortex. However, the above analysis shows that the vortex carries charge

$\pm 1/2$ under $A_r$ and its condensation leads to a superfluid of $A_r$, which correspond to the $(n_3, n_4)$ orders. A $Z_2$ symmetry from $T_1$ or $T_2$ is crucial. For example, in Eq. 40, one can have $\langle \varphi_1 \rangle \neq 0, \langle \varphi_2 \rangle = 0$, which locks $a_\mu = -\frac{1}{2} A_{c;\mu}$. Then we have $\mathcal{L} = -\frac{1}{4\pi} A_c dA_r - \frac{1}{8\pi} A_r dA_r$. In this case both $U(1)_c$ and $U(1)_r$ are preserved. However, $T_1$ is broken because $T_1$ flips the charge of $A_r$. This phase has the order $n_5$. Similar anomaly also exists in the familiar example of Neel to VBS DQCP[1,2,16].

The $U(1)_1$ $2\psi$ theory in Eq. 52 has already been proposed in a previous paper on the transition between a Laughlin state and a superfluid phase of boson[20]. Generically there is a relevant term $\bar{\psi}\sigma_z\psi$ which drives the system to an insulator in the middle between the Laughlin state and the superfluid phase. In order to have a direct transition between the superfluid and the Laughlin state, one needs extra crystal symmetry to forbid the relevant terms $\bar{\psi}\sigma_a\psi$ with $a = 1, 2, 3$. In the CSL-SC transition on square lattice at filling $n = 1$, $\bar{\psi}\sigma_a\psi$ correspond to three CDW order parameters and these terms are forbidden by translation symmetry. This is actually quite generic at filling with odd number of electrons per unit cell on any lattice. With odd number of electrons per unit cell, the Luttinger theorem requires a Fermi surface with size $1/2$ of Brillouin zone for any symmetric phase without fractionalization. Because the transition happens below a spin gap, a Fermi liquid is impossible. Then there is no symmetric gapped phase without fractionalization according to the Lieb-Schultz-Mattis (LSM)[17–19]. On the other hand, if one adds a $\bar{\psi}\sigma_a\psi$ term, one can reach a gapped phase without fractionalization following Eq. 52[60]. This phase nevertheless needs to break symmetry, i.e. $\bar{\psi}\sigma_a\psi$ must carry a non-trivial quantum number under lattice symmetry. Therefore the intertwinement of three other symmetry breaking order parameters at the CSL-SC transition is guaranteed by the Lieb-Schultz-Mattis (LSM) theorem for odd number of electrons per unit cell. Exactly at the QCP, there is an emergent $SO(3)$ symmetry rotating these three order parameters and the vortex of the SC order needs to carry $1/2$ spin under this $SO(3)$ rotation. As we will explicitly demonstrate below, at CSL-SC transition on triangular lattice and kagome lattice, $\bar{\psi}\vec{\sigma}\psi$ also corresponds to three CDW orders.

## VIII. CSL TO CDW TRANSITIONS

In the previous two sections we discuss two critical theories for the CSL-SC transition, which are argued to be dual to each other. The $U(1)_{-2}$ $2\varphi$ theory can be naturally derived from the type I $U(1)$ ansatz for the CSL phase. In this section we explore the other possibility starting from the type II $U(1)$ ansatz. As argued in the previous section, CSL to SC transition is still possible from a plateau transition of bosonic holon. However, the simple condensation of bosonic holons in this case leads to a charge density wave (CDW) Chern insulator phase. We discuss critical theories associated with the CSL-CDW transitions. Note that the symmetries for the CDW on square and triangular lattice are very different. On both lattices, there are three different CDW orders labeled as $(n_3, n_4, n_5)$. On triangular/kagome lattice, they have momenta $\mathbf{M}_1, \mathbf{M}_2, \mathbf{M}_3$ and are related by $C_6$ rotation symmetry. On the other hand, on square lattice, $(n_3, n_4)$ carry momenta $(\pi, 0)$ and $(0, \pi)$, and are related by $C_4$ rotation. $n_5$ carries momentum $(\pi, \pi)$ and is distinct from $(n_3, n_4)$. As a result, one goes to the CDW $(n_3, n_4)$ or $n_5$ depending on anisotropy terms. As an analog to Neel order, CDW $(n_3, n_4)$ can be called as CDW$_{xy}$ and $n_5$ CDW$_z$. On square lattice, we have CSL-CDW$_{xy}$ transition or CSL-CDW$_z$ transition depending on whether it is easy plane anisotropy or easy axis anisotropy. On triangular/kagome lattice, CDW$_{xy}$ and CDW$_z$ are related by symmetry. The CSL-CDW transition will be shown to be the same as the tri-critical point between CSL, CDW$_{xy}$ and CDW$_z$ on square lattice.

### A. CSL-CDW$_{xy}$ transition on square lattice

Following the same analysis in Sec. VI, from the type II $U(1)$ ansatz of CSL phase, one can easily obtain a critical theory for CSL-CDW transition in the holon condensation picture. The type II ansatz is equivalent to adding a perturbation $H' = -\Phi^\dagger \tau_3 \Phi$ to the $SU(2)$ ansatz (See Sec. IV). The low energy holon fields are $\varphi_1 = \Phi_{1;1}$ and $\varphi_2 = \Phi_{1;2}$, where $\Phi_1$ and $\Phi_2$ are the $SU(2)$ spinors introduced in the $SU(2)$ ansatz in Sec. III. The symmetry actions of $\varphi = (\varphi_1, \varphi_2)$ can be derived from Table. I and are shown in Table. VII. The difference from the theory in Sec. VI is that now the action of $U(1)_c$ and $U(1)_r$ get exchanged. As a result, $\varphi_1^* \varphi_2$ is now the easy plane CDW order $n_3 + in_4$ and the monopole operator $\tilde{\mathcal{M}} = \mathcal{M}(\varphi^\dagger i\sigma_2 \vec{\sigma}\varphi^*) \cdot (\varphi^\dagger \vec{\sigma}\varphi)$ is now the SC order $n_1 + in_2$. $\varphi^\dagger \sigma_3 \varphi$ remains as the easy axis CDW $n_5$.

| | $T_1$ | $T_2$ | $\tilde{C}_4$ | $R_1\mathcal{T}$ | C | $U(1)_c$ | $U(1)_r$ | comment |
|---|---|---|---|---|---|---|---|---|
| $\varphi = (\varphi_1, \varphi_2)^T$ | $i\sigma_1\varphi$ | $i\sigma_2\varphi$ | $-ie^{i\frac{\pi}{4}\sigma_3}\varphi$ | $i\sigma_1\varphi$ | $\varphi^*$ | $e^{i\frac{1}{2}\sigma_0\theta}\varphi$ | $e^{i\frac{1}{2}\sigma_3\theta}\varphi$ | |
| $\varphi^\dagger\sigma_1\varphi$ | $+$ | $-$ | $-\varphi^\dagger\sigma_2\varphi$ | $+$ | $+$ | $\varphi^\dagger\sigma_1\varphi$ | $\varphi^\dagger(\cos\theta\sigma_1 + \sin\theta\sigma_2)\varphi$ | $n_3$ |
| $\varphi^\dagger\sigma_2\varphi$ | $-$ | $+$ | $\varphi^\dagger\sigma_1\varphi$ | $+$ | $-$ | $\varphi^\dagger\sigma_2\varphi$ | $\varphi^\dagger(-\sin\theta\sigma_1 + \cos\theta\sigma_2)\varphi$ | $n_4$ |
| $\mathrm{Re}\,\mathcal{M}_a$ | $+$ | $+$ | $-$ | $+$ | $+$ | $\cos\theta\mathrm{Re}\,\mathcal{M}_a + \sin\theta\mathrm{Im}\,\mathcal{M}_a$ | $\mathrm{Re}\,\mathcal{M}_a$ | $n_1$ |
| $\mathrm{Im}\,\mathcal{M}_a$ | $+$ | $+$ | $-$ | $-$ | $-$ | $-\sin\theta\mathrm{Re}\,\mathcal{M}_a + \cos\theta\mathrm{Im}\,\mathcal{M}_a$ | $\mathrm{Im}\,\mathcal{M}_a$ | $n_2$ |
| $\varphi^\dagger\sigma_3\varphi$ | $-$ | $-$ | $+$ | $-$ | $+$ | $\varphi^\dagger\sigma_3\varphi$ | $\varphi^\dagger\sigma_3\varphi$ | $n_5$ |

TABLE VII: Symmetry transformations in the $U(1)_{-2}$ $2\varphi$ theory for the CSL-CDW transition on square lattice. C is the charge conjugation and $R_1\mathcal{T}$ is anti-unitary. Symmetry actions for $\varphi^\dagger\sigma_a\varphi$ inherits from that of the $SU(2)$ ansatz in Table. I. $\mathcal{M}_a = \mathcal{M}_a^0(\varphi^\dagger i\sigma_2\vec{\sigma}\varphi^*) \cdot (\varphi^\dagger\vec{\sigma}\varphi)$, where $\mathcal{M}_a^0$ is the bare monopole operator.

A critical theory similar to Eq. 38 can be written down in terms of $\varphi = (\varphi_1, \varphi_2)^T$:

$$\mathcal{L}_{u1-csl-cdw} = |(\partial_\mu - ia_\mu - i\frac{1}{2}A_{c;\mu} - i\frac{1}{2}\sigma_3 A_{r;\mu})\varphi|^2 - r|\varphi|^2$$
$$+ \frac{2}{4\pi}ada - g(|\varphi|^2)^2 + \lambda|\varphi_1|^2|\varphi_2|^2 - \frac{1}{8\pi}A_c dA_c \qquad (55)$$

where we still stack a $\nu = -2$ IQHE phase to cancel the quantum spin Hall response. $A_{c;\mu}$ and $A_{r;\mu}$ are probing fields as in Eq. 38. The only difference from Eq. 38 is that $A_c$ and $A_r$ get exchanged.

By a redefinition $a_\mu \to a_\mu - \frac{1}{2}A_\mu^c$ one obtains a new version:

$$\mathcal{L} = |(\partial_\mu - ia_\mu - i\frac{1}{2}A_{r;\mu}\sigma_3)\varphi|^2 - r|\varphi|^2$$
$$- g(|\varphi|^2)^2 + \lambda|\varphi_1|^2|\varphi_2|^2 + \frac{2}{4\pi}ada - \frac{1}{2\pi}A_c da. \qquad (56)$$

For now let us assume $\lambda > 0$, corresponding to easy-plane anisotropy. If $r > 0$, $\varphi$ is gapped and we are left with the following Lagrangian:

$$\mathcal{L} = \frac{2}{4\pi}ada - \frac{1}{2\pi}A_c da \qquad (57)$$

which is the CSL phase.

When $r < 0$, $\varphi$ needs to condense. For the fixed point with $\lambda > 0$, $\varphi \propto (1,1)^T$. This will higgs both $a_\mu$ and $A_{r;\mu}$, so $a_\mu$ is gapped and we have a superfluid phase for $A_{r;\mu}$. There is no other term left. Superfluid of $A_r$ means that there is a symmetry breaking order parameter $(n_3, n_4)$, i.e. a CDW insulator. Note that we have stacked a $\nu = -2$ IQHE phase. Without the stacking, the CDW phase is a Chern insulator with $C = 2$.

Table VII lists the symmetries of bilinears in $\varphi$ descending from those in the $SU(2)$ theory table I. $\varphi^\dagger\sigma_i\varphi$ transform as the 3 CDW order parameters. We have the transform of monopoles in the fourth and fifth line of table VII. The symmetry transformations are inferred from assuming that they are the same as in the $SU(2)$ theory described in section IX because Higgs term should not alter the symmetry properties of the order parameters.

The CSL-CDW$_{xy}$ critical theory is the same as the CSL-SC critical theory under exchange of $A_c \leftrightarrow A_r$. Then it is also dual to a $U(1)_{-1}$ $2\psi$ theory. Following the procedure to derive the boson-fermion duality in Sec. VII, the $U(1)_1$ $2\psi$ critical theory for the CSL-CDW$_{xy}$ QCP reads:

$$\mathcal{L} = \bar{\psi}\gamma_\mu(-i\partial_\mu - b_\mu - \frac{1}{2}A_{c;\mu}\sigma_3)\psi - m\bar{\psi}\psi$$
$$+ \frac{1}{2\pi}A_r db - \frac{1}{8\pi}A_r dA_r - \frac{1}{16\pi}A_c dA_c \qquad (58)$$

One can check that the $m < 0$ gives the CSL phase and $m > 0$ describes a superfluid phase of $A_r$. In the Dirac theory, there is again a $SO(3)$ symmetry. But now the $SO(3)$ vector $\bar{\psi}\vec{\sigma}\psi$ corresponds to the order parameter $(n_1, n_2, n_5)$. Then the duality implies that at the CSL-CDW$_{xy}$ QCP, the superconductor order and the easy axis CDW$_z$ order $n_5$ forms a $SO(3)$ vector together. In total, this QCP should have $SO(3) \times O(2)$ symmetry if we further include the $U(1)$rotation $U(1)_r$. The CSL-CDW$_{xy}$ QCP is dual to the CSL-SC QCP upon exchange of $(n_1, n_2)$ and $(n_3, n_4)$.

## B. CSL-CDW$_z$ transition on square lattice

We have shown that the critical theory in Eq. 56 with easy-plane anisotropy $\lambda > 0$ describes the transition between CSL and the CDW$_{xy}$ order. Now consider the easy-axis anisotropy $\lambda < 0$, then the $r < 0$ side selects the condensation $\varphi = (1,0)^T$ or $\varphi = (0,1)^T$ and one has the CDW$_z$ order. So $\lambda < 0$ corresponds to the CSL-CDW$_z$ transition.

One interesting observation is that the $\lambda < 0$ case of Eq. 40 also describes the CSL-CDW$_z$ transition. These two theories are related to each other by exchange of $A_c$ and $A_r$. Therefore, the CSL-CDW$_z$ QCP is self dual under exchange of $A_c$ and $A_r$. CDW$_z$ order corresponds to $\varphi^\dagger\sigma_3\varphi$ in both theories. However, $n_1 + in_2$ (or $n_3 + in_4$) corresponds to $\varphi^\dagger\sigma_1\varphi$ in one theory and the monopole operator $\mathcal{M}_a$ in the other theory. This suggests that for $\lambda < 0$, the $U(1)_{-2}$ $2\varphi$ theory has a hidden symmetry which relates $\varphi^\dagger\sigma_{1,2}\varphi$ to the monopole operator $\mathcal{M}_a$. Indeed, as shown in Sec. IX, the CSL-CDW$_z$ has an $O(4)$ symmetry rotating the vector $(n_1, n_2, n_3, n_4)$.

### C. CSL-CDW$_{xy}$-CDW$_z$ tricritical point on square lattice and CSL-CDW transition on triangular/kagome lattice

We have shown that $\lambda > 0$ and $\lambda < 0$ of Eq. 56 corresponds to the CSL-CDW$_{xy}$ and CSL-CDW$_z$ transitions on square lattice. Then naturally $\lambda = 0$ is the tri-critical point between CSL, CDW$_{xy}$ and CDW$_z$. On the other hand, on triangular/kagome lattice, there is no easy-plane or easy-axis anisotropy for the three CDW orders. $\lambda = 0$ is required by the $C_6$ rotation and the tri-critical point now becomes a bi-critical point on triangular/kagome lattice between CSL and isotropic CDW phase. At this QCP, there is a $SO(3) \times O(2)$ symmetry. $SO(3)$ rotates the three CDW orders $(n_3, n_4, n_5)$.

We list the symmetry actions for $\varphi$ and the five order parameters on triangular/kagome lattice in table VIII and IX. They can be derived from the type II $U(1)$ ansatz of the CSL phase by adding $-\Phi^\dagger\tau_3\Phi$ term to the $SU(2)$ ansatz.

| | $T_1$ | $T_2$ | $C_6$ | $R\mathcal{T}$ | C | $U(1)_c$ | note |
|---|---|---|---|---|---|---|---|
| $\varphi=(\varphi_1,\varphi_2)^T$ | $-i\sigma_1\varphi$ | $-i\sigma_3\varphi$ | $e^{i\frac{\pi}{3}}e^{-i\frac{\sigma_1+\sigma_2+\sigma_3}{\sqrt{3}}\frac{\pi}{3}}\varphi$ | $e^{-i\frac{\pi}{12}}e^{-i\sigma_1\frac{\pi}{4}}\varphi$ | $\varphi^*$ | $e^{i\frac{1}{2}\sigma_0\theta}\varphi$ | |
| $\varphi^\dagger\sigma_1\varphi$ | $+$ | $-$ | $\varphi^\dagger\sigma_2\varphi$ | $+$ | $+$ | $\varphi^\dagger\sigma_1\varphi$ | $n_3$ |
| $\varphi^\dagger\sigma_2\varphi$ | $-$ | $-$ | $\varphi^\dagger\sigma_3\varphi$ | $\varphi^\dagger\sigma_3\varphi$ | $-$ | $\varphi^\dagger\sigma_2\varphi$ | $n_4$ |
| $\varphi^\dagger\sigma_3\varphi$ | $-$ | $+$ | $\varphi^\dagger\sigma_1\varphi$ | $\varphi^\dagger\sigma_2\varphi$ | $+$ | $\varphi^\dagger\sigma_3\varphi$ | $n_5$ |
| Re $\mathcal{M}_a$ | $+$ | $+$ | $\cos(\frac{2\pi}{3})$Re $\mathcal{M}_a+\sin(\frac{2\pi}{3})$Im $\mathcal{M}_a$ | $\cos(\frac{\pi}{6})$Re $\mathcal{M}_a+\sin(\frac{\pi}{6})$Im $\mathcal{M}_a$ | $+$ | $\cos\theta$Re $\mathcal{M}_a+\sin\theta$Im $\mathcal{M}_a$ | $n_1$ |
| Im $\mathcal{M}_a$ | $+$ | $+$ | $\cos(\frac{2\pi}{3})$Im $\mathcal{M}_a+\sin(\frac{2\pi}{3})$Re $\mathcal{M}_a$ | $-\cos(\frac{\pi}{6})$Im $\mathcal{M}_a-\sin(\frac{\pi}{6})$Re $\mathcal{M}_a$ | $-$ | $-\sin\theta$Re $\mathcal{M}_a+\cos\theta$Im $\mathcal{M}_a$ | $n_2$ |

TABLE VIII: Symmetries of boson bilinears and monopoles for $U(1)_{-2}\ 2\varphi$ theory on triangular lattices to describe CSL-CDW transition with Eq. 56 with $\lambda = 0$. It can be viewed as descending from $SU(2)$ theory eq (59) by adding a mass $\Phi^\dagger\sigma_3\Phi$. We define $U(1)_r$ to be $\varphi \to e^{i\frac{1}{2}\sigma_3\theta}\varphi$ to preserve Eq.(56). Like on the square lattice, $U(1)_r$ rotates 2 CDW order parameters $(n_3, n_4) \sim (\varphi^\dagger\sigma_1\varphi, \varphi^\dagger\sigma_2\varphi)$ and preserves other operators.

| | $T_1$ | $T_2$ | $C_6$ | $R\mathcal{T}$ | C | $U(1)_c$ | comment |
|---|---|---|---|---|---|---|---|
| $\varphi=(\varphi_1,\varphi_2)^T$ | $i\sigma_2\varphi$ | $i\sigma_3\varphi$ | $e^{i\frac{\pi}{6}}e^{-i\frac{\sigma_1+\sigma_2+\sigma_3}{\sqrt{3}}\frac{\pi}{3}}\varphi$ | $e^{-i\frac{\pi}{4}}e^{-i\frac{-\sigma_3+\sigma_1}{\sqrt{2}}\frac{\pi}{2}}\varphi$ | $\varphi^*$ | $e^{i\frac{1}{2}\sigma_0\theta}\varphi$ | |
| $\varphi^\dagger\sigma_1\varphi$ | $-$ | $-$ | $\varphi^\dagger\sigma_2\varphi$ | $-\varphi^\dagger\sigma_3\varphi$ | $+$ | $\varphi^\dagger\sigma_1\varphi$ | $n_3$ |
| $\varphi^\dagger\sigma_2\varphi$ | $+$ | $-$ | $\varphi^\dagger\sigma_3\varphi$ | $-$ | $-$ | $\varphi^\dagger\sigma_2\varphi$ | $n_4$ |
| $\varphi^\dagger\sigma_3\varphi$ | $-$ | $+$ | $\varphi^\dagger\sigma_1\varphi$ | $-\varphi_1^\dagger\sigma_1\varphi$ | $+$ | $\varphi^\dagger\sigma_3\varphi$ | $n_5$ |
| Re $\mathcal{M}_a$ | $+$ | $+$ | $\cos(\frac{\pi}{3})$Re $\mathcal{M}_a+\sin(\frac{\pi}{3})$Im $\mathcal{M}_a$ | Im $\mathcal{M}_a$ | $+$ | $\cos\theta$Re $\mathcal{M}_a+\sin\theta$Im $\mathcal{M}_a$ | $n_1$ |
| Im $\mathcal{M}_a$ | $+$ | $+$ | $\cos(\frac{\pi}{3})$Im $\mathcal{M}_a+\sin(\frac{\pi}{3})$Re $\mathcal{M}_a$ | Re $\mathcal{M}_a$ | $-$ | $-\sin\theta$Re $\mathcal{M}_a+\cos\theta$Im $\mathcal{M}_a$ | $n_2$ |

TABLE IX: Symmetries of boson bilinears and monopoles for $U(1)_{-2}\ 2\varphi$ theory on Kagome lattices to describe CSL-CDW transitions. It can be viewed as descending from $SU(2)$ theory eq (59) by adding a mass $\Phi^\dagger\sigma_3\Phi$. We define $U(1)_r$ to be $\varphi \to e^{i\frac{1}{2}\sigma_3\theta}\varphi$ to preserve Eq.(56). Like on the square lattice, $U(1)_r$ rotates 2 CDW order parameters $(n_3, n_4) \sim (\varphi^\dagger\sigma_1\varphi, \varphi^\dagger\sigma_2\varphi)$ and preserves other operators.

### D. CSL-CDW$_z$-SC tricritical point on square lattice

We comment on the $\lambda = 0$ point of the CSL-SC critical theory Eq. 40. In this case $\lambda > 0$ describes the CSL-SC transition and $\lambda < 0$ describes the CSL-CDW$_z$ transition on square lattice. Then naturally $\lambda = 0$ is a tri-critical point on square lattice. This tri-critical point is dual to the tri-critical point between CSL-CDW$_{xy}$-CDW$_z$ because the action is in the same form up to an exchange of $A_c$ and $A_r$. Again one expects a $SO(3) \times O(2)$ symmetry with $SO(3)$ symmetry rotating $(n_1, n_2, n_5)$ now.

### IX. $SU(2)$ THEORY: A UNIFIED FRAMEWORK FOR CSL-SC AND CSL-CDW TRANSITIONS

As shown in previous sections, the CSL can have either $SU(2)$ ansatz or $U(1)$ ansatz at the mean field level. Both ansatz describe the same topological order. We can describe the CSL-SC transition starting from either ansatz. For the $U(1)$ ansatz, there are two types. In type I $U(1)$ ansatz, the mean field theory of spinon $f_\sigma$ can be gauge transformed to that of a translation invariant superconductor. Then CSL-SC transition can be captured by condensation of bosonic holons as demonstrated in Sec. VI. In contrast, for the type II $U(1)$ ansatz, the mean field theory of spinon $f_\sigma$ is not gauge equivalent to a translation invariant superconductor. In this case holon condensation leads to CDW order instead of superconductor as discussed in Sec. VIII. Even for this case, we can still reach a SC phase if the bosonic holon goes through a

plateau transition, as shown in Sec. VII. The final theory contains two Dirac fermions and is argued to be dual to the theory with bosonic holon fields. The shortcoming of the $U(1)_1$ theory with two Dirac fermions is that the microscopic symmetry actions on the Dirac fermions are not transparent. On triangular lattice and kagome lattice, there is no type I $U(1)$ ansatz, hence there is no obvious critical theory with two complex bosons $\varphi$ coupled to $U(1)$ gauge field to describ CSL-SC transition.

In this section, we will start from the $SU(2)$ ansatz for the CSL and derive a new critical theory for the CSL-SC transition where there are two $SU(2)$ bosonic spinors $\Phi_1, \Phi_2$ coupled to an $SU(2)$ gauge field with chern-simons term at level $-1$. Because the $SU(2)$ ansatz describes the same CSL phase as the $U(1)$ ansatz, we will argue that this $SU(2)$ critical theory is dual to the $U(1)_{-2}$ theory with $2\varphi$ and the $U(1)_1$ theory with $2\psi$ in the previ-

ous two sections. This offers a new perspective on the critical point. In the $SU(2)$ theory, the five order parameters are all bilinears of the bosonic fields and there is no monopole. Therefore the symmetry actions on the five order parameters can be easily obtained from mean field ansatz. In the $SU(2)$ theory, one can identify other fixed points corresponding to CSL-CDW transition and tricritical points at the intersection of CSL, SC and CDW. Therefore $SU(2)$ theory offers a unified framework to capture all critical theories discussed in the previous sections. Enlarged symmetry and self duality at certain fixed points in the $SU(2)$ theory are shown explicitly.

### A. $SU(2)$ theory on square lattice

| | $T_1$ | $T_2$ | $\tilde{C}_4$ | $R_1\mathcal{T}$ | C | $U(1)_c$ | $U(1)_r$ | comment |
|---|---|---|---|---|---|---|---|---|
| $\Phi = (\Phi_1, \Phi_2)^T$ | $-i\sigma_1\Phi$ | $-i\sigma_2\Phi$ | $-ie^{i\frac{\pi}{4}\sigma_3}\Phi$ | $i\sigma_1\Phi$ | $\Phi^*$ | $e^{i\frac{1}{2}\sigma_0\theta}\Phi$ | $e^{i\frac{1}{2}\sigma_3\theta}\Phi$ | |
| Re $\Phi^T\sigma_2\tau_2\Phi$ | $+$ | $+$ | $-$ | $+$ | $+$ | $\cos\theta$Re $\Phi^T\sigma_2\tau_2\Phi + \sin\theta$Im $\Phi^T\sigma_2\tau_2\Phi$ | $+$ | $n_1$ |
| Im $\Phi^T\sigma_2\tau_2\Phi$ | $+$ | $+$ | $-$ | $+$ | $-$ | $-\sin\theta$Re $\Phi^T\sigma_2\tau_2\Phi + \cos\theta$Im $\Phi^T\sigma_2\tau_2\Phi$ | $+$ | $n_2$ |
| $\Phi^\dagger\sigma_1\Phi$ | $+$ | $-$ | $\Phi^\dagger\sigma_2\Phi$ | $+$ | $+$ | $\Phi^\dagger\sigma_1\Phi$ | $\cos\theta$Re $\Phi^\dagger\sigma_1\Phi + \sin\theta\Phi^\dagger\sigma_2\Phi$ | $n_3$ |
| $\Phi^\dagger\sigma_2\Phi$ | $-$ | $+$ | $-\Phi^\dagger\sigma_1\Phi$ | $+$ | $-$ | $\Phi^\dagger\sigma_2\Phi$ | $-\sin\theta\Phi^\dagger\sigma_1\Phi + \cos\theta\Phi^\dagger\sigma_2\Phi$ | $n_4$ |
| $\Phi^\dagger\sigma_3\Phi$ | $-$ | $-$ | $+$ | $-$ | $+$ | $+$ | $+$ | $n_5$ |

TABLE X: Symmetry transformations in the SU(2)$_{-1}$ 2$\Phi$ theory for the CSL-SC transition on square lattice. C is the charge conjugation which exists only for the bandwidth tuned transition. Only $R_1\mathcal{T}$ is anti-unitary.

On square lattice, we start from the $SU(2)$ ansatz for the CSL listed in Sec. III A. As already shown in Sec. III A, at low energy there are two bosons $\Phi_1$ and $\Phi_2$ in the fundamental representation of the $SU(2)$ gauge field. They are related by translation symmetry and their degeneracy is guaranteed by the $T_1T_2 = -T_2T_1$ algebra. A critical theory can be written down corre-

sponding to the condensation of these two bosons. The symmetry transformations of $\Phi = (\Phi_1, \Phi_2)^T$ are listed in Table. X, which follows from Table. I. $U(1)_r$ symmetry and charge conjugation symmetry follows the same notation as in Sec. VI. Here $\sigma_a$ labels Pauli matrices acting in the $(\Phi_1, \Phi_2)$ space. $\tau_i$ labels generators of $SU(2)$ gauge field and acts in the $(\Phi_{a;1}, \Phi_{a;2})$ space for $a = 1, 2$. The $SU(2)$ critical theory is

$$\mathcal{L}_{SU(2)} = \sum_{i=1,2} |(\partial_\mu - ia_\mu^s\tau^s - i\frac{1}{2}A_{c;\mu}\tau_0\sigma_0 - \frac{1}{2}iA_{r;\mu}\tau_0\sigma_3)\Phi_i|^2 - r|\Phi|^2 + \frac{1}{4\pi}Tr[a \wedge da + \frac{2}{3}ia \wedge a \wedge a] - \frac{1}{8\pi}A_cdA_c - \mathcal{L}_{int}$$

$$\mathcal{L}_{int} = g|\Phi^\dagger\Phi|^2 + \lambda_0\mathbf{n}\cdot\mathbf{n} - \lambda(n_1^2 + n_2^2) - \lambda'(n_3^2 + n_4^2), \tag{59}$$

where $a^s, s = 1, 2, 3$ is an $SU(2)$ gauge field. $A_c$ and $A_r$ are the $U(1)$ probing fields for the $U(1)_c$ and $U(1)_r$ global symmetry. $\mathbf{n} = (n_1, n_2, n_3, n_4, n_5) = (\text{Re}\Phi^T\sigma_2\tau_2\Phi, \text{Im}\Phi^T\sigma_2\tau_2\Phi, \Phi^\dagger\sigma_1\Phi, \Phi^\dagger\sigma_2\Phi, \Phi^\dagger\sigma_3\Phi)$. $\frac{1}{4\pi}Tr[a \wedge da + \frac{2}{3}ia \wedge a \wedge a]$ is the Chern-Simons term for $SU(2)$ gauge field coming from the integration of the fermionic spinons. Here $a_\mu = \sum_{s=1,2,3} a_\mu^s\tau^s$. The $-\frac{1}{8\pi}A_cdA_c$ term again is from the stacking of the $\nu = -2$ IQHE phase to cancel the spin Hall effect. All symmetry allowed quartic terms are included(see Appendix. B). The theory has a charge conjugation symmetry

$$C : \Phi(x) \to \Phi^*(x), a_\mu(x) \to -a_\mu(x),$$
$$A_\mu^c(x) \to -A_\mu^c(x), A_\mu^r(x) \to -A_\mu^r(x). \tag{60}$$

Under $C$, $(n_1, n_2) \to (n_1, -n_2)$ and $(n_3, n_4, n_5) \to (n_3, -n_4, n_5)$. The same as the discussion for the $U(1)_{-2}$ 2$\varphi$ theory, the $C$ symmetry exists only for the bandwidth tuned transition. For the chemical potential tuned transition, there is a $i\Phi^\dagger\partial_t\Phi$ term and the dynamical exponent is $z = 2$. We focus on the bandwidth tuned transition and thus a charge conjugation symmetry is present.

The phase transition is tuned by the sign of $r$. When $r > 0$, $\Phi$ is gapped out and one is left with the $SU(2)_{-1}$ Chern Simons theory, which is known to be equivalent to the $U(1)_2$ CSL phase by level-rank duality. When $r < 0$, condensation of $\Phi$ higgses the $SU(2)$ gauge fields and leads to a symmetry breaking phase. There are various different possible phases corresponding to different con-

densation patterns of $\Phi$, which are decided by the quartic terms.

It can be shown that $\lambda > 0$ favors the SC order parameter $(n_1, n_2)$. In contrast, $\lambda' > 0$ favors the CDW order parameter $(n_3, n_4)$. $\lambda < 0, \lambda' < 0$ favors the CDW order $n_5$.

Therefore there should be several different fixed points in the parameter space $(\lambda, \lambda')$, corresponding to transition between the CSL and different symmetry breaking phases. The energy cost of different symmetry breaking orders can be found in Table. XI.

| Order parameter | SC $(n_1, n_2)$ | CDW $(n_3, n_4)$ | CDW $n_5$ |
|---|---|---|---|
| Condensation | $\Phi_1 = \frac{1}{\sqrt{2}}\begin{pmatrix} 1 \\ 0 \end{pmatrix}, \Phi_2 = \frac{1}{\sqrt{2}}\begin{pmatrix} 0 \\ 1 \end{pmatrix}$ | $\Phi_1 = \frac{1}{\sqrt{2}}\begin{pmatrix} 1 \\ 0 \end{pmatrix}, \Phi_2 = \frac{1}{\sqrt{2}}\begin{pmatrix} 1 \\ 0 \end{pmatrix}$ | $\Phi_1 = \begin{pmatrix} 1 \\ 0 \end{pmatrix}, \Phi_2 = \begin{pmatrix} 0 \\ 0 \end{pmatrix}$ |
| Energy | $-\lambda$ | $-\lambda'$ | $0$ |

TABLE XI: The role of quartic terms in the $SU(2)$ theory to select the symmetry breaking order parameter in the ordered side. The energy cost is defined on top of $g + \lambda_0$.

### B. $SU(2)$ theory on triangular and kagome lattices

On triangular and kagome lattices, we also have $SU(2)$ ansatz for the CSL phase. Again the low energy holon fields are captured by $\Phi_1$ and $\Phi_2$, whose symmetry transformations are listed in Table. II and in Table. III. As in square lattice, $T_1$ and $T_2$ relates $\Phi_1$ and $\Phi_2$ and protect their degeneracy. $\Phi^\dagger \vec{\sigma} \Phi$ all break translation symmetry and now carry momenta $\mathbf{M}_1, \mathbf{M}_2, \mathbf{M}_3$, corresponding to three CDW orders $(n_3, n_4, n_5)$. Together they form a three dimensional vector $\vec{n} = (n_3, n_4, n_5)$ and lattice symmetries act as one element of a $SO(3)$ rotation on $\vec{n}$. As on square lattice, $T_1$ and $T_2$ act as a $180°$ rotation around one of $\vec{n}_3, \vec{n}_4, \vec{n}_5$. In contrast to square lattice, the $C_6$ acts as an rotation around the direction along $\frac{1}{\sqrt{3}}(\vec{n}_3 + \vec{n}_4 + \vec{n}_5)$ and thus rotates $\vec{n}_3, \vec{n}_4, \vec{n}_5$ to each other. The $C_4$ on square lattice instead rotates around $\vec{n}_5$ and the CDW on square lattice has an easy-plane anisotropy meaning $(n_3, n_4)$ can not be rotated to $n_5$ by any symmetry.

A critical theory in terms of $\Phi = (\Phi_1, \Phi_2)^T$ reads as Eq. 59, albeit with the easy-plane anisotropy terms $\lambda'$ forbidden by the $C_6$ symmetry which rotates $n_3, n_4, n_5$ to each other. $U(1)_r$ symmetry is now enlarged to $SO(3)$, however we can still keep the probing field $A_r$, which acts as $\Phi \to e^{i\frac{1}{2}\sigma_3 \theta} \Phi$. In summary the critical theory is Eq. 59 with $\lambda' = 0$.

### C. Enlarged symmetry and duality

Here we discuss the symmetry in the $(\lambda, \lambda')$ space. It is convenient to construct a $2 \times 2$ matrix field for each field $\Phi_a$:

$$X_a = \frac{1}{\sqrt{2}} \begin{pmatrix} \Phi_{a;1} & \Phi_{a;2} \\ -\Phi_{a;2}^* & \Phi_{a;1}^* \end{pmatrix} \tag{61}$$

It can be shown that $(\Phi_{a;1}, \Phi_{a;2})^T$ and $(-\Phi_{a;2}^*, \Phi_{a;1}^*)^T$ transform in the same way under the $SU(2)$ gauge transformation. Therefore, the $SU(2)$ gauge transformation acts as:

$$X_a \to X_a U_g \tag{62}$$

where $U_g \in SU(2)$.

The four elements of $X_a$ are not independent. They are constrained by the condition:

$$X_a^* = \tau_2 X_a \tau_2. \tag{63}$$

Note that the gauge transformation $e^{i\tau_a \theta}$ acts on the right of $X_a$. On the other hand, $X_a \to U X_a$ with $U \in SU(2)$ generically does not belong to the gauge group.

We can then define a $4 \times 2$ matrix field: $X = \begin{pmatrix} X_1 \\ X_2 \end{pmatrix}$. Each element can be labeled as $X_{\sigma\tau;\tau'}$ with $\sigma = 1, 2$ as the index for the 'valley' degree of freedom. We need to apply the following constraint:

$$X^* = \sigma_0 \otimes \tau_2 X \tau_2 \tag{64}$$

where $\sigma_0 \otimes \tau_2$ is a $4 \times 4$ matrix.

The gauge invariant bilinear operators can be organized in $\text{Tr} X^\dagger \sigma_a \tau_b X$ with $a, b = 0, 1, 2, 3$. Among the 16 operators, we find that ten of them vanish (see Appendix. B). $\text{Tr} X^\dagger X = \Phi^\dagger \Phi$. The remaining five are the five symmetry breaking order parameters: $\mathbf{n} = (n_1, n_2, n_3, n_4, n_5) = (\text{Re}\Phi^T \sigma_2 \tau_2 \Phi, \text{Im}\Phi^T \sigma_2 \tau_2 \Phi, \Phi^\dagger \sigma_1 \Phi, \Phi^\dagger \sigma_2 \Phi, \Phi^\dagger \sigma_3 \Phi) = \text{Tr} X^\dagger(-\sigma_2\tau_2, -\sigma_2\tau_1, \sigma_1, \sigma_2\tau_3, \sigma_3)X$. $SU(2)$ gauge transformation acts as $X \to X U_g^\dagger$, $a_\mu \to U_g a_\mu U_g^\dagger - i U_g \partial_\mu U_g^\dagger$. The critical theory can be rewritten as:

$$\mathcal{L} = \text{Tr}(\partial_\mu X^\dagger + i a_\mu X^\dagger)(\partial_\mu X - i X a_\mu) + r \text{Tr} X^\dagger X + \frac{1}{4\pi} \text{Tr}[a \wedge da + \frac{2}{3} i a \wedge a \wedge a] - \mathcal{L}_{int} \tag{65}$$

where $a_\mu$ is the abbreviation of $a_\mu^s \tau_s$ with $s = 1, 2, 3$.

The interaction term reads:

$$\mathcal{L}_{int} = g|\text{Tr}X^\dagger X|^2 + \lambda_0 \mathbf{n} \cdot \mathbf{n} - \lambda(n_1^2 + n_2^2) - \lambda'(n_3^2 + n_4^2) \tag{66}$$

### 1. $SO(5) \rtimes Z_2^T$ symmetry at $\lambda = \lambda' = 0$

At $\lambda = \lambda' = 0$, there is a global symmetry SO(5). First, let us set $\lambda = \lambda' = \lambda_0 = 0$, then the action is invariant under $X \to UX$ where $U \in U(4)$. To satisfy the constraint in Eq. 64, we need $U^T \sigma_0 \otimes \tau_2 U = \sigma_0 \otimes \tau_2$, which forms an $Sp(4)$ group. Because $X \to -X$ is shared with the $SU(2)$ gauge transformation, the global symmetry is $SO(5) \cong Sp(4)/Z_2$. The $SO(5)$ group has 10 generators, which correspond to $X \to e^{i\Gamma\theta}X$ with $\Gamma = \sigma_0\tau_3, \sigma_3\tau_2, \sigma_0\tau_1, \sigma_1\tau_2, \sigma_3\tau_1, \sigma_0\tau_2, \sigma_1\tau_1, \sigma_3\tau_3, \sigma_2, \sigma_1\tau_3$. This $SO(5)$ symmetry rotates the five dimensional vector $\mathbf{n} = (n_1, n_2, n_3, n_4, n_5) = \text{Tr}X^\dagger(-\sigma_2\tau_2, -\sigma_2\tau_1, \sigma_1, \sigma_2\tau_3, \sigma_3)X$. More specifically, one can label the generator of $SO(5)$ as $L_{\alpha\beta}$ with $\alpha < \beta$ and $\alpha, \beta = 1, 2, 3, 4, 5$. One can check that $\Gamma = \sigma_0\tau_3, \sigma_3\tau_2, \sigma_0\tau_1, \sigma_1\tau_2, \sigma_3\tau_1, \sigma_0\tau_2, \sigma_1\tau_1, \sigma_3\tau_3, \sigma_2, \sigma_1\tau_3$ corresponds to $L_{12}, L_{13}, L_{14}, L_{15}, L_{23}, L_{24}, L_{25}, L_{34}, L_{35}, L_{45}$ up to a sign convention. Here $L_{\alpha\beta}$ generates an $SO(2)$ rotation in the $(n_\alpha, n_\beta)$ subspace. $\mathbf{n} \cdot \mathbf{n}$ is invariant under the $SO(5)$ rotation and $\mathcal{L}$ has the $SO(5)$ symmetry with a finite $\lambda_0$ as long as $\lambda = \lambda' = 0$. There is also an anti-unitary symmetry $R\mathcal{T}$, so the final symmetry is $SO(5) \rtimes Z_2^{T61}$.

### 2. $O(4) \rtimes Z_2^T$ symmetry at $\lambda = \lambda'$

Consider the high symmetry line with $\lambda = \lambda' \neq 0$ and $g \neq 0, \lambda_0 \neq 0$. The interaction term can be rewritten as:

$$\mathcal{L}_{int} = g\text{Tr}X^\dagger X + (\lambda + \lambda_0)(n_1^2 + n_2^2 + n_3^2 + n_4^2) + \lambda_0 n_5^2. \tag{67}$$

Then one still has an $SO(4)$ symmetry which rotates the vector $(n_1, n_2, n_3, n_4)$. $SO(4)$ has 4 generators, corresponding to $X \to e^{i\Gamma\theta}X$ with $\Gamma = \sigma_0\tau_3, \sigma_3\tau_2, \sigma_0\tau_1, \sigma_3\tau_1, \sigma_0\tau_2, \sigma_3\tau_3$. Meanwhile, there is a $Z_2$ action from $X \to e^{i\sigma_1\tau_3\frac{\pi}{2}}X$, which acts as $(n_1, n_2, n_3, n_4, n_5) \to (n_1, n_2, n_3, -n_4, -n_5)$, an improper rotation in the $(n_1, n_2, n_3, n_4)$ space. Together we have a $O(4)$ symmetry. Including the anti-unitary symmetry $R\mathcal{T}$, together the symmetry becomes $O(4) \rtimes Z_2^T$.

### 3. Duality under $\lambda \leftrightarrow \lambda'$

A special operation: $X \to e^{i\tau_2\frac{\pi}{4}}e^{i\sigma_3\tau_2\frac{\pi}{4}}X$ corresponds to $\Phi_1 \to -i\tau_2\Phi_1^*, \Phi_2 \to \Phi_2$. This belongs to a special element in the $SO(4)$ group discussed in the previous subsection for $\lambda = \lambda'$. This operation maps $(n_1, n_2, n_3, n_4, n_5) \to (n_3, n_4, -n_1, -n_2, n_5)$. It flips $A_{c;\mu} \leftrightarrow A_{r;\mu}$, but leaves the action invariant at the special line $\lambda = \lambda'$. When $\lambda \neq \lambda'$, it is no longer a symmetry. Instead, it induces a duality which maps one critical theory with $(\lambda, \lambda')$ to a different critical theory with

$(\lambda', \lambda)$. As shown below, the CSL-SC fixed point and the CSL-CDW$_{xy}$ fixed point on square lattice are related by this duality. This duality constrains the renormalization group (RG) flow in the $(\lambda, \lambda')$ space to be symmetric under the reflection $\lambda \leftrightarrow \lambda'$. This duality precisely corresponds to the duality between Eq. 40 and Eq. 56 with $A_c$ and $A_r$ exchanged.

### D. Various fixed points in the $SU(2)_{-1}$ $2\Phi$ theory

We now discuss possible fixed points in the $SU(2)_{-1}$ $2\Phi$ theory. There is one obvious relevant direction tuned by $r$. When $r > 0$, the CSL phase with $\Phi$ gapped occurs. When $r < 0$, $\Phi$ condenses and higgses the $SU(2)$ gauge field, leading to a symmetry breaking phase. The exact symmetry breaking pattern depends on the quartic term and there are various different fixed points in the $g, \lambda_0, \lambda, \lambda'$ space. $g, \lambda_0$ terms are $SO(5)$ invariant and presumably flow to a fixed point value. They do not select the symmetry breaking pattern after $r < 0$. Therefore one focuses on the $(\lambda, \lambda')$ space.

A schematic phase diagram and fixed points are shown in Fig. 6. First consider the triangular or kagome lattice. Then $C_6$ rotation guarantees $\lambda' = 0$. In this case, CSL-SC QCP must be the fixed point at $(\lambda, \lambda') = (\lambda_+^*, 0)$ with $\lambda_+^* > 0$, shown as the blue point in Fig. 6(a). Similarly there is a fixed point at $\lambda < 0$ for CSL-CDW transition, shown as light blue point in Fig. 6(a). The red point at $(\lambda, \lambda') = (0, 0)$ is then a tri-critical point on triangular/kagome lattice.

Next square lattice. We argue that the CSL-SC transition still corresponds to the same fixed point as on triangular lattice for the following two reasons: (I) The CSL phase and SC phase on square and triangular lattice are the same. So it is natural to expect that the QCP on these two lattices are also the same. (II) As discussed in the $U(1)_1$ $2\psi$ theory for the CSL-SC transition, there is an $SO(3) \times O(2)$ symmetry, which is possible only when $\lambda' = 0$. Now that the blue point is identified as the CSL-SC transition, we use the duality map $\lambda \leftrightarrow \lambda'$ to obtain another fixed point at $(\lambda, \lambda') = (0, \lambda_+^*)$ (the orange point in Fig. 6(a)), which corresponds to the CSL-CDW$_{xy}$ transition on square lattice. There should be a tri-critical point between CSL, SC and CDW$_{xy}$, though not necessarily at the $(\lambda, \lambda') = (0, 0)$ point. The tri-critical point, if exists, most naturally occurs somewhere along the $\lambda = \lambda'$ line marked as the pink point. Note that the RG flow constrained by the $\lambda \leftrightarrow \lambda'$ duality fixes this fixed point to be along the $\lambda = \lambda'$ line, so it has an $O(4)$ symmetry rotating $(n_1, n_2, n_3, n_4)$.

The light blue point is believed to describe the CSL-CDW transition on triangular/kagome lattice. According to the analysis in Sec. VIII C, it should also be the tri-

critical point between CSL, CDW$_z$, CDW$_{xy}$ on square lattice. Using the $\lambda \leftrightarrow \lambda'$ duality, we know there is another fixed point labeled as the grey point. This should be the tri-critical point between CSL, SC, CDW$_x$ (see Sec. VIII D). The RG flow suggests a fixed point along the line $\lambda = \lambda' < 0$. This yellow point is exactly the CSL-CDW$_z$ critical point discussed in Sec. VIII B. This QCP has an additional symmetry relating $(n_1, n_2)$ to $(n_3, n_4)$. The $SU(2)$ theory enables one to make a stronger statement that there is an $O(4)$ symmetry rotating $(n_1, n_2, n_3, n_4)$ and a self-duality symmetry from $\lambda \leftrightarrow \lambda'$ for the CSL-CDW$_z$ QCP.

The $SU(2)$ theory offers a unified framework for all of the critical points and tri-critical points discussed in previous sections using bosonic holons or Dirac fermions in $U(1)$ theories. The blue and orange points are known to be dual to the $U(1)_1$ $2\psi$ theory. The blue, light blue, orange, yellow and grey points can all be described by $U(1)_{-2}$ $2\varphi$ theory in Eq. 40 and Eq. 56 by tuning the easy-plane anisotropy term $\lambda$. The red and pink point, however, are not captured by the simple $U(1)_{-2}$ $2\varphi$ theory. $SU(2)$ theory proves to be the most convenient way to describe these two tri-critical points.

Although not shown in Fig. 6(a), the SC-CDW$_{xy}$ transition on square lattice and SC-CDW transition on triangular/kagome lattice are in the same universality class as the famous DQCP between Neel order and VBS order with or without easy-plane anisotropy respectively. The DQCP can be viewed as descendant from the tri-critical points (the red and pink fixed point). Let us start from the red fixed point at $\lambda = \lambda' = 0$ with a $SO(5)$ symmetry. After $r < 0$, the CSL phase is higgsed and the theory should be reduced to a non-linear sigma model in terms of $\vec{n} = (n_1, n_2, n_3, n_4, n_5)$. It can be derived that the $SU(2)_{-1}$ Chern-Simons term leads to a Wess-Zumino-Witten (WZW) term with level $k = 1$ for the non-linear sigma model in terms of $\vec{n}$[62], which exactly corresponds to the isotropic DQCP[2]. With an easy-plane anisotropy, the $SO(5)$ non-linear sigma model with WZW term reduces to an $O(4)$ non-linear sigma model with $\theta = \pi$, corresponding to the easy-plane DQCP between CDW$_{xy}$ and SC. This will be discussed in more details in Sec. XI.

The CDW$_{xy}$-SC and CDW$_{xy}$-CDW$_z$ transitions should be first order because there are only three independent order parameters and there is no non-trivial Wess-Zumino-Witten term or $\theta$ term for order parameters living on $S^2$ manifold in $2+1$ dimension.

## X. CSL-SC AND CSL-CDW$_{xy}$ TRANSITION: DUALITY BETWEEN $SU(2)_{-1}$ $2\Phi$ AND $U(1)_{-2}$ $2\varphi$ THEORY

We have shown that the blue and the orange fixed point in Fig. 6 describe the CSL-SC and CSL-CDW$_{xy}$ transition. Sec. VI and Sec. VII also discussed $U(1)$ theories with $2\varphi$ or $2\psi$ for the same CSL-SC transition. The same is true for the CSL-CDW$_{xy}$ transition discussed in

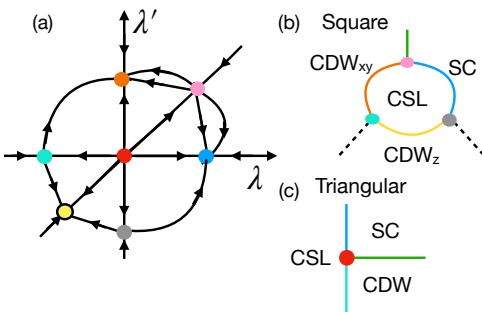

FIG. 6: (a)Fixed points of the $SU(2)$ theory with $N_b = 2$ bosons. On triangular and kagome lattice, $\lambda' = 0$ is enforced by lattice symmetry. There is a duality transformation $\lambda \leftrightarrow \lambda'$. The red point has $SO(5)$ symmetry. Pink and bright yellow points have $O(4)$ symmetry.(b,c) plot phase diagram on square and triangular lattice, respectively. The critical line or point has the same color as the fixed points in renormalization group(RG) flow of (a). Note that fixed points in the RG flow diagram may correspond to phase boundaries (line) or the intersection points of 3 phases in (b,c) depending on the lattice and symmetries. The green line in (b,c) separating superconducting and CDW phases are described by isotropic or easy-plane DQCP not displayed in the flow diagram (a). The dashed line represents first-order transitions.

Sec. VIII. It is natural to expect these three theories are dual to each other at the blue and orange fixed points in Fig. 6(a). The boson-fermion duality between the two $U(1)$ theories have already been demonstrated. Here we discuss the duality between the $SU(2)$ $2\Phi$ and $U(1)$ $2\varphi$ theories. The duality between the $SU(2)$ $2\Phi$ and $U(1)$ $2\psi$ was already proposed previously[62–64]. One interesting property about the CSL-SC and CSL-CDW$_{xy}$ transition is the enlarged $SO(3)$ symmetry among $(\varphi^\dagger \sigma_3 \varphi, \mathcal{M}_{a(b)})$, which is not obvious in the $U(1)$ $2\varphi$ theory. This section provides an understanding of this $SO(3)$ symmetry in the $U(1)$ $2\varphi$ theory. The CSL-CDW$_{xy}$-CDW$_z$ tri-critical point also has $SO(3)$ symmetry, simply from fine tuning to $\lambda = 0$. An enlarged $SO(3)$ theory in the $U(1)$ $2\varphi$ theory with easy-plane anisotropy $\lambda > 0$ in Eq. 40 or Eq. 56 is more nontrivial.

Starting from the $SU(2)_{-1}$ $2\Phi$ theory defined in Eq. 59, one obtains a $U(1)_{-2}$ $2\varphi$ theory by higgsing the $SU(2)$ gauge field down to $U(1)$ generated by $\tau_3$. There are two different types of the Higgs term:

(I) $-\vec{m} \cdot \Phi^\dagger \vec{\sigma} \tau_3 \Phi$,

(II) $\vec{m} \cdot (\text{Re}\Phi^T \sigma_1 \tau_1 \Phi, \text{Im}\Phi^T \sigma_1 \tau_1 \Phi, \Phi^\dagger \tau_3 \Phi)$,

where $\vec{m}$ is a three dimensional unit vector. These two groups are related by the duality transformation $X \to e^{i\tau_2 \frac{\pi}{4}} e^{i\sigma_3 \tau_2 \frac{\pi}{4}} X$ or equivalently $\Phi_1 \to -i\tau_2 \Phi_1^*, \Phi_2 \to \Phi_2$ (see Appendix B). The type I $U(1)$ and type II $U(1)$ ansatz in Sec. IV are obtained with $\vec{m} = (0, 0, 1)$ in the

two groups respectively. In the first group, other values of $\vec{m}$ can be generated from $\vec{m} = (0,0,1)$ with an $SO(3)$ rotation generated by $L_{34}, L_{35}, L_{45}$, which rotate in the subspace of $(n_3, n_4, n_5)$. On the other hand, $\vec{m}$ in the second group is generated from $\vec{m} = (0,0,1)$ with a $SO(3)$ rotation in the subspace of $(n_1, n_2, n_5)$.

The $SU(2)_{-1}$ theory will flow to the $U(1)$ $2\varphi$ theory in Eq. 40 and Eq. 56 by adding the Higgs term in the two groups respectively, corresponding to the CSL-SC and CSL-CDW$_{xy}$ transition. The $SU(2)$ theories at the fixed point $(\lambda, \lambda') = (\lambda_+^*, 0)$ and $(\lambda, \lambda') = (0, \lambda_+^*)$ are dual to Eq. 40 and Eq. 56 respectively. There is a manifold of $U(1)$ theories specified by the vector $\vec{m}$ in the Higgs term rotated by $SO(3)$ symmetry.

We discuss CSL-SC transition as an example. CSL-CDW$_{xy}$ is in parallel as related by the $\lambda \leftrightarrow \lambda'$ duality. For the CSL-SC transition, we propose the following duality:

$$
\begin{aligned}
\mathcal{L}_{SU(2)} &= |(\partial_\mu - ia_\mu^s \tau^s - i\frac{1}{2}A_{c;\mu}\tau_0\sigma_0)\Phi_a|^2 - r|\Phi|^2 + \frac{1}{4\pi}Tr[a \wedge da + \frac{2}{3}ia \wedge a \wedge a] - \frac{1}{8\pi}A_c dA_c \\
&\quad - g|\Phi^\dagger\Phi|^2 - \lambda_0 \mathbf{n}\cdot\mathbf{n} + \lambda(n_1^2 + n_2^2) \\
&\leftrightarrow \mathcal{L}_{SU(2)} - h\Phi^\dagger\vec{m}\cdot\vec{\sigma}\tau_3\Phi \\
&\leftrightarrow \mathcal{L}_{U(1),\vec{m}} = |(\partial_\mu - ia_\mu - i\frac{1}{2}A_{c;\mu}\sigma_3)\varphi|^2 - r|\varphi|^2 + \frac{2}{4\pi}ada - \frac{1}{8\pi}A_c dA_c - \tilde{g}(|\varphi|^2)^2 + \tilde{\lambda}|\varphi_1|^2|\varphi_2|^2.
\end{aligned}
\tag{68}
$$

$\varphi = (\varphi_1, \varphi_2)^T$ descends from $\Phi = (\Phi_1, \Phi_2)^T$ after adding the Higgs term $-h\Phi^\dagger\vec{m}\cdot\vec{\sigma}\tau_3\Phi$ on energetic grounds. For example, if $\vec{m} = (0,0,1)$, we have $\varphi_1 = \Phi_{1;1}$ and $\varphi_2 = \Phi_{2;2}^*$. The probing field $A_r$ is omitted because the $U(1)_r$ is not explicit in the $U(1)$ theory for a generic $\vec{m}$, unless $\vec{m} = (0,0,\pm1)$.

Note that $L_{U(1),\vec{m}}$ is gauge equivalent to $L_{U(1),-\vec{m}}$ after a gauge transformation $\Phi \to i\tau_1\Phi$. Therefore the real manifold of $U(1)_{-2}$ $2\varphi$ theories specified by a unit vector $\vec{m}$, is $RP^2 = S^2/Z_2$ after one mods out the equivalence between $\vec{m}$ and $-\vec{m}$. For any $\vec{m}$, $L_{U(1),\vec{m}}$ reduces to the $U(1)_{-2}$ $2\varphi$ theory, albeit the symmetry actions are different as discussed in the following. The implication of the duality is that all of these seemingly gauge nonequivalent theories flow to the same IR fixed point, which is also the $(\lambda, \lambda') = (\lambda_+^*, 0)$ fixed point of the $SU(2)$ theory without Higgs term.

The $SU(2)$ $2\Phi$ theory at the $(\lambda, \lambda') = (\lambda^*, 0)$ fixed point has a $SO(3) \times O(2)$ symmetry. The $SO(3)$ symmetry is also transparent in the $U(1)_1$ $2\psi$ theory. Lattice symmetry like $T_1, T_2, C_4$ (or $C_6$ on triangular/kagome lattice) are special elements in the $SO(3)$. In contrast, the $L_{U(1),\vec{m}}$ theory above does not have the $SO(3)$ symmetry and the lattice symmetry explicitly for a generic $\vec{m}$. So how does one understand the symmetry in the $U(1)$ theory? The answer is that the $SO(3)$ symmetry needs to act non-locally in the sense that it leaves the partition function invariant but not the action. It needs to transform the field $\Phi_{\vec{m}}$ in $L_{U(1),\vec{m}}$ to another field $\Phi_{\mathbf{R}\cdot\vec{m}} = U\Phi_{\vec{m}}$ in a different theory $L_{U(1),\mathbf{R}\cdot\vec{m}}$, where $\mathbf{R} \in SO(3)$ is generated accordingly from $U \in SU(2)$.[65] In the low energy regime of the $U(1)$ theory, one uses the $CP^1$ field variable $(\varphi_{1;\vec{m}}, \varphi_{2;\vec{m}}^*)$ for each $\vec{m}$, which are obtained by projecting to the two components of lowest energy from the term $-\Phi^\dagger\vec{m}\cdot\vec{\sigma}\Phi$. After an $SO(3)$ rotation, $(\varphi_{1;\vec{m}}, \varphi_{2;\vec{m}}^*)^T$ maps to $(\varphi_{1;\pm\mathbf{R}\cdot\vec{m}}, \varphi_{2;\pm\mathbf{R}\cdot\vec{m}}^*)^T$.

Let us illustrate this procedure using one example with $\vec{m} = (0,0,1)$ and consider the translation $T_1 : -i\sigma_1$ in the original $SU(2)$ theory. Now, it will first transform $\vec{m}$ to $-\vec{m}$ with $\Phi_{-\vec{m}} = -i\sigma_1\Phi_{\vec{m}}$. Then we map it back to $\vec{m}$ using $\Phi_{\vec{m}} = i\tau_1\Phi_{-\vec{m}} = \sigma_1\tau_1\Phi_{\vec{m}}$. For this particular example, the translation symmetry $T_1$ acts locally in the sense that it maps $\Phi$ to the same $U(1)$ theory specified by $\vec{m} = (0,0,1)$. Focusing on the low energy degree $(\varphi_1, \varphi_2^*) = (\Phi_{1;1}, \Phi_{2,2})$, $T_1$ acts simply as $\sigma_1$. In the similar way one can see that $T_2$ acts as $-i\sigma_2$, $\vec{m} = (0,0,1) \to \vec{m} = (0,0,1)$. However, consider a different $U(1)$ theory labeled by $\vec{m} = (1,1,0)$, then $T_1$ needs to act non locally and it maps $(\varphi_1, \varphi_2^*)$ to the field of a different $U(1)$ theory labeled by $\vec{m}' = (1,-1,0)$.

On square lattice, all of the lattice symmetries map $\vec{m} = (0,0,1)$ to $\vec{m} = (0,0,1)$. Therefore for the $U(1)$ theory with $\vec{m} = (0,0,1)$, lattice symmetries act locally. Thus the $U(1)$ theory can be regularized by a lattice model, as derived from the parton mean field theory in Sec. VI. A generic $SO(3)$ rotation still acts non-locally, but it does not correspond to a microscopic lattice symmetry. In contrast, on triangular lattice, the $C_6$ symmetry transforms $\vec{m} = (0,0,1)$ to $\vec{m} = (1,0,0)$ and then to $\vec{m} = (0,1,0)$, following the rule in Table. II, since $C_6$ relates $(n_3, n_4, n_5)$ to each other. The $\varphi^\dagger\sigma_z\varphi$ in the $U(1)$ theory with these three different $\vec{m}$ correspond to these three CDW orders. As a result, $C_6$ needs to act non-locally in the $U(1)$ theory with a generic $\vec{m}$, unless $\vec{m} = \frac{1}{\sqrt{3}}(1,1,1)$. But $T_1$ needs to act non-locally for $\vec{m} = \frac{1}{\sqrt{3}}(1,1,1)$. Therefore, on triangular/kagome lattice, one can not find any $\vec{m}$ such that the microscopic lattice symmetries can act locally. This is consistent with the observation that there is no type I $U(1)$ ansatz on triangular/kagome lattice discussed in Sec. IV. It is impossible to derive the $U(1)$ $2\varphi$ theory for the CSL-SC transition starting from a parton mean field construction on triangular/kagome lattice, because otherwise the lattice symmetry should act locally in the resulting theory.

A similar discussion can be made for the CSL-CDW$_{xy}$ transition following the $\lambda \leftrightarrow \lambda'$ duality. At the fixed

point $(\lambda, \lambda') = (0, \lambda_+^*)$, we expect the duality $\mathcal{L}_{SU(2)} \leftrightarrow \mathcal{L}_{SU(2)} - h\vec{m} \cdot (\mathrm{Re}\Phi^T \sigma_1\tau_1\Phi, \mathrm{Im}\Phi^T \sigma_1\tau_1\Phi, \Phi^\dagger \tau_3\Phi)$, which again leads to a manifold of $U(1)_{-2}$ $2\varphi$ theory specified by $\vec{m}$. The symmetry action is different. For example, the $\varphi^\dagger \sigma_z \varphi$ operator in the resulting $U(1)$ theory correspond to the order parameter $(n_1, n_2, n_5)$ now for $\vec{m} = (1,0,0), (0,1,0), (0,0,1)$ respectively. The $SO(3)$ symmetry still needs to act non-locally, mapping one $\vec{m}$ to a different $\vec{m}$. However, in this case all lattice symmetries on square, triangular and kagome lattice act locally for $\vec{m} = (0,0,1)$, since microscopically the SC order $(n_1, n_2)$ are very different from $\mathrm{CDW}_z$ order $n_5$ and no microscopic symmetry can rotate $n_5$ to mix with $(n_1, n_2)$. Hence the $U(1)$ $2\varphi$ theory for the CSL-$\mathrm{CDW}_{xy}$ transition can be derived from parton mean field theory on square, triangular and kagome lattice.

## XI.  DQCP BETWEEN CDW AND SC

We have discussed CSL-SC and CSL-CDW transitions on square and triangular/kagome lattice. It is then interesting to ask whether there can be a direct transition between SC and CDW phase. Both phases are symmetry breaking phases without fractionalization, so a direct transition needs to be beyond Landau framework although nearby phases are conventional phases. In this section we show that the SC-$\mathrm{CDW}_{xy}$ on square lattice and SC-CDW transition on triangular lattice are in the same universality class as the DQCP of easy plane or isotropic Neel to VBS transition[1]. The key is that the topological superconductor can be understood as from condensation of skyrmion. The $\mathrm{CDW}_{xy}$ or CDW order live on the manifold $S^1$ or $S^2$ and the CDW phase is a Chern insulator with $C = 2$. Similar to the quantum Hall ferromagnetism, the skyrmion defect of the $S^2$ order carries physical charge $2e$ and can be identified as a bosonic Cooper pair. Condensation of these skyrmions will disorder the CDW order and lead to a superconductor at the same time.

Skyrmion superconductor has been discussed from a quantum spin Hall insulator (QSHI)[3,4]. There the resulting superconductor is topologically trivial. In contrast, the skyrmion superconductor in our case still inherits the chiral central charge $c = 2$ from the $C = 2$ Chern insulator and is thus topologically equivalent to a $d + id$ superconductor. Nevertheless the CDW-SC transition in the bulk is the same as the QSHI-SC transition if we ignore the edge physics. Skyrmion superconductor was also proposed from a spin polarized Chern insulator with $C = 2$ in moiré systems[8]. The physics is similar to our case and the skyrmion superconductor there, if possible, should also be topological with chiral central charge $c = 2$.

Technically, the easiest way to describe this transition is to use $\mathrm{CP}^1$ representation of the $SO(3)$ CDW order parameter. To obtain the topological superconductor, the $\mathrm{CP}^1$ boson needs to be in a bosonic integer quantum Hall insulator phase, instead of a trivial insulator. We

start from the type II $U(1)$ ansatz and the low energy boson field is $\varphi = (\varphi_1, \varphi_2)^T$ with $\varphi^\dagger \vec{\sigma}\varphi$ represents the CDW order. Following the discussion in Sec. VII, we let these bosons enter a bIQHE phase to provide a term $-\frac{2}{4\pi}ada$ term, which cancels the Chern-Simons term from the fermionic spinons. Finally there is no Chern-Simons term for $a_\mu$ anymore. Now one has gapped bosonic holon $\varphi$ excitations and the ground state is a Superconductor whose order parameter is the monopole of the gauge field $a$. Then consider a transition between the bIQHE phase and the superfluid phase for the boson $\varphi$, which leads to the SC to CDW transition for the physical system. The superfluid transition from bIQHE for bosons is the same as that from a trivial insulator and is simply captured by condensation of $\varphi$, so the SC-CDW transition is described by the following critical theory:

$$\mathcal{L} = |(\partial_\mu - ia_\mu - i\frac{1}{2}\sigma_3 A_{r;\mu})\varphi|^2 - \frac{1}{2\pi}A_c da$$
$$- r|\varphi|^2 - g(|\varphi|^2)^2 + \lambda|\varphi_1|^2|\varphi_2|^2 \qquad (69)$$

When $r > 0$, $\varphi$ is gapped and one obtains the SC phase. When $r < 0$ and $\lambda > 0$, $\varphi$ condenses with easy-plane anisotropy, describing the Chern insulator phase with $\mathrm{CDW}_{xy}$ order. This is exactly the same critical theory for the easy plane DQCP between Neel and VBS order on square lattice[1]. If $\lambda = 0$, then this is the isotropic DQCP. When $\lambda > 0$, naively the above action can describe the transition between SC and $\mathrm{CDW}_z$, but this transition should be first order. The above theory also has a self duality symmetry, which will be discussed in the next section.

## XII.  TRI-CRITICAL POINT AND SELF-DUALITY

We have provided critical theories for CSL-SC, CSL-CDW and SC-CDW transitions. This section discusses the tri-critical point at the intersection of these three phases. For square lattice we only consider the $\mathrm{CDW}_{xy}$ order. For triangular/kagome lattice we consider the isotropic CDW order with $SO(3)$ symmetry. The tri-critical points on square and triangular lattice correspond to the pink and red fixed points in Fig. 6 for the $SU(2)$ theory. There is a self-duality symmetry from the $\lambda \leftrightarrow \lambda'$ transformation in the $SU(2)$ theory. Here we provide alternative theories with two $U(1)$ gauge fields coupled to both $2\varphi$ and $2\psi$.

We discuss the CSL-$\mathrm{CDW}_{xy}$-SC tri-critical point on square lattice first. We start from the type II $U(1)$ ansatz for the CSL phase. As discussed previously, when the holons go through a plateau transition into IQHE, the system goes into an SC phase. While if the holons simply condense with $\langle\varphi\rangle \propto (1,1)^T$, the condensation pattern breaks lattice symmetry and results in a $\mathrm{CDW}_{xy}$ state. We see there are 2 different tuning parameters controlling transition into either SC or CDW. Hence we arrive at a tri-critical point among CSL, CDW and SC. To formulate

the tricritical theory, we use two Dirac fermions to describe the CSL-SC transition as in Eq. 45. Meanwhile we also keep track the bosonic holons $\varphi$ and consider the mass of $\varphi$ as another tuning parameter.[66] Putting them together, the following critical theory emerges:

$$\mathcal{L} = \bar{\psi}(-i\gamma_\mu\partial_\mu - b_\mu\gamma_\mu + \frac{1}{2}A_{r;\mu}\sigma_3\gamma_\mu)\psi + m\bar{\psi}_i\psi_i - \frac{1}{4\pi}(\frac{1}{2}A_c + a)d(\frac{1}{2}A_c + a) + \frac{1}{2\pi}bd(\frac{1}{2}A_c + a) + \frac{1}{16\pi}A_r dA_r$$

$$+ |(\partial_\mu - ia_\mu - i\frac{1}{2}\sigma_0 A_{c;\mu} - i\frac{1}{2}\sigma_3 A_{r;\mu})\varphi|^2 - r|\varphi|^2 - g(|\varphi|^2)^2 + \lambda|\varphi_1|^2|\varphi_2|^2$$

$$+ \frac{2}{4\pi}ada - \frac{1}{8\pi}A_c dA_c - \mathcal{L}_{int} \tag{70}$$

where the first line describes the plateau transition of the bosonic holon and the second line the condensation of the holon. The third line is from the integration of the fermionic spinons and the stacking of the $\nu = -2$ IQHE phase. Interactions between $\varphi$ and $\psi$ $\mathcal{L}_{int}$ will be specified in the next equation.

With a simplification $a_\mu \rightarrow a_\mu - \frac{1}{2}A_{c;\mu}$, the tri-critical point theory is cast into:

$$\mathcal{L} = \bar{\psi}(-i\gamma_\mu\partial_\mu - b_\mu\gamma_\mu + \frac{1}{2}A_{r;\mu}\sigma_3\gamma_\mu)\psi + m\bar{\psi}_i\psi_i + |(\partial_\mu - ia_\mu - i\frac{1}{2}\sigma_3 A_{r;\mu})\varphi|^2 - r|\varphi|^2 - g(|\varphi|^2)^2 + \lambda|\varphi_1|^2|\varphi_2|^2$$

$$+ \frac{1}{4\pi}ada + \frac{1}{2\pi}(b - A_c)da + \frac{1}{16\pi}A_r dA_r - g_0(\bar{\psi}\psi)(\varphi^\dagger\varphi) - \tilde{g}(\bar{\psi}\vec{\sigma}\psi)\cdot(\varphi^\dagger\vec{\sigma}\varphi) - \tilde{\lambda}(\bar{\psi}\sigma_3\psi)(\varphi^\dagger\sigma_3\varphi) \tag{71}$$

where $\varphi = (\varphi_1, \varphi_2)^T$ and $\psi = (\psi_1, \psi_2)^T$.

We have also included the interaction terms between $\psi$ and $\varphi$. $\lambda$ and $\tilde{\lambda}$ are easy-plane anisotropy terms. When $m < 0$, one can integrate $\psi$ and then integrate $b_\mu$, after which we recover Eq. 56 for the CSL-CDW$_{xy}$ transition. When $r > 0$, $\varphi$ is gapped and one can integrate $a_\mu$ and get the $U(1)_1$ $2\psi$ theory for the CSL-SC transition in Eq. 52. In summary, $m < 0, r > 0$ is the CSL phase, $m > 0, r > 0$ is the SC phase. $r < 0$ is the CDW phase regardless of the sign of $m$. $r = m = 0$ corresponds to the tri-critical point. These are summarized in Fig. 7(b). When $\lambda = \tilde{\lambda} = 0$, this is the CSL-SC-CDW tri-critical point on triangular lattice, shown in Fig. 7(a). When $\lambda > 0, \tilde{\lambda} > 0$, this is the CSL-SC-CDW$_{xy}$ tri-critical point. In this theory, $\varphi^\dagger\vec{\sigma}\varphi \sim \bar{\psi}\vec{\sigma}\psi$ correspond to the CDW order $(n_3, n_4, n_5)$.

In the $SU(2)$ theory, these two tri-critical points have a self-duality which exchanges $A_c$ and $A_r$. Thus we believe the above tri-critical theory is also self dual to itself except the exchange between $A_c$ and $A_r$. One can derive this self-duality in the following way on square lattice. On square lattice, starting from the type I $U(1)$ ansatz, then the CSL-SC transition is described by the $U(1)_{-2}$ $2\varphi$ theory. Alternatively one could let the holon $\varphi$ goes through a plateau transition and get a CDW$_{xy}$ order. Putting them together, the following tri-critical theory reads:

$$\mathcal{L} = \bar{\psi}(-i\gamma_\mu\partial_\mu - b_\mu\gamma_\mu - \frac{1}{2}A_{c;\mu}\sigma_3\gamma_\mu)\psi + m\bar{\psi}_i\psi_i + |(\partial_\mu - ia_\mu - i\frac{1}{2}\sigma_3 A_{c;\mu})\varphi|^2 - r|\varphi|^2 - g(|\varphi|^2)^2 + \lambda|\varphi_1|^2|\varphi_2|^2$$

$$+ \frac{1}{4\pi}ada + \frac{1}{2\pi}(b - A_r)da + \frac{1}{8\pi}A_r dA_r - \frac{1}{16\pi}A_c dA_c - g_0(\bar{\psi}\psi)(\varphi^\dagger\varphi) - \tilde{g}(\bar{\psi}\vec{\sigma}\psi)\cdot(\varphi^\dagger\vec{\sigma}\varphi) - \tilde{\lambda}(\bar{\psi}\sigma_3\psi)(\varphi^\dagger\sigma_3\varphi) \tag{72}$$

where $\varphi = (\varphi_1, \varphi_2)^T$ and $\psi = (\psi_1, \psi_2)^T$.

When $r > 0$ and $\varphi$ is gapped, we recover Eq. 58 after integrating $a$. When $m < 0$, we recover Eq. 40 after integrating $\psi$ and $b$. $m < 0, r > 0$ is the CSL phase. $m > 0, r > 0$ is the CDW$_{xy}$ order. $r < 0$ is the SC phase regardless of the sign of $m$. $m = r = 0$ is the tri-critical point. This time $\varphi^\dagger\vec{\sigma}\varphi \sim \bar{\psi}\vec{\sigma}\psi$ correspond to $(n_1, n_2, n_5)$. In the above we assume the easy plane anisotropy terms $\lambda, \tilde{\lambda}$ so the tri-critical point is between CSL-SC-CDW$_{xy}$. We believe the $\lambda = \tilde{\lambda} = 0$ corresponds to the CSL-SC-CDW tri-critical point on triangular/kagome lattice, though now $\varphi$ transforms non-locally under lattice symmetry. This means that both tri-critical points on square and triangular/kagome lattice have the self-duality at $r = m = 0$ of Eq. 71 and Eq. 72.

Both theories only have explicit $U(1)_c \times U(1)_r \rtimes Z_2$ symmetry with $Z_2$ coming from either translation $T_1$ or $T_2$. In Eq. 71, one can see the mixed anomaly that $dA_c = 2\pi$ carries charge $1/2$ under $A_r$ because one needs to attach $\varphi$ to cancel the charge under $a$. Similarly $dA_r = 2\pi$ carries charge $1/2$ under $A_c$. This mixed anomaly is crucial for the DQCP between SC and CDW and it already exits at the tri-critical point. The self-duality relates order parameter $(n_1, n_2)$ to $(n_3, n_4)$ and suggests an enlarged $O(4)$ symmetry when there is easy-plane anisotropy terms $\lambda, \lambda'$. When $\lambda = \lambda' = 0$, there is explicit SO(3)$\times$ O(2) symmetry in either theory. In Eq. 71 the SO(3) rotates $(n_3, n_4, n_5)$, while in Eq. 72 the SO(3) rotates $(n_1, n_2, n_5)$. The self-duality then implies a $SO(5)$ symmetry. Nevertheless, $O(4)$ and $SO(5)$ symmetries are not explicit in the above form. To explicitly

see the symmetry, we still need to use the $SU(2)$ theory.

We plot phase diagrams for Eq. 71 and Eq. 72 in Fig. 7. The tri-critical critical theories here are different from that of the $SU(2)$ theory in Sec. IX. Let us take the square lattice as an example. In the $SU(2)$ theory, bosonic holons are in either trivial insulator and superfluid phases. SC and CDW correspond to different condensation patterns of the superfluid phase of the bosonic holon $\Phi$. In contrast, here we start from the $U(1)$ ansatz of the CSL. If we start from the type I $U(1)$ ansatz, then CSL, SC and CDW$_{xy}$ correspond to trivial insulator, superfluid and bIQHE insulator of the bosonic holons $\varphi$. If we start from type II $U(1)$ ansatz, CSL, CDW$_{xy}$ and SC correspond to trivial insulator, superfluid and bIQHE insulator of bosonic holon $\varphi$. One can clearly see the duality between type I and type II $U(1)$ with exchange of SC and CDW$_{xy}$. For triangular and kagome lattice, we only have type II ansatz and can only derive Eq. 71 from parton construction. But we believe it has a dual theory as Eq. 72, just the lattice symmetry needs to act non-locally as discussed in Sec. X.

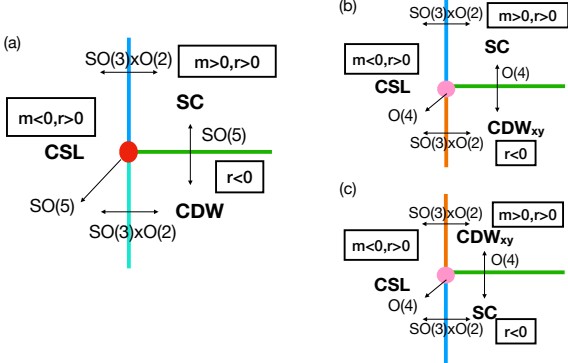

FIG. 7: The phase diagram summarizing the transitions between CSL,d+id SC and CDW Chern insulator on triangular (a) and square lattices (b,c). (b,c) are described by Eq (71) and Eq (72),respectively.

## XIII.   HONEYCOMB LATTICE: $U(1)$ $1\varphi$ THEORY AND ABSENCE OF SYMMETRY BREAKING ORDER

We have shown various critical theories and proximate phases nearby a chiral spin liquid phase on square, triangular and kagome lattice. In contrast, on honeycomb lattice, we do not expect a direct CSL-SC transition because there are two electrons per unit cell in the CSL phase of a Mott insulator. On honeycomb lattice, the natural mean field ansatz of the CSL phase is the Hal-

dane model with $T_1 T_2 = T_2 T_1$. As a result there is only one single bosonic holon mode $\varphi$, which leads to a critical theory:

$$\mathcal{L} = |(\partial_\mu - ia_\mu)\varphi|^2 - r|\varphi|^2 - g|\varphi|^4 + \frac{2}{4\pi}ada - \frac{1}{2\pi}A_c da \quad (73)$$

In this case the $r < 0$ side is a symmetry Chern insulator with $C = 2$, simply described by the Haldane model. The critical theory is known to be dual to a $U(1)$ theory with one Dirac fermion, an $SU(2)$ theory with one boson and an $SU(2)$ theory with one Dirac fermion[67,68]. There is also an emergent $SO(3)$ symmetry which rotates $(\nabla \times \mathbf{a}, \text{Re}(M\varphi^*), \text{Im}(\mathcal{M}\varphi^*))$. They represent the density fluctuation and the Cooper pair creation operators which transform trivially under lattice symmetries. Gapless charge mode is expected at the QCP. There is no other symmetry breaking order parameter fluctuating at the QCP. One can clearly see that the difference of the square, triangular and kagome lattice arises from the projective translation symmetry of the semion, which guarantees the existence of two bosonic modes in the $U(1)$ $2\varphi$ theory.

## XIV.   EXPERIMENTAL SIGNATURES

We discuss the possible experimental realization and signatures of the CSL-SC transition. Recently CSL was observed numerically in triangular lattice Hubbard model in the intermediate regime of U/t[32]. The next phase close to the CSL may be a superconductor[69] and a CSL-SC transition as described by our paper can be naturally realized by reducing $U/t$. Bandwidth tuned metal insulator transition has recently been observed in moiré superlattice based on transition metal dichalcogenide(TMD)[44]. Although a CSL phase was not reported in the current experiment, it may be found at lower temperature or a different parameter regime. Therefore it may be interesting to search for CSL-SC transition in moiré materials. Here we will provide experimental predictions in the critical regime, focusing on the transport at the CSL-SC critical point. One can access the conductivity tensor from the $U(1)$ $2\varphi$ theory, $U(1)$ $2\psi$ theory or $SU(2)$ $2\Phi$ theory, and we mainly use the two $U(1)$ theories.

First, let us start with the $U(1)$ $2\varphi$ theory in Eq. 40. We will ignore $A_r$ as the transport measured in the experiments is associated only with $A_c$. In the original action a term $-\frac{1}{8\pi}A_c dA_c$ was added coming from the stacking of $\nu = -2$ IQHE phase, which will be ignored because the $\nu = -2$ IQHE phase is stacked only for simplicity and does not really exist. $\varphi_1$ couples to $a + \frac{1}{2}A$ and $\varphi_2$ couples to $a - \frac{1}{2}A$. Translation symmetry acts as $T_1 : \varphi_1 \to \varphi_2^*, \varphi_2 \to \varphi_1^*$. Integrating $\varphi_1, \varphi_2$, we get:

$$\mathcal{L}_{eff} = \mathcal{L}_a + \frac{1}{2}\sum_{\omega,\mathbf{q}}\frac{1}{4}(A_x(\omega,\mathbf{q}), A_y(\omega,\mathbf{q}))(\Pi_{11}(\omega,\mathbf{q}) + \Pi_{22}(\omega,\mathbf{q}) - \Pi_{12}(\omega,\mathbf{q}) - \Pi_{21}(\omega,\mathbf{q}))\begin{pmatrix}A_x(-\omega,-\mathbf{q})\\A_y(-\omega,-\mathbf{q})\end{pmatrix} \quad (74)$$

where $\Pi_{11}$ and $\Pi_{22}$ are from the conductivity tensor of $\varphi_1$ and $\varphi_2$. $\Pi_{12}, \Pi_{21}$ encode the drag conductivity between $\varphi_1$ and $\varphi_2$. Note that under $T_1$, we have $\mathbf{A} \rightarrow \mathbf{A}, \mathbf{a} \rightarrow -\mathbf{a}$, so that there is no crossing term between $A$ and $a$. $\mathcal{L}_a$ includes the terms for $a_\mu$, which is not interesting for physical transport.

We simply define $\Pi(\omega, \mathbf{q}) = \frac{1}{4}(\Pi_{11}(\omega, \mathbf{q}) + \Pi_{22}(\omega, \mathbf{q}) - \Pi_{12}(\omega, \mathbf{q}) - \Pi_{21}(\omega, \mathbf{q}))$. It encodes the conductivity tensor of the physical electron:

$$\Pi(\omega, \mathbf{q}) = \frac{e^2}{\hbar} \frac{1}{2\pi} (-i\omega) \begin{pmatrix} \sigma_0 & \sigma_{xy} \\ -\sigma_{xy} & \sigma_0 \end{pmatrix} \qquad (75)$$

where $\mathbf{A}^T = (A_x, A_y)$ and $\mathbf{b}^T = (b_x, b_y)$. $I$ and $\epsilon$ are $2 \times 2$ matrix. $I$ is the identity matrix. $\epsilon = \begin{pmatrix} 0 & 1 \\ -1 & 0 \end{pmatrix}$.

Integration of $b_\mu$ leads to

$$\mathcal{L}_{eff} = \frac{1}{2} \sum_{\omega, \mathbf{q}} (A_x(\omega, \mathbf{q}), A_y(\omega, \mathbf{q})) \frac{-i\omega}{2\pi} \sigma(\omega, \mathbf{q}) \begin{pmatrix} A_x(-\omega, -\mathbf{q}) \\ A_y(-\omega, -\mathbf{q}) \end{pmatrix} \qquad (77)$$

with

$$\sigma(\omega, \mathbf{q}) = \left(\sigma_\psi I + (\sigma_\psi^{xy} - 1)\epsilon\right)^{-1} - \frac{1}{2}\epsilon$$
$$= \frac{\sigma_\psi}{\sigma_\psi^2 + (1 - \sigma_\psi^{xy})^2} I + \left(\frac{1 - \sigma_\psi^{xy}}{\sigma_\psi^2 + (1 - \sigma_\psi^{xy})^2} - \frac{1}{2}\right)\epsilon \qquad (78)$$

Compared to Eq. 75, we have the following constraint for the two dual theories, where similar results are obtained in ref[70]:

$$\sigma_0 = \frac{\sigma_\psi}{\sigma_\psi^2 + (1 - \sigma_\psi^{xy})^2}$$
$$\sigma_{xy} = \frac{1 - \sigma_\psi^{xy}}{\sigma_\psi^2 + (1 - \sigma_\psi^{xy})^2} - \frac{1}{2} \qquad (79)$$

We have argued that $\sigma_{xy}$ should be very small from the $U(1)$ $2\varphi$ theory. This will impose non-trivial constraint for the theory with Dirac fermion.

In addition to a universal conductivity, there also should be quasi long range fluctuation of the CDW order at the CSL-SC critical point. The CDW fluctuation can persist to the superconductor phase and the CDW can be stabilized in the vortex core of the superconductor phase. This can be tested by X-ray scattering and scanning tunneling microscope (STM) experiments.

We have restricted to the transition tuned by bandwidth with the density fixed at integer filling. For the

$\sigma_0$ and $\sigma_{xy}$ are universal numbers. $\sigma_0$ is the universal conductivity usually present in 2+1 d CFT for bosons. There is no symmetry to forbid the Hall conductivity $\sigma_{xy}$. However, note that $\langle da \rangle = 0$ on average and it does not couple to the physical gauge field $A_\mu$ due to symmetry $T_1$. So we expect that $\sigma_{xy}$ should be very small even if it exist.

We can also derive the conductivity tensor from the $U(1)$ theory with two Dirac fermions in Eq. 52. Again we ignore $A_r$ and add back a term $\frac{1}{8\pi}A_c dA_c$. Integration of $\psi$ leads to

$$\mathcal{L}_{eff} = \frac{1}{2}\mathbf{b}^T(\omega, \mathbf{q})\frac{-i\omega}{2\pi}\left(\sigma_\psi I + (\sigma_\psi^{xy} - 1)\epsilon\right)\mathbf{b}(-\omega, -\mathbf{q}) - \frac{1}{4}\mathbf{A}^T(\omega, \mathbf{q})\frac{-i\omega}{2\pi}\epsilon\mathbf{A}(-\omega, -\mathbf{q})$$
$$+ \frac{1}{2}\mathbf{A}^T(\omega, \mathbf{q})\frac{-i\omega}{2\pi}\epsilon\mathbf{b}(-\omega, -\mathbf{q}) + \frac{1}{2}\mathbf{b}^T(\omega, \mathbf{q})\frac{-i\omega}{2\pi}\epsilon\mathbf{A}(-\omega, -\mathbf{q}) \qquad (76)$$

chemical potential tuned transition, we can still use the $SU(2)$ theory in Sec. IX. But now there is a term $i\Phi^* \partial_t \Phi$ and we have $z = 2$. The fixed point should still be decided by the quartic terms $(\lambda, \lambda')$. On triangular lattice, the CSL-SC transition is still fixed at the $\lambda' = 0$ axis by the $C_6$ symmetry. If we assume that the chemical potential tuned CSL-SC transition is the same on square and triangular lattice, then the transition on square lattice is at the same fixed point and still has SO(3)×O(2) symmetry. The $\lambda \leftrightarrow \lambda'$ duality is still there and the structure of the fixed points should remain the same as the $z = 1$ case. We should still have a dual $U(1)$ theory with $2\varphi$ by adding Higgs terms to the $SU(2)$ theory, though now it is also $z = 2$. There is no obvious theory with Dirac fermion. We no longer expect a universal conductivity, but the intertwinement of the SC and CDW order in the critical regime should still exist.

## XV. CONCLUSION

In summary we present critical theories for transitions between chiral spin liquid, topological superconductor and CDW Chern insulator. In the CSL to SC transition, the are also CDW orders transforming under an emergent $SO(3)$ symmetry. Such an intertwinement of an additional symmetry breaking order is guaranteed by the LSM theorem at odd electron filling per unit cell. We present the critical theories in three forms: $U(1)$ theory with two bosons, $U(1)$ theory with two Dirac fermions and $SU(2)$ theory with two bosons. Our work demonstrates the duality between these three theories and possible experimental realizations of these interesting CFTs. In the $SU(2)$ theory, there are several fixed points decided by the quartic terms $\lambda, \lambda'$, corresponding to bi-critical and tri-critical points with $SO(5), O(4)$ or $SO(3) \times O(2)$ global symmetry. There is also a duality transformation $\lambda \leftrightarrow \lambda'$ which exchanges the easy-plane CDW order and the SC order. We offer a new perspec-

tive to understand the $d + id$ superconductor as from skyrmion condensation of the CDW order. The CDW-SC transitions are in the same universality classes as the usual Neel to VBS DQCP, but now both of them are proximate to a CSL phase. The CDW-SC DQCP theories, along with the enlarged symmetry and self-duality, are simply descendants of unified tri-critical theories. We also discuss possible experimental realizations and detection of the CSL-SC transition.

## XVI.   ACKNOWLEDGEMENT

We thank Ashvin Vishwanath, Cenke Xu and Maissam Barkeshli for useful discussions. X-Y Song thanks Chong Wang and Max Metlitski for helpful discussions. YHZ is supported by a startup grant from Johns Hopkins University. X-Y S is supported by the Gordon and Betty Moore Foundation EPiQS Initiative through Grant No. GBMF8684 at the MIT.

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

## Appendix A: $SU(2), U(1)$ mean-field states for holons and symmetries

Here we list the solution for the $SU(2)$ mean-field ansatz for holons and the projective symmetry group of the holons.

### 1. Square lattice

The unit cell for the mean-field for holons eq (14) is enlarged to contain 2 sites $A = (0,0), B = (1,0)$ connected by a horizontal bond. It translates into $k$ space form in terms of $Z = (z_{A,1}, z_{B,1})^T$:

$$H_k = \begin{pmatrix} \sin k_2 & -\sin k_1 + i\eta(\cos(k_1+k_2) + \cos(k_2-k_1)) \\ -\sin k_1 - i\eta(\cos(k_1+k_2) + \cos(k_2-k_1)) & -\sin k_2 \end{pmatrix}. \tag{A1}$$

The Dirac nodes at $(0,0(\pi))$ are gapped out by a mass of order $2\eta$. For $z_2$ it differs by a minus sign.

The holon dynamics at low-energy is dominated by the fluctuations around the lowest energy states of $H_k$, located at $Q_{1(2)} = (\pi/2, \pm\pi/2)$. The eigenvectors for $z_1, z_2$ are

$$Q_1 : (z_{1,A}, z_{1,B}) = (\cos\frac{\pi}{8}, -\sin\frac{\pi}{8}), (z_{2,A}, z_{2,B}) = (\sin\frac{\pi}{8}, \cos\frac{\pi}{8}),$$
$$Q_2 : (z_{1,A}, z_{1,B}) = (-\sin\frac{\pi}{8}, \cos\frac{\pi}{8}), (z_{2,A}, z_{2,B}) = (\cos\frac{\pi}{8}, \sin\frac{\pi}{8}) \tag{A2}$$

at $Q_{1,2}$, respectively.

The symmetry actions on spinons and holons are projective, albeit the composite action for electrons is a faithful representation. The actions on spinons read:

$$T_{1,2} : (f_{r,\uparrow}^\dagger, f_{r,\downarrow})^T \to i\epsilon_r \tau^2 (f_{r+\hat{r}1(2),\uparrow}^\dagger, f_{r+\hat{r}1(2),\downarrow})^T,$$
$$C_4 : (f_{r,\uparrow}^\dagger, f_{r,\downarrow})^T \to i\epsilon_r \tau^2 (f_{C(r),\uparrow}^\dagger, f_{C(r)),\downarrow})^T,$$
$$R_1\mathcal{T} : (f_{r,\uparrow}^\dagger, f_{r,\downarrow})^T \to \mathcal{K} i\epsilon_r \tau^2 (f_{R_1(r),\downarrow}^\dagger, f_{R_1(r)),\uparrow})^T, \tag{A3}$$

where $\mathcal{K}$ denoting anti-unitary operations and $\tau^{1,2,3}$ matrices are Pauli matrices acting on the spinor indices. The transformation for holons follows from those of spinons by requiring the electron operators transform faithfully under the symmetries, i.e. $\Psi_{r,f} \to g_R(r)\Psi_{R(r),f}, Z_{r,b} \to g_R(r)^{-1}Z_{R(r),b}$ for a symmetry operation $R$ with a site-dependent gauge transform $g_R(r)$.

The lattice translation along $r_1$ direction $T_1$ gives a momentum boost of $(0,\pi)$, i.e.

$$T_1 : z_{1(2),r} \to i(-1)^{r_2} z_{1(2),r+\widehat{r}_1}$$
$$(\Phi_1, \Phi_2)^T \to -i\sigma_1(\Phi_1, \Phi_2)^T \tag{A4}$$

where we use $\sigma_a$ to label the Pauli matrix that rotates $\Phi = (\Phi_1, \Phi_2)$. Translation along $r_2$ just sends $z_{1(2),r} \to z_{1(2),r+\widehat{r}_2}$ and at low-energy:

$$T_1 : (\Phi_1, \Phi_2)^T \to -i\sigma_1(\Phi_1, \Phi_2)^T,$$
$$T_2 : (\Phi_1, \Phi_2)^T \to -i\sigma_3(\Phi_1, \Phi_2)^T, \tag{A5}$$

For $C_4$ rotation around a site e.g. at $B$ sublattice with coordinate $(1,0)$, one choice of transform that leave the ansatz invariant reads

$$z_{1,r} \to g(C4(r))z_{1,C4(r)}, g(r) = \begin{cases} -(-1)^{r_1} & if \mod (r_2, 2) = 0 \\ 1 & if \mod (r_2, 2) = 1 \end{cases}. \tag{A6}$$

The transform for $z_2^*$ differs by an overall minus sign to make the rotation of $\Phi$ free of $SU(2)$ gauge transform.

It sends the low energy field as:

$$C_4 : (\Phi_1, \Phi_2))^T \to -\sigma_3 e^{i\frac{\pi}{4}\sigma_2}(\Phi_1, \Phi_2))^T. \tag{A7}$$

For $R_1\mathcal{T}, R_2\mathcal{T}$, the holon fields are sent to (identical for $z_1, z_2$)

$$R_{1,2}\mathcal{T} : z_r \to (-1)^{r_{2,1}}\mathcal{K}z_{R_{1,2}T(r)}, (\Phi_1, \Phi_2) \to -(\Phi_1, \Phi_2). \tag{A8}$$

To simplify our notation, we make a redefinition $(\Phi_1, \Phi_2)^T \to e^{-i\sigma_1\pi/4}(\Phi_1, \Phi_2)^T$, so that the symmetry transformation now is: $T_1 : -i\sigma_1$, $T_2 : i\sigma_2$, $C_4 : \sigma_2 e^{i\frac{\pi}{4}\sigma_3}$, $R_{1,2}\mathcal{T} : i\sigma_1$.

## 2. Triangular lattices

As a starting point, we consider the $U(1)$ CSL mean field ansatz. We define the coordinate to be $\mathbf{r} = x\mathbf{a_1} + y\mathbf{a_2}$. $\mathbf{a_1}$ is along x direction, $\mathbf{a_2}$ is along $120°$ direction.

$$H_\Psi = t_f \Psi^\dagger(\mathbf{r} + \widehat{r}_1)ie^{i(\frac{\pi}{2}+\theta)\tau_3}\Psi(\mathbf{r}) + h.c.$$
$$+ t_f \Psi^\dagger(\mathbf{r} + \widehat{r}_2)(-1)^{r_1}ie^{i(\frac{\pi}{2}+\theta)\tau_3}\Psi(\mathbf{r}) + h.c.$$
$$+ t_f \Psi^\dagger(\mathbf{r} + \widehat{r}_1 + \widehat{r}_2)(-1)_1^r ie^{-i\theta\tau_3}\Psi(\mathbf{r}) + h.c. \tag{A9}$$

We then do a gauge transformation $\Psi(\mathbf{r}) \to e^{i\frac{\pi}{2}(r_1+r_2)\tau_3}\Psi(\mathbf{r})$, and get the mean-field eq (17).

The mean-field composes of 2-site unit cells denoted as sublattice $A, B$. In the basis $(Z_A(\mathbf{k}), Z_B(\mathbf{k}))$, we get

$$H_Z = (Z_A^\dagger(\mathbf{k}), Z_B^\dagger(\mathbf{k}))h(\mathbf{k})\begin{pmatrix} Z_A(\mathbf{k}) \\ Z_B(\mathbf{k}) \end{pmatrix} \tag{A10}$$

where

$$h(\mathbf{k}) = \begin{pmatrix} 2t_b \cos\theta \sin(-\frac{1}{2}k_x + \frac{\sqrt{3}}{2}k_y) & 2t_b \cos\theta \sin k_x - 2it_b \cos\theta \cos(\frac{1}{2}k_x + \frac{\sqrt{3}}{2}k_y) \\ 2t_b \cos\theta \sin k_x + 2it_b \cos\theta \cos(\frac{1}{2}k_x + \frac{\sqrt{3}}{2}k_y) & -2t_b \cos\theta \sin(-\frac{1}{2}k_x + \frac{\sqrt{3}}{2}k_y) \end{pmatrix}$$
$$+ \begin{pmatrix} -2t_b \sin\theta\tau_3 \cos(-\frac{1}{2}k_x + \frac{\sqrt{3}}{2}k_y) & -2t_b \sin\theta \cos k_x\tau_3 + 2it_b \sin\theta \sin(\frac{1}{2}k_x + \frac{\sqrt{3}}{2}k_y) \\ -2t_b \sin\theta \cos k_x\tau_3 - 2it_b \sin\theta \sin(\frac{1}{2}k_x + \frac{\sqrt{3}}{2}k_y) & 2t_b \sin\theta\tau_3 \cos(-\frac{1}{2}k_x + \frac{\sqrt{3}}{2}k_y) \end{pmatrix} \tag{A11}$$

When taking $\theta = 0$, the mean-field is invariant under $SU(2)$, with symmetry transformation:

$$T_1 : \ \Psi(\mathbf{r}) \to (-1)^{r_2} \Psi(\mathbf{r} + \widehat{r}_1)$$
$$T_2 : \ \Psi(\mathbf{r}) \to \Psi(\mathbf{r} + \widehat{r}_2)$$
$$C_6 : \ \Psi(\mathbf{r}) \to \begin{cases} i(i)^{r_2} \Psi(C_6(r)) & Mod(r_2, 2) = 1 \\ -(-1)^{r_1}(i)^{r_2} \Psi(C_6(r)) & Mod(r_2, 2) = 0 \end{cases}$$
$$R\mathcal{T} : \ \Psi(\vec{r}) \to \begin{cases} (-1)^{r_1} \Psi(R(r)) & Mod(r_2, 4) = 0, 1 \\ -(-1)^{r_1} \Psi(R(r)) & Mod(r_2, 4) = 2, 3 \end{cases} \tag{A12}$$

The holons hop in the same ansatz as the spinons, with the k space Hamiltonian reads,

$$H_Z(\mathbf{k}) = \sin k_2 \eta^3 - \sin k_1 \eta^1 - \cos(k_1 + k_2)\eta^2, \tag{A13}$$

where $k_{1,2}$ are coordinates in reciprocal space spanned by $\mathbf{b}_{1,2}$, that satisfies $\mathbf{b}_i \cdot \mathbf{a}_j = 2\pi \delta_{ij}$. $\eta$ rotates $A, B$ sublattices.

There are 2 holon minima at $Q_{1,2} = (\pi/2, \pm \pi/2)$, with states of minimal energy denoted as $\Phi = (\Phi_1, \Phi_2)$, at $Q_{1,2}$, respectively. They transform as

$$T_1 : \Phi \to -i\sigma_1 \Phi,$$
$$T_2 : \Phi \to -i\sigma_3 \Phi,$$
$$C_6 : \Phi \to e^{i\frac{\pi}{3}} e^{-i\frac{\sigma_3 + \sigma_2 + \sigma_1}{\sqrt{3}} \frac{\pi}{3}} \Phi,$$
$$R\mathcal{T} : \Phi \to \frac{e^{-i\frac{\pi}{12}}}{\sqrt{2}}(1 - i\sigma_1)\Phi \tag{A14}$$

where $\sigma$ rotates two valleys $Q_{1,2}$.

For the $U(1)$ ansatz on triangular lattices discussed in section IV, the projective symmetry group for the spinons reads:

$$T_1 : \psi_r \to (-1)^{r_2} \psi_{r+\widehat{r}_1},$$
$$T_2 : \psi_r \to \psi_{r+\widehat{r}_2},$$
$$C_6 : \psi_r \to \begin{cases} \tau_1 \psi_{C_6 r} & (C_6 r)_2 \mod 2 = 0 \\ -(-1)^{(C_6 r)_x} i\tau_1 \psi_{C_6 r} & (C_6 r)_2 \mod 2 = 1 \end{cases}, \tag{A15}$$

those for the holons in terms of $(z_1, z_2^*)^T$ are the same for the spinon symmetry transforms.

There are 2 holon minima at $Q_{1,2} = (\pi/2, \pm \pi/2)$ for each holon species, respectively[46]. The flux is translation invariant but breaks naive $C_6$ rotation defined by eq (A14). Note compared to $SU(2)$ invariant ansatz, the additional hopping for small $\theta$ projects to the lowest-energy holon states to be 0.

Denote the states of minimal energy as $\Phi = (\Phi_1, \Phi_2)$, at $Q_{1,2}$. They transform as

$$T_1 : \Phi \to -i\sigma_1 \Phi,$$
$$T_2 : \Phi \to -i\sigma_3 \Phi,$$
$$C_6 : \Phi \to e^{i\frac{\pi}{3}} \tau_1 e^{-i\frac{\sigma_3 + \sigma_2 + \sigma_1}{\sqrt{3}} \frac{\pi}{3}} \Phi,$$
$$R\mathcal{T} : \Phi \to \frac{e^{-i\frac{\pi}{12}}}{\sqrt{2}}(1 - i\sigma_1)\Phi \tag{A16}$$

where $\sigma$ rotates two valleys $Q_{1,2}$.

### 3. Kagome lattices

The projective symmetry group on the spinons act as

$$T_1 : f_i \to (-1)^{R_1 + R_2} f_{i+\widehat{R}_1}$$
$$T_2 : f_i \to f_{i+\widehat{R}_2}$$
$$C_6 : f_i \to G(C_6(i)) f_{C_6(i)}, (G(C_6(i)) \text{in fig } 8(a))$$
$$R\mathcal{T} : f_i \to G_R(R(i)) f_{R(i)}, (G_R(R(i)) \text{in fig } 8(b)) \tag{A17}$$

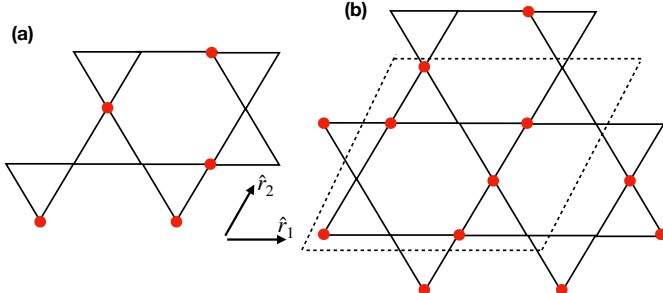

FIG. 8: (a)The gauge transform for $C_6$ where the red dot denotes $-1$ gauge transform and no point denotes trivial gauge transform. With the $2 \times 2$ cell enclosed by the parallelogram, the gauge transform carries a $(\pi, \pi)$ momentum, i.e. gauge transform is invariant upon translation by 4 units of primitive lattice vectors. (b) Gauge transform for $R\mathcal{T}$, with a $2 \times 2$ unit cell. (d) The electron pairing(spin singlet) and hopping amplitude when condensing holons at low-energy fields $\Phi_{1;1} = \Phi_{2;1} = \Phi_{1;2} = -\Phi_{2;2} = 1$, which makes $\Phi^T \tau_2 \sigma_2 \Phi = -2$.The corresponding BCS Hamiltonian is translation invariant, though pairing breaks inversion.

For the holons $z$ they hop in a similar ansatz as for spinons from eq (20), with the ansatz differ by a negative sign (for purely imaginary hopping) for $z_{1,2}$. This results in a degenerate line of lowest-energy states in the Brillouin zone. To simplify the case, we focus on a degenerate pair of momentum points $Q_{1,2} = (0, \pm\pi/4)$ and do not require the states at $Q_{1,2}$ to the lowest energy, since we are concerned about the symmetry properties of the resulting electron Hamiltonian, not energetics. The holons condense at states at $Q_{1,2}$ with a condensation value of $\Phi_{1,2;i}$ for $z_i$, respectively.Note that the lowest-energy state wavefunctions of $z_{1,2}$ are related by $\psi_1(Q_i) = \psi_2(Q_{3-i})^*$. For a generic CSL ansatz with NNN hopping, the lowest-energy states for holons reads,

$$\psi_1(Q_1) = (e^{-5i\pi/12}, \sqrt{2}e^{i\pi/6}, i, -e^{-5i\pi/12}, 0, 1),$$
$$\psi_1(Q_2) = (-e^{-5i\pi/12}, 0, i, -e^{-5i\pi/12}, -\sqrt{2}e^{i\pi/6}, 1), \tag{A18}$$

independent of NNN hopping.

The PSG on low-energy holon fields at $Q_{1,2}$, arranged in the form $\Phi = (\Phi_{1;1}, \Phi_{2;2}^*, \Phi_{2;1}, \Phi_{1;2}^*)$, since $(z_1, z_2^*)$ transform as a vector under the $SU(2)$ gauge group, reads as

$$T_1 : \Phi \to i\sigma_2 \Phi$$
$$T_2 : \Phi \to i\sigma_3 \Phi$$
$$C_6 : \Phi \to e^{i\pi/6} e^{i\frac{\sigma_1+\sigma_2+\sigma_3}{\sqrt{3}}\pi/3} \Phi$$
$$R\mathcal{T} : \Phi \to e^{i\frac{\pi}{4}} \frac{-\sigma_3 + \sigma_1}{\sqrt{2}} \Phi \tag{A19}$$

## Appendix B: Details in the $SU(2)$ theory

Here we list more details about the $SU(2)$ $2\Phi$ theory in Sec. IX and its global symmetry. First, let us consider the square lattice. Because the $\tilde{C}_4$ rotation: $\Phi \to -ie^{i\frac{\pi}{4}\sigma_3\tau_0}\Phi$, the most generic action is:

$$\mathcal{L} = \sum_{a=1,2} |(\partial_\mu - ia_\mu^s \tau^s - iA_\mu^c \sigma_0\tau_0 - iA_\mu^r \sigma_3\tau_0)\Phi_a|^2 + r|\Phi|^2 + \frac{1}{4\pi}Tr[a \wedge da + \frac{2}{3}ia \wedge a \wedge a] - \mathcal{L}_{int},$$

$$L_{int} = g|\Phi|^4 + \lambda_\tau \sum_i |\Phi^\dagger \tau^i \Phi|^2 + \lambda_\sigma \sum_i |\Phi^\dagger \sigma_i \Phi|^2 + \lambda_{\sigma3}|\Phi^\dagger \sigma_3 \Phi|^2 + \lambda_m \sum_{i,j} |\Phi^\dagger \tau^i \sigma_j \Phi|^2 + \lambda_{m3} \sum_i |\Phi^\dagger \tau^i \sigma_3 \Phi|^2 \tag{B1}$$

where $a^s, s = 1, 2, 3$ is an $SU(2)$ gauge field and $\sum_i = \sum_{i=1,2,3}$. $A_c$ is the physical probing field. We have used the symmetry transforms above to simplify the possible quartic interaction. Using identities of

$$\sum_i |\Phi^\dagger \tau^i \Phi|^2 = \sum_i |\Phi^\dagger \sigma_i \Phi|^2,$$

$$\sum_{i,j} |\Phi^\dagger \tau^i \sigma_j \Phi|^2 = -2 \sum_i |\Phi^\dagger \sigma_i \Phi|^2 + 3|\Phi^\dagger \Phi|^2, \tag{B2}$$

, we further simplify the quartic terms to

$$\mathcal{L}_{int} = g|\Phi^\dagger \Phi|^2 + \lambda_1 \sum_i |\Phi^\dagger \sigma_i \Phi|^2 + \lambda_2 \sum_i |\Phi^\dagger \tau^i \sigma_3 \Phi|^2 + \lambda_3 |\Phi^\dagger \sigma_3 \Phi|^2 \tag{B3}$$

With some algebra, we can show that:

$$\sum_i |\Phi^\dagger \tau^i \sigma_3 \Phi|^2 = (\Phi^T \sigma_2 \tau_2 \Phi)^* (\Phi^T \sigma_2 \tau_2 \Phi) + (\Phi^\dagger \sigma_3 \Phi)^2 \tag{B4}$$

Then in terms of $\mathbf{n} = (n_1, n_2, n_3, n_4, n_5) = (\mathrm{Re}\Phi^T \sigma_2 \tau_2 \Phi, \mathrm{Im}\Phi^T \sigma_2 \tau_2 \Phi, \Phi^\dagger \sigma_1 \Phi, \Phi^\dagger \sigma_2 \Phi, \Phi^\dagger \sigma_3 \Phi)$, we can rewrite the quartic terms to be:

$$\mathcal{L}_{int} = g|\Phi^\dagger \Phi|^2 + \lambda_1 (n_3^2 + n_4^2 + n_5^2) + \lambda_2 (n_1^2 + n_2^2 + n_5^2) + \lambda_3 n_5^2 \tag{B5}$$

Note that the $\tilde{C}_4$ rotates $n_3 \to n_4, n_4 \to -n_3$ and thus forbids terms like $n_3^2 - n_4^2$ and $n_3 n_4$.

We will group the interaction into the form:

$$\mathcal{L}_{int} = g|\Phi^\dagger \Phi|^2 + \lambda_0 \mathbf{n} \cdot \mathbf{n} + \lambda(n_1^2 + n_2^2) + \lambda'(n_3^2 + n_4^2) \tag{B6}$$

On triangular and kagome lattice, because $(n_3, n_4, n_5)$ can be rotated to each other by $C_6$, we must have $\lambda' = 0$. When $\lambda$ or $\lambda'$ vanishes, there is a $SO(3) \times O(2)$ global symmetry. At special line $\lambda = \lambda'$, there is a $O(4)$ symmetry. When $\lambda = \lambda' = 0$, the symmetry is further enlarged to be SO(5). To see these enlarged symmetries, it is more convenient to use the $4 \times 2$ matrix field $X$ introduced in Sec. IX C.

### 1. Action in terms of the matrix field $X$

We have $X = \begin{pmatrix} X_1 \\ X_2 \end{pmatrix}$ with

$$X_a = \frac{1}{\sqrt{2}} \begin{pmatrix} \Phi_{a;1} & \Phi_{a;2} \\ -\Phi_{a;2}^* & \Phi_{a;1}^* \end{pmatrix} \tag{B7}$$

under the constraint

$$X_a^* = \tau_2 X_a \tau_2 \tag{B8}$$

One can derive these equations:

$$\mathrm{Tr} X_a^\dagger X_b = \frac{1}{2}(\Phi_a^\dagger \Phi_b + \Phi_b^\dagger \Phi_a), \quad \mathrm{Tr} X_a^\dagger \tau_3 X_b = \frac{1}{2}(\Phi_a^\dagger \Phi_b - \Phi_b^\dagger \Phi_a) \tag{B9}$$

$$\mathrm{Tr} X_a^\dagger(-i\tau_1) X_b = \mathrm{Im}\Phi_a^T i\tau_y \Phi_b, \quad \mathrm{Tr} X_a^\dagger(-i\tau_2) X_b = \mathrm{Re}\Phi_a^T i\tau_y \Phi_b \tag{B10}$$

Then we can derive:

$$\Phi^\dagger \sigma_0 \tau_0 \Phi = \mathrm{Tr} X^\dagger X, \quad \mathrm{Tr} X^\dagger \sigma_3 X = \Phi^\dagger \sigma_3 \Phi \tag{B11}$$

$$\text{Tr}X^\dagger \sigma_1 X = \Phi^\dagger \sigma_1 \Phi, \quad \text{Tr}X^\dagger \sigma_2 \tau_3 X = \Phi^\dagger \sigma_2 \Phi \tag{B12}$$

$$\text{Tr}X^\dagger(-\sigma_2\tau_1)X = \text{Im}\Phi_1^T \sigma_2 \tau_2 \Phi, \quad \text{Tr}X^\dagger(-\sigma_2\tau_2)X = \text{Re}\Phi_1^T \sigma_2 \tau_2 \Phi \tag{B13}$$

$$\text{Tr}X^\dagger \sigma_2 X = 0, \quad \text{Tr}X^\dagger \sigma_{0,1,3}\tau_3 X = 0, \quad \text{Tr}X^\dagger \sigma_{0,1,3}\tau_{1,2}X \tag{B14}$$

The five symmetry breaking order parameters now can be rewritten as: $(n_1, n_2, n_3, n_4, n_5) =$ $(\text{Re}\Phi^T \sigma_2 \tau_2 \Phi, \text{Im}\Phi^T \sigma_2 \tau_2 \Phi, \Phi^\dagger \sigma_1 \Phi, \Phi^\dagger \sigma_2 \Phi, \Phi^\dagger \sigma_3 \Phi) = \text{Tr}X^\dagger(-\sigma_2\tau_2, -\sigma_2\tau_1, \sigma_1, \sigma_2\tau_3, \sigma_3)X$,

$SU(2)$ gauge transformation acts as $X \to XU_g^\dagger$, $a_\mu \to U_g a_\mu U_g^\dagger - iU_g \partial_\mu U_g^\dagger$. The critical theory at $(\lambda = \lambda' = \lambda_3 = 0$ can be rewritten as:

$$\mathcal{L} = \text{Tr}(\partial_\mu X^\dagger + ia_\mu X^\dagger)(\partial_\mu X - iXa_\mu) + r\text{Tr}X^\dagger X + \frac{1}{4\pi}\text{Tr}[a \wedge da + \frac{2}{3}ia \wedge a \wedge a] - \mathcal{L}_{int} \tag{B15}$$

where $a_\mu$ is the abbreviation of $a_\mu^s \tau_s$ with $s = 1, 2, 3$.

The interaction term is now:

$$\mathcal{L}_{int} = g\text{Tr}X^\dagger X + \lambda_0 \mathbf{n} \cdot \mathbf{n} + \lambda(n_1^2 + n_2^2) + \lambda'(n_3^2 + n_4^2) \tag{B16}$$

## 2. Higgs term

We also discuss the Higgs term needed to reach the $U(1)$ theory from $SU(2)$ theory.
It is easy to derive:

$$\text{Tr}X_a^\dagger \tau_3 X_b \tau_3 = \frac{1}{2}(\Phi_a^\dagger \tau_3 \Phi_b + \Phi_b^\dagger \tau_3 \Phi_a) \tag{B17}$$

Then we obtain:

$$\Phi^\dagger \tau_3 \Phi = \text{Tr}X^\dagger \tau_3 X \tau_3, \quad \Phi^\dagger \sigma_3 \tau_3 \Phi = \text{Tr}X^\dagger \sigma_3 \tau_3 X \tau_3 \tag{B18}$$

$$\Phi^\dagger \sigma_1 \tau_3 \Phi = \text{Tr}X^\dagger \sigma_1 \tau_3 X \tau_3, \quad \Phi^\dagger \sigma_2 \tau_3 \Phi = \text{Tr}X^\dagger \sigma_2 X \tau_3 \tag{B19}$$

$$\text{Re}\Phi^T \sigma_1 \tau_1 \Phi = -\text{Tr}X^\dagger \sigma_1 \tau_1 X \tau_3, \quad \text{Im}\Phi^T \sigma_1 \tau_1 \Phi = \text{Tr}X^\dagger \sigma_1 \tau_2 X \tau_3 \tag{B20}$$

Under the duality transformation $X \to e^{i\tau_2\frac{\pi}{4}}e^{i\sigma_3\tau_2\frac{\pi}{4}}X$, $\text{Tr}X^\dagger \sigma_3 \tau_3 X \tau_3 \to -\text{Tr}X^\dagger \tau_3 X \tau_3$, $\text{Tr}X^\dagger \sigma_1 \tau_3 X \tau_3 \to$ $\text{Tr}X^\dagger \sigma_1 \tau_1 X \tau_3$, $\text{Tr}X^\dagger \sigma_2 X \tau_3 \to -\text{Tr}X^\dagger \sigma_1 \tau_2 X \tau_3$

## Appendix C: Monopole quantum numbers in $U(1)_{-2}$ $2\varphi$ theory for CSL-SC transition

Monopoles in the $U(1)_{-2}$ Chern-Simons theory with $2\varphi$ are dressed with 2 units of gauge charges to be gauge-invariant. We consider

$$\varphi^T \epsilon \sigma_i \varphi \mathcal{M}^\dagger (i = 1, 2, 3) \tag{C1}$$

where $\epsilon = i\sigma_2 = \begin{pmatrix} 0 & 1 \\ -1 & 0 \end{pmatrix}$ is the antisymmetric tensor and is used to make symmetric combinations of two gauge charges $\varphi$. The dressed charges $\varphi^T \epsilon \sigma_i \varphi$ transform identically as $\varphi^\dagger \sigma_i \varphi$, which carry the quantum numbers of order

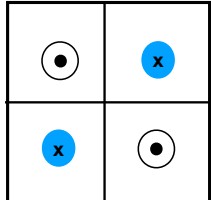

FIG. 9: The holon center that determines monopole $\mathcal{M}_a$ symmetry transforms in CSL-SC transition on square lattices.Close, open circles represent holons with $\pm$ gauge charge, respectively, with $\cdot, \times$ sign indicating a holon of the $\varphi_{1,2}$ species.

parameters $(n_1, n_2, n_5)$ . To identify monopoles that transform as the remaining two order parameters $(n_3, n_4)$, we consider the composite operator which is singlet under the $SO(3)$ symmetry transforming $(n_1, n_2, n_5)$:

$$\mathcal{M}_a^\dagger = (\varphi^\dagger \vec{\sigma}_i \varphi) \cdot (\varphi^T \epsilon \vec{\sigma} \varphi) \mathcal{M}^\dagger. \tag{C2}$$

Since the dressed gauge charges now preserve symmetries, we need only to concern about the Berry phase or symmetry transforms of the bare flux $\mathcal{M}$. The berry phase for rotation symmetries can be inferred from the atomic limit of the holons[71]. $q$ units of gauge charges located at the rotation center contribute to the angular momentum of the monopoles from the Ahronov-Bohm effects, i.e. $l_M = q$. We remark that the atomic limit of holons that preserves space group symmetries and gauge-invariant (total gauge charge 0) is not unique, e.g. the vacuum state is a trivial example. This is due to the fact that we are dealing with an effective low-energy theory after integrating out chern bands from the fermionic spinons. We show in fig 9 a symmetric atomic limit for $U(1)_{-2}$ theories on square lattices that describe CSL-SC and CSL-CDW transitions. Monopoles carry CDW, $d + id$ SC order parameters in these two cases, respectively. Close and open circles represent $\pm$ gauge charges, respectively, and they sum to zero as required by gauge invariance.

We now elaborate on the case for CSL-SC transition listed in table IV.

Note that $\mathcal{M}_a^\dagger$ changes to an anti-monopole under translations or $C_4$ as the symmetry actions exchanges $\varphi_1 \leftrightarrow \varphi_2^*$ in section IV, with opposite charge of $a$. There is hence a phase ambiguity for the symmetry actions that send $\mathcal{M}_a^\dagger \to \mathcal{M}_a$ from a $U(1)$ phase attachment to $\mathcal{M}_a$. We fix the phase by fixing e.g. the $T_1$ action as simply sending $\mathcal{M}_a^\dagger \to \mathcal{M}_a$ with a trivial sign. The relative sign of symmetry actions among $T_{1,2}, C_4$ is meaningful and can be further determined by finding the atomic limit of the holon state that obeys the symmetries in table IV, as shown in fig 9.

The translation

$$T_2 = C_{4A} C_{4B}^{-1} T_1, \tag{C3}$$

where $C_{4A,4B}$ is the four-fold rotation around the plaquette center of $A, B$ plaquettes, i.e. occupied by $\varphi_{1,2}$ charges, respectively in fig 9(a). Since $\varphi_{1,2}$ carry 1 unit of charge for $a$, the holons with charge $\pm 1$ act as a source of angular momentum of $\pm 1$ for the monopole. The monopole hence obtains a factor $-1$ from $C_{4A} C_{4B}^{-1}$ in eq (C3). We thus obtain $T_2$ action. For site-center rotation $C_4$, it is related to translation by

$$C_4 = T_1 C_{4A}. \tag{C4}$$

$C_{4A}$ contributes a factor of $i$ for the monopoles and we arrive at $C_4$ action for monopoles in table IV.

For the action $R_1 \mathcal{T}$ on monopoles, since it exchanges $\varphi_1, \varphi_2^*$ with an additional anti-unitary time reversal action, it sends a monopole to itself up to an arbitrary phase factor. Therefore we obtain the symmetry actions for gauge-invariant operators in $U(1)_{-2}$ theory.