# Peer review of "Deconfined criticalities and dualities between chiral spin liquid, topological superconductor and charge density wave Chern insulator"

_SciPost Physics_

## Round 1 · Referee Report · Anonymous (Referee 1) · 2022-12-22

Report

In this manuscript the authors consider quantum phase transitions between chiral spin liquid (CSL), d+id superconductor (SC) and charge density wave (CDW) Chern insulator. They derive critical theories describing these phase transitions on three different lattice configurations: triangular, square and Kagome. The critical theories are presented in three forms shown to be equivalent to each other: a U(1) theory with two bosons, a U(1) theory for two fermions and an SU(2) theory with two bosons. By matching symmetries of the phases and mapping order parameters the authors demonstrate that the theories are dual to each other. This implies a boson-fermion duality which the authors also derive via a flux attachment approach.

The SU(2) theory is viewed as a unified framework of phase transitions between phases with broken symmetries, since it yields several fixed points corresponding to bi-critical and tri-critical points. The latter are self-dual and possess enlarged global symmetries. The authors argue that deconfined quantum critical points arising in the SC-CDW transitions inherit these properties from the tri-critical points. With this analysis authors draw a connection between deconfined criticality and topologically ordered phases. Additionally, the authors consider possible experimental transport signatures of the CSL-CS transition, deriving universal and Hall conductivities from both sides of the boson-fermion duality.

I think that this paper is well-written and adds new features to the previous work in the field. However, while the main results seem to be correct (I was able to reproduce some of their calculations), my final recommendation comes after the authors address the following points:

  1. In Sec. XI authors argue that the SC-CDW$_{xy}$ transition on a square lattice belongs to the same universality class as the DQCP of the Neel-VBS transition with easy-plane anisotropy. However, numerical simulations [Kuklov et al. Annals of Physics 321, 1602–1621 (2006), Kragset et al. PRL 97, 247201 (2006), D’Emidio and Kaul, PRL 118, 187202 (2017), Desai and Kaul PRB 102, 195135 (2020)] performed on models thought to capture the transition have so far been controversial about whether the transition is second-order and, therefore, if the DQCP exists for the anisotropic case. How does this align with the proposed critical theory of the SC-CDW$_{xy}$ transition? Is the analysis presented in the manuscript only valid if the critical point exists but does not prove that it exists?

  2. In Sec. VII.B on boson-fermion duality authors note that the theory they derive in Eq. (40) can be viewed as a topological generalization of the theory capturing a deconfined quantum critical point in the Neel to VBS transition. They also show that the bosonic theory has a fermionic dual both from the U(1) fermionic ansatz and the flux attachment approach. Deconfined criticality and subsequent bosonization duality for a theory similar to the one in Eq. (40) (when probing fields are set to zero) has been considered by Shyta et al. PRL 127, 045701 (2021). With regard to results of this reference I have a comment to make and a question to raise:

2.1 Comment: In the reference the theory is shown to be self-dual with regard to exact duality transformations on the lattice and a bosonization duality is established to massless Dirac fermions at the DQCP. Therefore, the suggestion by the authors in the last paragraph of Sec. VII.B that for m=0 case the self-duality holds is illustrated by the exact derivations.

2.2 Question. According to the renormalization group analysis in the reference, the theory in Eq. (40) features two fixed points: one belonging to a modified XY universality class with large anomalous dimension and the other having an enlarged O(4) symmetry, with the latter being more stable. Does the critical theory in Eq. (40) provide insights about the stability of the fixed points that it contains? Does the critical point of the CSL-SC transition flow to a more symmetric O(4) case?

  1. Sec. XIV presents transport related calculations of the conductivities obtained from bosonic and fermionic sides of the U(1) critical theory of the CSL-SC transition. The results are reminiscent of the current correlation functions obtained for the U(1)×U(1) topological Abelian Higgs model in [Shyta et al. PRD 105, 065019 (2022)]. There the current correlations functions of the dual bosonic and fermionic theories are shown to be equal up to a universal prefactor. Can one expect that this prefactor is related to the universal conductivity the authors discuss? Is one correct to assume that Eqs. (75) and (77) when written explicitly resemble results of Sec. VI in [Shyta et al. PRD 105, 065019 (2022)]?

  2. In Eq. (49) the BF term coupling a and b gauge fields has an opposite sign to the ½ Chern-Simons of b when the “minus” sign is chosen. I believe that generally the BF term has to have the same sign as the ½ Chern-Simons in the flux attachment (as in Refs. [55, 56] of the manuscript). Otherwise this corresponds to the field $a_\mu$ having an opposite sign on the LHS of Eq. (49). If one accepts this change, the sign of $A_r$ changes when coupled to fermions and a minus sign appears in front of $A_c db$ in Eqs. (51), (52). While this is a minor change, this influences the critical theory, leads to minor inconsistencies with Eq. (45) and may surface in transport calculations.

  3. Although this is a known fact I think it would benefit the reader if the authors elaborate on the connection between the thermal Hall effect and gravitational Chern-Simons. Why should one account for the thermal Hall effect in such systems? How is the gravitational Chern-Simons term generated?

  4. A remark on clarity: As the general theory for a tri-critical point is one of the main results of the manuscript, I would like the authors to explain in more detail how the Lagrangian in Eq. (70) is derived. By that point in the manuscript there has been a fair share of equivalent representations of the critical theories for both bosonic and fermionic U(1) theories, so it is worth stating explicitly which Lagrangians add up to form the tri-critical theory.

Requested changes

I stated conceptual points for the authors to address in the Report field. Here I only enter typos that should be fixed: 1. In the first term of the $L_{SU(2)}$ in Eq. (2) should be a sum over a or $\Phi_a$ should not have an index; 2. Left parenthesis is missing in the first term of $L_{U(1), \vec{m}}$ Eqs. (2), (68). 3. Line under Eq.(7) contains a typo “symmetrey” → “symmetry”. 4. Eq. (11) contains a spinon mean field Hamiltonian to which authors refer as $H_{spinon}$ later in that section. Therefore, a change H → $H_{spinon}$ is required. 5. Related to my previous point, the notation $H_{spinon}$, $H_{holon}$ changes when one goes from the subsection III.A on a square lattice to the subsection III.B on a triangular lattice where the notation $H_\Psi$, $H_Z$ is used. I think that the presentation of the results would benefit from a unified notation. 6. In a line below Eq. (13) the equation for the spinon hopping amplitude is missing a \rangle and should feature different indices $i, j$ on the right-hand side. 7. In the first sentence of Sec. III.C Fig.4(a) should be referenced, not Fig.8(a). 8. Second paragraph of Sec. III.C ends with a comma. 9. Title of Sec. IV.A should not contain a colon or title of Sec. IV.B should too. 10. Eqs. (13) and (20) are the same. Maybe it makes sense to highlight this as the same steps are repeated in Sec. IV.A.1 as in Sec. III.A. 11. In the first paragraph of Sec. V the abbreviation FQHE is used for “fractional quantum Hall state”. Same with the next sentence for the integer quantum Hall state. 12. At the top of the second column of page 18, there is a typo “holonl” → “holon”. 13. Left parenthesis is missing in the first term of the Lagrangian in Eqs. (40), (55), (56), (69), second line of Eq. (70), third term of Eqs. (71) and (72) 14. In a few terms of the form $\varphi^\dagger\sigma_I\varphi$ in Tables IV, VII, VIII, IX, $\varphi_1^\dagger\sigma_I\varphi$ is written instead. I believe that the index of the bosonic field is a typo. 15. In the line above Eq. (58) there is a typo in the abbreviation: “critical theory of the QCP-CDW_{xy} QCP” → “critical theory of the CSL-CDW_{xy} QCP”. 16. $\gamma_\mu$ should contract with $b_\mu$ and $A_{r;\mu}$ terms in the first term of Eqs. (51), (52), (58)

  • validity: -
  • significance: -
  • originality: -
  • clarity: -
  • formatting: -
  • grammar: -

Author:  Xue-Yang Song  on 2023-09-09  [id 3966]

(in reply to Report 1 on 2022-12-22)

We thanks referee 1 for her/his careful reading and useful suggestions. Hereby we submit our revised manuscript with replies listed below.

Reply to referee 1: Indeed whether the DQCP exists and the Neel-VBS transition is second-order is an ongoing issue under investigation. We construct a transition theory with the same low-energy structure as DQCP and whether the transition of SC-CDW* also needs numerics to address. 2. 2.1 We thank the referee for pointing the useful reference and have add a citation of the paper at the end of Sec VII B. 2.2 For the CSL-SC theory eq (40), it is important to have the lambda term that breaks the O(4) rotation among (\varphi_1,\varphi_2) to phase rotations of \varphi_i separately. So the condensed \varphi \propto (1,1) that gives non vanishing SC order. So this theory does not flow to the O(4) fixed point. 3. The conductivity calculation performed in sec XIV should agree with the result in Shyta et al. PRD 105, 065019 (2022). Since both calculations are built on the duality between bosonic and femionic theories, we expect that if one expands the conductivity tensor in eqs (74),(76) of the manuscript, the results will resemble those in sec VI of the mentioned paper by the referee.

  1. We thank the referee for pointing out the inconsistency in applying the boson-fermion duality and have changed the coupling of b and a in eq (49). Subsequent derivation eqs (50-52) were also modified. This amounts to changing the sign of the coupling to probe field A_r. Later results are unaffected.

  2. We add some discussion on CS[g] term and thermal hall effect under eq (30).

  3. We elaborate the derivation of the tricritical theory above eq (70) and added footnote (ref[66]).

---

## Round 1 · Referee Report · Anonymous (Referee 2) · 2023-1-4

Strengths

  1. Very thorough discussion of the subject;
  2. Detailed and careful analysis;
  3. efforts of experimental connection;

Weaknesses

  1. the paper is a bit too long without a clear emphasis;
  2. a lot of phase transitions are discussed, not sure if any of the phase transition is qualitatively beyond previous understanding. If there is such phase transition, it needs to be explicitly highlighted.

Report

I apologize for the late report.

Exotic quantum phase transitions beyond the Landau's paradigm has been a very active and exciting subfield of condensed matter theory. The current manuscript is a very thorough discussion of such quantum phase transitions between different types of ordered phases and also topological order. All these transitions are beyond the classic Landau's paradigm, and connection/analogy between the quantum phase transitions discussed here and previous literatures were also thoroughly discussed. In particular, the web of duality developed in the last five years has been used in this manuscript, which gives us guidance of understanding the nature of these transitions.

Unfortunately I did not have time to go through the manuscript as carefully as the first referee did, and I did not have chance to check the details of the manuscript. My general impression is that, the quantum phase transitions discussed here are interesting, but it is not clear whether they are qualitatively beyond what have been discussed in the existing literature. If there is such qualitatively new example of transition, the authors had better point it out explicitly in the introduction and abstract.

But in any case, I find this paper meets the standard of scipost, and deserves publication.
  • validity: high
  • significance: good
  • originality: good
  • clarity: high
  • formatting: good
  • grammar: good

Author:  Xue-Yang Song  on 2023-09-09  [id 3965]

(in reply to Report 2 on 2023-01-04)

We thank the referee for her/his report. The transition theory studied incorporates both topological order and symmetry breaking phases. This tricritical theory unifies the standard DQCP phenomenology (CDW-SC) and topological phase transitions.

---

## Editorial Decision

resubmitted